# Fostering temporal crop diversification to reduce pesticide use

**Maé Guinet** [1,2] ✉, **Guillaume Adeux** [1,2], **Stéphane Cordeau**[1], **Emeric Courson**[1], **Romain Nandillon** [1], **Yaoyun Zhang**[1] **& Nicolas Munier-Jolain**[1]

Temporal crop diversification could reduce pesticide use by increasing the proportion of crops with low pesticide use (dilution effects) or enhancing the regulation of pests, weeds and diseases (regulation effects). Here, we use the French National DEPHY Network to compare pesticide use between 16 main crops (dilution effect) and to assess whether temporal crop taxonomic and functional diversification, as implemented in commercial farms specialized in arable field crops, could explain variability in total pesticide use within 16 main crops (regulation effect). The analyses are based on 14,556 crop observations belonging to 1334 contrasted cropping systems spanning the diversity of French climatic regions. We find that cropping systems with high temporal crop diversity generally include crops with low pesticide use. For several crops, total pesticide use is reduced under higher temporal crop functional diversity, temporal crop taxonomic diversity, or both. Higher cover crop frequency increases total pesticide use through an increase in herbicide use. Further studies are required to identify crop sequences that maximize regulation and dilution effects while achieving other facets of cropping system multiperformance.

Decrease in crop diversity[1–3], coupled with intensive pesticide use[4], has led to environmental pollution, biodiversity loss, human health concerns and the selection of pest resistance[5–7]. Managing pests, weeds and diseases to ensure sufficient agricultural production and revenues for farmers, while drastically reducing pesticide use, is therefore a major challenge to improve agroecosystem sustainability[8,9] Meeting this objective will require the identification of prophylactic measures, which reduce the incidence of pests, weeds and diseases and the severity of their impact in agroecosystems[8,10]. Crop diversification, that is, increasing the diversity of crop cultivars, crop species and crop functional groups in space and time, is a fundamental pillar of agroecology[11,12] and appears as a promising approach to enhance regulation ecosystem services and reduce anthropogenic inputs[13–16].

Temporal crop diversification can limit pest, weed and disease pressure through several mechanisms. Alternating crop species belonging to different botanical families and sowing periods can disrupt the biological cycles of pests, weeds and diseases, such as soil-borne diseases and soil-dwelling insects or weeds[8,17–19]. The introduction of crops with high competitive ability against weeds or low sensibility to insects, diseases, etc. can also result in reduced pesticide reliance. Authors have also argued that temporal crop diversification could promote more diversified communities of soil microorganisms[13], thereby increasing the probability of maintaining soil pest-pathogen antagonists[18]. The introduction of unharvested crops, such as cover crops, represents another option to diversify crop rotations and improve weed management, especially when combined with no-till strategies[20]. Yet, their potential to reduce pesticide use in other tillage systems is debated[21]. Finally, temporal crop diversification can upscale to diversified landscapes with more complex mosaics of crop fields which strengthen resource dilution effects for pests and limit their spread from one field to another[22,23].

The effect of temporal crop diversification on pesticide use still needs to be quantified. Previous studies have mainly investigated the effect of spatial crop diversification on pest, weed and disease pressure

[1]Agroécologie, INRAE, Institut Agro, Univ. Bourgogne, Univ. Bourgogne Franche-Comté, F-21000 Dijon, France. [2]These authors contributed equally: Maé Guinet, Guillaume Adeux. ✉e-mail: mae.guinet@inrae.fr

or their regulation by natural enemies[13,22,23], and only a few explored how this transcribed in terms of pesticide use[24,25]. Evidence that temporal crop diversification can allow a substantial reduction in pesticide use is currently supported by a few long-term experiments[26–29]. However, these experimental crop rotations tend to be designed to maximize this objective, often at the expense of others (e.g., financial profitability, labor requirements) and hence, are not necessarily adopted by farmers. Studies have also shown that scientists, experimenters, and farmers do not have the same perception of pests, weeds and diseases, which can result in contrasted management practices when confronted with the same field conditions[19,30,31]. Thus, it remains unknown whether temporal crop diversification, as currently implemented in commercial farms, can achieve substantial pesticide reduction. Lechenet et al.[32] identified temporal crop diversification, among other management practices, as an important determinant of low pesticide use in most French production situations, but its effect on pesticide reduction was not quantified. Moreover, further investigation is still required to isolate regulation (i.e., reduction of pesticide use for a given crop in a more diversified crop rotation) from dilution (i.e., introduction of less pesticide-dependent crops in crop rotations) effects, which both underpin temporal crop diversification.

The aim of this study was to assess whether temporal crop diversification, as currently implemented across commercial farms, could reduce pesticide use in 16 main crops. We hypothesized that temporal crop diversification could reduce pesticide use through both regulation and dilution effects. We mobilized data from the French national DEPHY network, which spans 1334 commercial cropping systems (i.e., set of fields that follow the same crop rotation, constraints, and decision rules) resorting to synthetic pesticides across six climatic regions (Fig. 1). Only a small proportion of cropping systems were monitored for more than five years so temporal crop diversification was assessed through space-for-time substitution. Temporal crop diversity was assessed through the diversity of crops present across a given cropping system at a given time point (time point refers to either the two-to-three-year average provided by farmers upon entry in the network or the subsequent annual descriptions, Fig. 1). Temporal crop diversification was assessed through five indicators: functional crop diversity (i.e., effective number of botanical families computed using Hill's numbers[33]), taxonomic crop diversity (i.e., average effective number of crops per botanical family computed using Hill's numbers, weighted by the proportion of each botanical family), crop diversity (i.e., product between the two latter), diversity of sowing periods (i.e., effective number of sowing periods computed using Hill's numbers) and cover crop frequency (i.e., proportion of fields of a given cropping system with cover crops a given year). Pesticide use (total and per pesticide type) was assessed at the crop level for each cropping system and time point by the number of applications at the full recommended dose (i.e., treatment frequency index) and related to crop (dilution effects) and temporal crop diversity indicators (regulation effects). Overall, results suggest that cropping systems with high temporal crop diversity generally include crops with low pesticide use and that crop total pesticide use decreases when temporal crop diversity increases, except for straw cereals.

## Results

### Support for space-for-time substitution hypothesis
One of the working hypotheses of this work was that temporal crop diversity of a cropping system could be captured at a given time point (i.e., the two-to-three-year average provided by farmers upon entry in the network or the subsequent annual descriptions) through crop composition (i.e., the crop species present and their relative proportion in space). To investigate whether crop composition was indeed stable over time for a given cropping system, we retained all cropping systems for which crop composition was described at more than one time point and assessed the percentage of variance in crop composition explained by cropping system identifier. Permutational multivariate analysis of variance (i.e., PERMANOVA) highlighted that cropping system identifier explained 83% of the variance in crop composition ($F_{(764,2427)} = 15.75$, $P < 0.001$). However, PERMANOVA results are known to be sensitive to multivariate heterogeneity of group variances in the case of unbalanced designs, as was the case here ($F_{(764,2427)} = 2.67$; $P < 0.0001$). Nevertheless, the percentage of variance in crop composition explained by cropping system identifier remained stable (i.e., 71 to 88%) when the dataset was split according to the number of observations available per cropping system identifier, thereby showing high support for our space-for-time substitution working hypothesis.

### Relationships between climatic region, crops, and crop diversity indicators
To investigate the effect of climatic region on crop taxonomic and functional diversity, and their product (i.e., crop diversity), one generalized linear mixed effect model (GLMM) with climatic region as the only fixed effect was fit for each of the three response variables (models 1–3 in Supplementary Table 1). Distribution of crop diversity indicators and correlations can be found in Supplementary Figs. 1 and 2a. Analyses of deviance highlighted a significant effect of climatic region on the three response variables (Supplementary Table 1). Crop functional diversity was higher in the Deteriorated Oceanic (2.19 on average) than the Altered Oceanic (1.78) and indistinguishable from the two latter in the four other climatic regions (1.81 to 2.01, Supplementary Fig. 3a). Crop taxonomic diversity was highest in the Mountain (2.23) intermediate in the Semi-Continental (1.85) and Deteriorated Oceanic (1.84), lowest in the Altered Oceanic (1.66) and the Southwest Basin climatic region (1.57), and indistinguishable from the two latter groups in the Oceanic climatic region (1.84, Supplementary Fig. 3b). Crop diversity was highest in the Deteriorated Oceanic (3.92), Mountain (3.86), and Oceanic climatic regions (3.52), lowest in the Altered Oceanic climatic region (2.88), and indistinguishable from the two latter groups in the Semi-Continental (3.43) and Southwest Basin climatic regions (3.03, Supplementary Fig. 3c).

To investigate the effect of crops on crop taxonomic and functional diversity, and their product (i.e., crop diversity), one GLMM with climatic region and crop as fixed effects was fit for each of the three response variables (models 4–6 in Supplementary Table 1). Analyses of deviance highlighted a significant effect of crop on all three response variables (Supplementary Table 1). Cropping systems with high crop functional diversity generally included potato (2.85 on average), spring pea (2.47), alfalfa (2.42), and/or sugar beet (2.42) whereas cropping systems with low crop functional diversity generally included ryegrass (1.56) and/or maize (1.79)(Supplementary Fig. 4). Cropping systems with high crop taxonomic diversity generally included ryegrass (2.95), triticale (2.16), winter barley (2.07), and/or spring barley (2.02) whereas cropping systems with low crop taxonomic diversity generally included potato (1.38), sugar beet (1.55), and/or sunflower (1.74, Supplementary Fig. 5). Indeed, crop taxonomic diversity was highly driven by diversity of cereal crops (r = 0.84, Supplementary Fig. 2b), the most diversified (30 crop species for cereals vs. 27 for legumes, 12 for crucifers, and 1 to 5 for all other families, Supplementary Table 2) and dominant (69% for cereals vs. 7% for legume and 9% for crucifers) botanical family of the dataset. Cropping systems with high crop diversity generally included alfalfa (4.46 on average), spring pea (4.45), ryegrass (4.42), and/or spring barley (4.30) whereas cropping systems with low crop diversity generally included maize (3.41), winter wheat (3.57), and/or sugar beet (3.63, Supplementary Fig. 6).

### Crop dilution effects on pesticide use
To investigate the effect of crop on pesticide use (total, herbicides, fungicides, and insecticides), pesticide use values in the 16 main crops of the dataset were combined and used to fit one GLMM with climatic

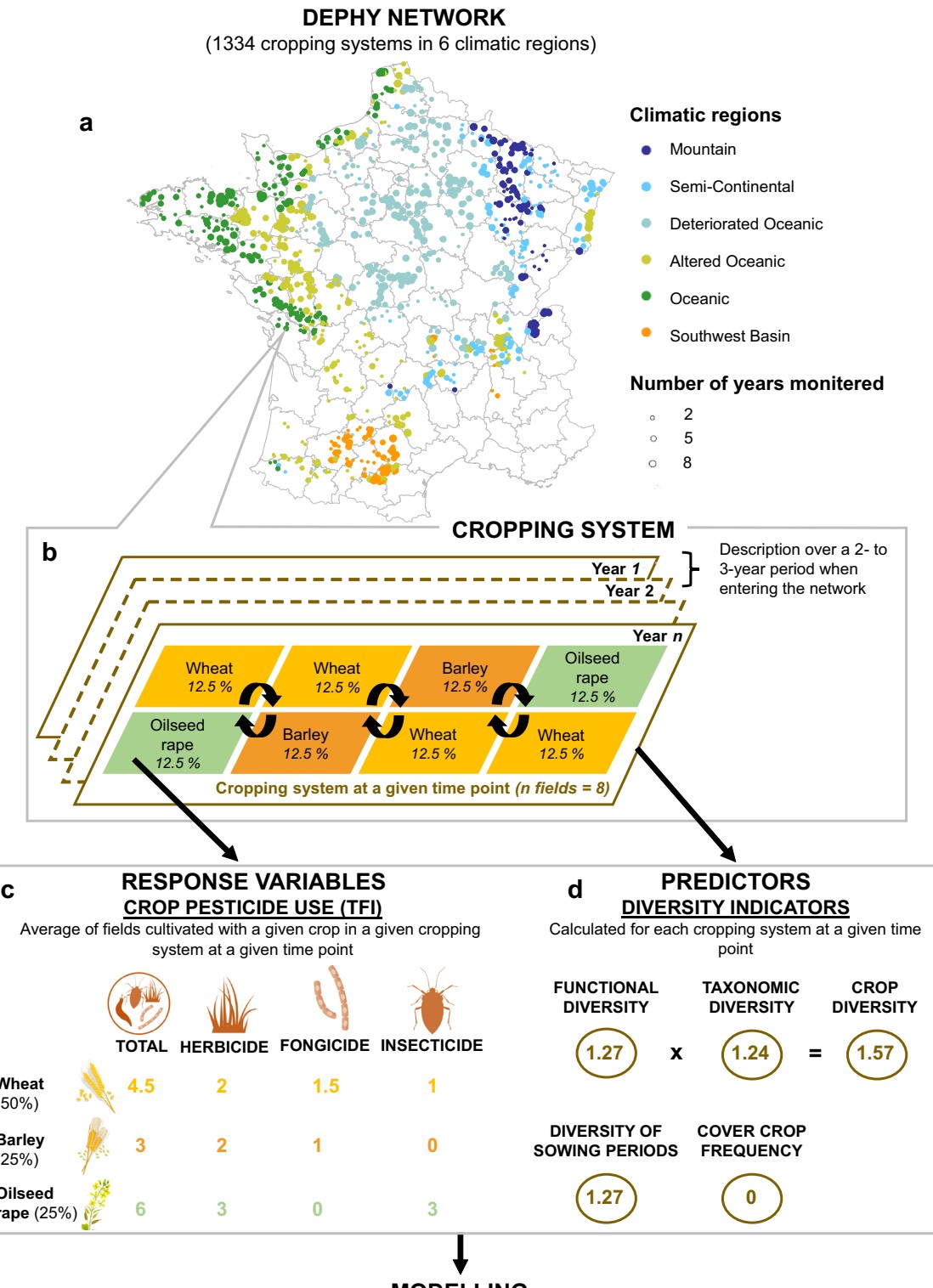

**DEPHY NETWORK**
(1334 cropping systems in 6 climatic regions)

a

**Climatic regions**
- Mountain
- Semi-Continental
- Deteriorated Oceanic
- Altered Oceanic
- Oceanic
- Southwest Basin

**Number of years monitered**
- 2
- 5
- 8

**CROPPING SYSTEM**

b

Description over a 2- to 3-year period when entering the network

Year 1
Year 2
Year n

| Wheat 12.5 % | Wheat 12.5 % | Barley 12.5 % | Oilseed rape 12.5 % |
| Oilseed rape 12.5 % | Barley 12.5 % | Wheat 12.5 % | Wheat 12.5 % |

Cropping system at a given time point (n fields = 8)

c **RESPONSE VARIABLES**
**CROP PESTICIDE USE (TFI)**
Average of fields cultivated with a given crop in a given cropping system at a given time point

|  | TOTAL | HERBICIDE | FONGICIDE | INSECTICIDE |
|---|---|---|---|---|
| Wheat (50%) | 4.5 | 2 | 1.5 | 1 |
| Barley (25%) | 3 | 2 | 1 | 0 |
| Oilseed rape (25%) | 6 | 3 | 0 | 3 |

d **PREDICTORS**
**DIVERSITY INDICATORS**
Calculated for each cropping system at a given time point

| FUNCTIONAL DIVERSITY | | TAXONOMIC DIVERSITY | | CROP DIVERSITY |
|---|---|---|---|---|
| 1.27 | x | 1.24 | = | 1.57 |

| DIVERSITY OF SOWING PERIODS | COVER CROP FREQUENCY |
|---|---|
| 1.27 | 0 |

**MODELLING**
CROP TFI ~ CLIMATIC REGION + CROP SPECIES x DIVERSITY INDICATORS
+ random effects

region and crop as fixed effects for each response variable (models 7–10 in Supplementary Table 1). The distribution of crop total pesticide use and each pesticide type can be found in Supplementary Fig. 7. Analyses of deviance highlighted a significant effect of crop on total pesticide use and on all three pesticide types (Supplementary Table 1). Pesticide use (total and all three types) was extremely low for grasslands, ryegrass, alfalfa, and cereal-legume mixtures (Fig. 2, Supplementary Fig. 8).

Among grain and industrial crops, total pesticide use was lowest for soybean and sunflower (due to lower fungicide and insecticide use), intermediate for all cereals (maize, spring oat, triticale, durum wheat, winter barley, winter wheat) and spring pea, and highest for oilseed rape (due to higher insecticide use), sugar beet (due to higher herbicide and insecticide use), and to an even greater extent potatoes (due to higher insecticide use and particularly fungicide use).

**Fig. 1 | Illustration of the DEPHY network and data handling procedure.**
**a** Geographical localization of the cropping systems considered in this study. Cropping systems (points) are colored depending on the climatic region to which they belong. Point size is proportional to the number of years the cropping system was monitored. **b** Cropping systems are defined as a set of fields within a farm that are subject to the same crop rotation, constraints, and decision rules (*N* = 1334). For a given cropping system, the diversity of the crop rotation can hence be captured at different time points (time points refers to either the two-to-three-year average provided by farmers upon entry in the network or the subsequent annual descriptions) through space-for-time substitution. **c** For each of the 16 main crops included in the cropping systems considered (*N* = 14,456), pesticide use was assessed using the treatment frequency index (TFI). TFI quantifies the number of pesticide applications at the full recommended dose. Total TFI was decomposed into herbicide, fungicide and insecticide TFI. **d** For each cropping system and time point (*N* = 3761), five diversity indicators were calculated and used as predictors for pesticide use at the crop level: functional diversity, taxonomic diversity, crop diversity, diversity of sowing periods and cover crop frequency.

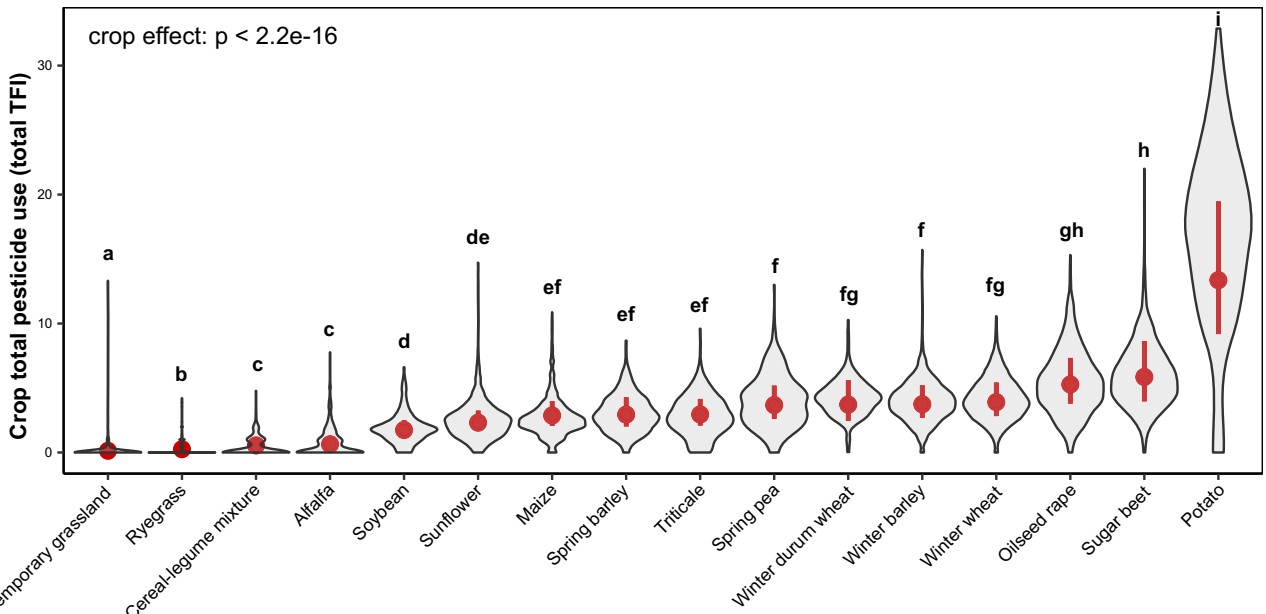

**Fig. 2 | Violin plots highlighting the effect of the 16 main crops on crop total pesticide use.** Crop pesticide use was assessed using the Treatment Frequency Index (TFI), which quantifies the number of applications at the full recommended dose. Violin plots were created using the observed data. Red points represent estimated marginal means and red lines associated 95% Wald confidence intervals (computed using the delta method) obtained from a generalized linear mixed effect model (model 7 Supplementary Table 1). Estimated means were averaged (or "marginalized") across climatic regions. Violin plots sharing the same letter are not significantly different at *P* < 0.05 based on a set of two-tailed Wald tests, which assess whether the pairwise differences in crop means are different from zero. Multiple comparisons were adjusted using the false discovery rate method. A total of 14,456 observations (i.e., one crop of a cropping system at a given time point) were available for this figure.

### Regulation effects of crop diversification on pesticide use

To investigate the effect of crop diversification on crop pesticide use, diversity indicators (crop functional and taxonomic diversity, their interaction, and cover crop frequency) were added to the four previous models (models 11–15 in Supplementary Table 1). Higher-order interactions between crop and crop diversity indicators were selected based on AIC. Grasslands, ryegrass, alfalfa, and cereal-legume mixtures were however not considered in these models due to overall low pesticide use (i.e., less than 0.7, Fig. 2). Similarly, crops with null to low fungicide (i.e., less than 0.1) and/or insecticide use (i.e., less than 0.2) were not included in the corresponding models. For herbicide use, an alternative model focusing on diversity of sowing periods was investigated (model 13, Supplementary Table 1) but was not as parsimonious (AIC = 31,253) as the one focusing on crop functional diversity (AIC = 31,211). For all four response variables, analyses of deviance on the most parsimonious model highlighted a significant effect of crop and the interaction between crop and crop functional diversity (Supplementary Table 1). The interaction between crop and crop taxonomic diversity was significant for all models except the fungicide use model, for which the main effect of crop taxonomic diversity was nevertheless significant. Figure 3 shows the combined effect of crop taxonomic and functional diversity on total pesticide use in the twelve crops considered for that analysis.

Overall, increasing crop functional diversity from 1 to 4 significantly reduced total pesticide use in soybean by 23%, in sugar beet by 21%, in sunflower by 20%, and in maize by 19% (see Supplementary Table 3 for significance of slopes and standardized effect sizes for all crops). These effects were mediated by the fact that crop functional diversity reduced herbicide use for all four crops (soybean: −31%; sugar beet: −21%; sunflower: −20%; maize: −13%, Supplementary Fig. 9) and fungicide use for sugar beet (−34%, Supplementary Fig. 10). Increasing crop taxonomic diversity from 1 to 4 significantly reduced total pesticide use in potato by 37%, in oilseed rape by 20%, in spring pea by 19%, and in maize by 14% (Supplementary Table 3). These effects were mediated by a negative effect of crop taxonomic diversity on fungicide use for all crops (−15%, Supplementary Fig. 10), on herbicide use for maize (−9%) and potato (−61%, Supplementary Fig. 9), and on insecticide use for spring pea (−65%, Supplementary Fig. 11). Crop functional and taxonomic diversity had no significant effect on total pesticide use for all straw cereals (Supplementary Table 3). Increasing cover crop frequency from 0 to 1 significantly increased total pesticide use by 13% and this effect was mediated by a 10% increase in herbicide use.

### Discussion

Results highlighted that temporal crop diversification could reduce total pesticide use in all dominant field crops in which pesticides are

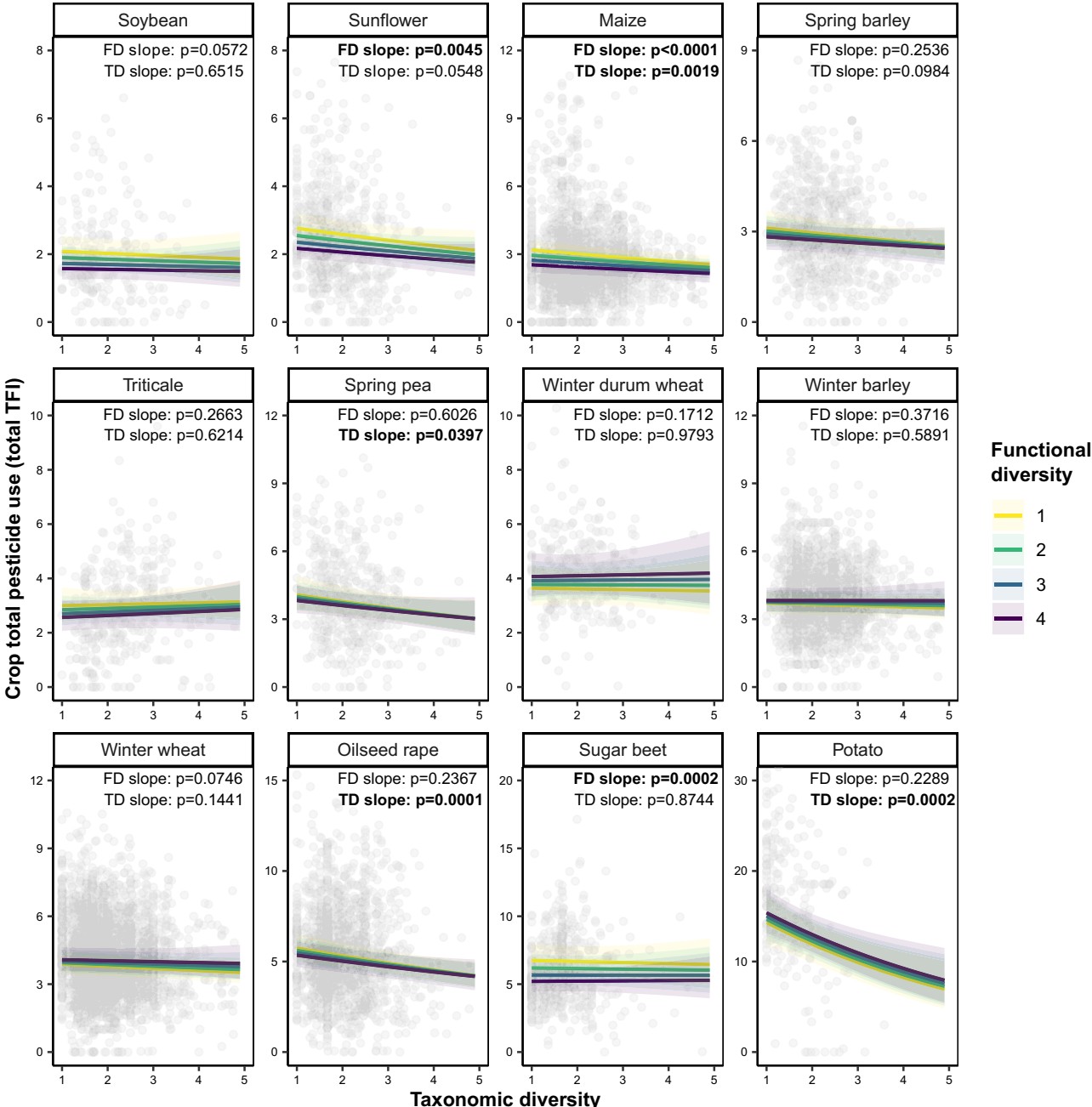

**Fig. 3 | Relationship between crop diversity (taxonomic and functional) and crop pesticide use for 12 main crops.** Crop pesticide use was assessed using the Treatment Frequency Index, which quantifies the number of applications at the full recommended dose. Crops with null to low total pesticide use (i.e., less than 0.7) were not included in this analysis. One data point represents a crop in a given cropping system at a given time point (either the two-to-three-year average provided by farmers upon entry in the network or the subsequent annual descriptions). When a given crop was grown over multiple fields of a given cropping system at a given time point, values were averaged. Regression lines represent population-level predictions (i.e., do not account for random effects) based on a Tweedie mixed effect model (model 11 in Supplementary Table 1). Predictions were averaged (or "marginalized") across climatic regions. The upper and lower limits represent Wald 95% confidence intervals (CI) obtained using the delta method. Slopes of Functional (FD) and Taxonomic diversity (TD) were tested against zero for each crop using a two-tailed Wald test. *P*-values (p) are provided and written in bold for slopes significantly different from zero at *P* < 0.05.

commonly applied (7 out of 12 crops), exception made of winter straw cereals (5 out of 12 crops). Crop taxonomic diversification reduced total pesticide use in potato, oilseed rape, and spring pea, and was highly correlated to the diversity of cereal crops. Crop functional diversification reduced total pesticide use in sunflower, soybean, and sugar beet. Both diversity indicators reduced total pesticide use in maize. These contrasted responses may reflect the different facets of crop diversity (either taxonomic or functional) commonly lacking in the dominant crop rotations in which these crops are included[34].

Indeed, potatoes are grown in Northern France, where mild climate, deep soils, and market outlets for industrial crops promote high crop functional diversity but still allow for further crop taxonomic diversification. Spring pea and oilseed rape are grown together with winter cereals in Central France, where temperate climate, shallow calcareous soils and lack of market outlets for industrial crops mainly allow the introduction of functionally similar crop species (e.g., spring barley)[26]. Sunflower (South-Western France), soybean (South-Western and Eastern France), and sugar beet (Northern France) are often associated

with a high proportion of cereals (either wheat or maize) and such rotations are preferentially diversified with crops belonging to other botanical families, such as oilseed rape, soybean in the case of sunflower (and vice versa)[27], or flax, green pea, or green beans in the case of sugar beet. Maize is grown in highly simplified cropping systems (either as a monoculture or with wheat or ryegrass) which can be diversified through multiple pathways[35], including new functional types or not, thereby explaining the negative effect of both diversity indicators on total pesticide use in maize. Two non-exclusive hypotheses could be formulated to explain the lack of effect of crop temporal diversification on total pesticide use for all winter straw cereals. First, farmers could take advantage of the diversity of efficient herbicides in these crops to manage dicotyledonous weeds at the cropping system scale. Second, the economic importance of straw cereals in French cropping systems could favor risk aversion and a low return delay of these crops even in diversified cropping systems.

The effects of crop diversification on total crop pesticide use appeared weaker than those reported in previous studies focusing on diversified crop rotations and pest pressure. In this study, increasing crop functional diversity from 1 to 4 reduced crop total pesticide use by 19 to 23% depending on the crop considered, and an identical increase in crop taxonomic diversity reduced crop total pesticide use by 13 to 31% depending on the crop considered. In comparison, a global meta-analysis based on 298 paired observations from 54 studies across six continents[19] showed that diversifying simple rotations (i.e., monocultures or two-year rotations) reduced weed density by 49%, although no significant effect was found on weed biomass. Indeed, decrease in pest, weed and disease pressure may have a limited impact on pesticide use if the pressure remains above the thresholds that trigger the decision to use pesticides. Furthermore, it is important to note that increasing crop functional diversity from 1 to 4 would imply a drastic change in crop rotations (e.g., introducing three new and functionally distinct crop species in a maize monoculture). Results also highlighted significant differences in total pesticide use between crops. Crops associated with livestock (feed crops such as grasslands, ryegrass, alfalfa, and cereal-legume mixtures) showed the lowest level of pesticide reliance, followed by sunflower and soybean, whereas dominant (winter cereals, oilseed rape) and industrial crops (potatoes, sugar beet) showed an intermediate to high level of pesticide reliance, respectively. These differences are paramount because they stress that crop diversification could also affect total pesticide use through dilution effects (for example when sunflower or soybean is introduced in a crop rotation composed essentially of cereals), in addition to the regulation effects previously described. Nevertheless, the importance of such dilution effects will be highly dependent on the level of pesticide reliance of the crops composing the original crop rotation and of the newly selected crop species. For example, introducing potatoes in a cereal-based rotation could even have the opposite effect and increase total pesticide use. Diversified cropping systems could combine both regulation and dilution effects. Indeed, analyses showed that diversified cropping systems usually included ryegrass and alfalfa, i.e., two crops in which pesticides are not usually applied but which require specific markets, machinery, and/or livestock integration, and spring barley, i.e., the least pesticide reliant straw cereal of the dataset.

Temporal crop diversification reduced total pesticide use mainly through a reduction in herbicide use. Surprisingly, no significant effect of diversity of sowing periods was found on herbicide use, even though previous studies showed that temporal variation of sowing dates was the main mechanism explaining the negative effect of more diversified crop rotations on weed abundance[19,36]. A possible explanation for these unexpected results could be a greater ease to diversify crop rotations with species known to have a suppressive effect on weeds (e.g., winter straw cereals such as triticale, oats, rye, cereal-legume mixtures or alfalfa) than through the introduction of new sowing periods, which may result in the selection of crops not adapted to local climatic and agronomic constraints. Temporal crop diversification significantly reduced insecticide use in spring pea only. This result was also unexpected as the majority of pests targeted in field crops are mobile at the landscape scale (e.g., aphids, cabbage-stem flea beetle, European corn borer…). Possible explanations include an increase in the return delay between susceptible crops (e.g., pea midge, field thrips) and potential covariance between temporal and spatial crop diversification, which is well-known to promote natural enemies as well as to reduce pest pressure[13,22,23] and insecticide use[24,25]. Fungicide use was highly associated with the management of soil-borne diseases on potato, namely potato blight, a disease specific to the Solanaceae family. Prophylactic measures for the management of potato blight include increasing the return delay of potato[37] in order to break the life cycle of the disease. This can be achieved by increasing the proportion of non-host crops already in the rotation, usually winter cereals, in order to maintain a suitable return delay for other industrial crops (e.g., sugar beet) or legumes (i.e., four to five years). Finally, farmers mobilizing a greater diversity of crops may more frequently resort to variety mixtures, which have been shown to increase disease resistance in the case of winter straw cereals[38,39].

Increasing cover crop frequency resulted in a slight increase in herbicide use. On the one hand, this result may appear surprising as numerous studies have shown that cover crops could reduce the development of weeds during the fallow period[21]. A recent meta-analysis even reported that cover crops had a greater effect on pest and disease control (+125%) than intercropping (+66%) or agroforestry (+59%)[13]. However, the few studies focusing on the carry-over effect of cover crops on weeds in the subsequent crops have reported limited effects[40] or an overriding effect of management intensity (e.g., tillage, herbicides)[41]. On the other hand, this result may appear as expected because the implementation of cover crops does not allow the implementation of other effective weed management tools, such as false seedbed practices, and because herbicides (e.g., glyphosate) are often used for the simultaneous destruction of both cover crops and weeds growing within. Alternatively, this positive relationship could reflect a confounding effect between cover crop frequency and tillage intensity. Permanent cover is one of the pillars of conservation agriculture, which is often associated with increased herbicide reliance, in part due to the absence or reduction of tillage intensity[42].

Crop diversification can also be viewed as a means to foster a greater diversity of pest, weed, and disease management tools. For example, introducing a large inter-row winter crop (e.g., winter faba bean) in winter cereal-based rotations can facilitate mechanical weeding operations such as hoeing. Introducing a summer crop (e.g., soybean) in the same rotation can generate a diversification of sowing periods and allow to introduce false seedbed practices at a time of year when a crop is usually in place. Similarly, in most grain-based cropping systems, the introduction of a perennial crop can generate new selection pressures on weeds, such as mowing[43–45].

It appears likely that crop diversification could be increased further even in the most diversified cropping systems of the DEPHY network, whether in terms of crop taxonomic or functional diversity and/or pest management practices. Reasons can be manifold: need to invest in specific machinery, lack of knowledge or markets and changes in farm labor organization[46]. Moreover, it cannot be ignored that farmers within the DEPHY network may diversify their crop rotations for other reasons than pesticide reduction, such as increasing profitability (e.g., potato) or protein autonomy for livestock. Hence, we argue that the effect of crop diversification on pesticide reduction observed in this study may be potentially greater, as observed in other long-term experiments where crop diversification was specifically implemented to reduce pesticide use[27,29,47]. As a matter of fact, the potential for pesticide reduction may still exist under current farming conditions if regulation effects are not integrated by farmers. Indeed,

other farming practices or market opportunities may explain pesticide use, such as farmers' perception of pests, weeds and diseases[19,31] or crop end use (e.g., feed barley vs. malting barley).

Future experiments could attempt to disentangle the different effects underlying temporal crop diversification (i.e., regulation, dilution, and diversification of pest management tools fostered by crop diversification) to highlight their relative importance on pest pressure and hence, guide cropping system design. Further studies are required to quantify the effect of spatial crop diversification (e.g., cultivar or species mixtures, agroforestry) on pesticide use and their relative importance compared to temporal crop diversification (e.g., crop rotation, cover crops). Furthermore, the effect of temporal diversification was here assessed through time for space substitution (with caution that all fields followed the same crop rotation within a given cropping system). Future studies focusing on crop order and crop return delay (i.e., truly temporal datasets) could shed light on specific crop sequences (i.e., preceding crop–crop, rotation) and their characteristics that enhance pest, weed and disease control, and lower pesticide use.

Crop diversification will only be implemented by farmers and incentivized by political instances if it allows to combine pesticide reduction with several other objectives, such as achieving reasonable financial profitability and food sovereignty, respectively. Trade-offs between crop diversification and other objectives were not investigated in this study but analyses were based on crop rotations implemented in real farms, which all seek financial profitability. Previous work has shown that increasing temporal crop diversity could have a positive effect on cereal yields[48] but productivity assessments at the rotation scale are still required for diversified systems. Nilsson et al.[49] showed that functional diversification could have a positive effect on the profitability of Swedish farms but this relationship could be driven by the introduction of high-value industrial crops which show high pesticide reliance, as shown here. Nevertheless, Lechenet et al.[32] showed that crop diversification could contribute to pesticide reduction across a diversity of farming contexts and that no relationship between pesticide use and profitability (or productivity) could be established for the majority of farms within these contexts[50]. Identifying the different forms of crop diversification required to reach cropping system multiperformance represents a major avenue for agronomical and ecological research.

## Methods
### Data collection
Data was mobilized from the DEPHY network, a national network of commercial farms established within the framework of the French national plan Ecophyto[51], which planned to halve pesticide use across the French territory. Farmers joined the network on a voluntary basis and aimed to reduce pesticide use, albeit without jeopardizing profitability (i.e., pesticides are sprayed whenever farmers think they are required to maximize farm profitability). Only cropping systems (i.e., set of fields that are subject to the same crop rotation, constraints, and decision rules) that resorted to synthetic pesticides and integrated field crops ($N = 1334$) were included in this study. Pure grassland cropping systems were discarded. Cropping systems covered a diversity of climatic regions, soil types, crops, and farming practices[32].

Upon entry into the network, farmers were asked to describe one of their cropping systems, in the case of several present on the farm. For this initial description, farmers described cropping systems in terms of crop proportion (i.e., frequency of each crop over all fields belonging to the same cropping system) and management practices associated with each of these crops (including whether or not they were preceded by cover crops), based on realized farming practices over the last two-to-three-year description period (Supplementary Fig. 12). The proportion of each crop remained stable over the course

of this two-to-three-year description period whereas farming practices could slightly vary, in response to contrasted weather conditions and pest pressure (and were hence averaged by farmers during the network entry questionnaire). In the following years, farmers were asked to provide—on an annual basis—information on the crops and management practices associated with all fields of the cropping system monitored (Fig. 1). Only half of the cropping systems were monitored more than once over time (Supplementary Fig. 13).

The number of fields monitored was defined to encompass the diversity of crops present in a given cropping system but only cropping systems described over a minimum of 8 fields were included as a safeguard. One of the working hypotheses was that crop diversity over a given set of fields at a given time point (i.e., the two-to-three-year average provided by farmers upon entry in the network or the subsequent annual descriptions) reflected temporal crop diversity (space-for-time substitution hypothesis). A given crop in a given cropping system at a given time point was considered as one observation, yielding a total of 16,822 observations (14,456 observations when only considering the 16 main crops).

### Climatic regions and soil classification
As climate is well-known to drive pest, weed and disease pressure and the potential choice of crops (and hence pesticide use), cropping systems were classified according to their geographical location into six climatic regions (Fig. 1; Supplementary Fig. 14), namely, Altered Oceanic, Deteriorated Oceanic, Mountain, Oceanic, Semi-Continental and Southwest basin[52]. The Mountain climate is characterized by high precipitations and low mean annual temperature, and a predominance of winter wheat (33.2%), maize (16.7%), oilseed rape (15.1%), winter barley (12%), and to a lesser extent, spring barley (7.5%) (Supplementary Fig. 15). The Altered Oceanic climate is characterized by intermediate rainfall and relatively high mean annual temperature, the Oceanic climate by high rainfall and low mean annual temperature, and the Semi-Continental climate has slightly lower rainfall and warmer temperatures than the Mountain climate. All three climatic regions have a predominance of winter wheat (on average 27.8%), maize (on average 29.8%) and a high proportion of feed crops, such as grassland, alfalfa, cereal-legume mixture, ryegrass and triticale, compared to the other three climatic regions (20.8% vs 6.6%). The Deteriorated Oceanic climate is characterized by low rainfall and intermediate mean annual temperature, and industrial crops such as sugar beet and potato (5.5%), alongside winter wheat (36.4%), oilseed rape (12.1%) and winter barley (9.0%). The Southwest basin climate is characterized by low rainfall and high mean annual temperature. In addition to winter wheat (23.3%) and maize (21.5%), the latter climatic region has the highest proportions of sunflower (17.1%), winter durum wheat (12.7%) and soybean (4.5%) compared to the other climatic regions

Like climate, soil type can determine crop selection and weed and soil-borne disease pressure. Each cropping system was assigned to one of sixteen soil types on the basis of the French soil map[53]. By cross-referencing the geographical positions of the cropping systems and the map of soils on a GIS tool, we assigned to each cropping system the most frequent soil type of the municipality where the cropping system was located. The interaction between climatic region and soil type was included as a random effect in the analyses (see "Statistical analyses" section).

### Diversity indicators
The 3761 combinations of cropping systems and time points (2–3-year average description when entering the network or annual description afterward) considered encompassed 103 cash crops. Across all cropping systems and time points, the 16 main crops represented 96.5% of the total crop proportion when combined. Within the 16 main crops, crop proportion ranged from 30.7% for winter wheat to 0.9% for potato (Supplementary Table 2).

For each cropping system and time point, five diversity indicators were computed. Crop diversification can be calculated based on crop richness and crop relative abundance using the Hill Index to estimate the effective number of crops (i.e., the number of equally abundant crops required to yield the same value of a diversity measure)[33]. This index is calculated using the following equation where $i$ corresponds to the different crops ($i = 1, ..., n$) and $p_i$ to the proportion of each crop in the cropping system.

$$^{q=1}D = \exp\left(-\sum_{i=1}^{n} p_i \times \ln(p_i)\right) \quad (1)$$

An order of $q = 1$ was chosen to give similar weight to rare and abundant crop species[33]. However, this index does not take into account the proximity of crops in terms of functional traits involved in pest, weed and disease regulation. Crops belonging to different botanical families may differ in their susceptibility to pests and diseases and in their competitiveness against weeds. Yet, crop diversification within a botanical family can lead to the introduction of more robust crops that manage biotic interactions and help break pest and disease cycles and control weeds. To account for crop functional similarity, the Hill index was decomposed into two crop diversity indicators that capture (1) functional diversity ($^1D_{FD}$, the number of equally abundant botanical families), and (2) taxonomic diversity ($^1D_{TD}$, the average number of equally abundant crops per botanical family). The decomposition of crop diversity is equivalent to that performed by Nilsson et al.[49] Each crop $i$ was assigned to one of the 14 botanical families (Supplementary Table 2) and functional diversity ($^1D_{FD}$) was calculated analogously to $^1D$ with $p_k$ the proportion of each botanical family $k = 1, ..., 14$. $p_k$ was calculated by summing the proportion of all crops $i$ belonging to the same botanical family $k$. A botanical family-specific Hill index ($^1D_k$) was calculated from the relative proportion of crops within each botanical family. Taxonomic diversity ($^1D_{TD}$) was obtained by summing all $^1D_k$ weighted by the proportion of each botanical family ($p_k$) within each cropping system at a given time point:

$$^1D_{TD} = \sum_{k=1}^{14} p_k \times {}^1D_k \quad (2)$$

The multiplication of $^1D_{FD}$ and $^1D_{TD}$ results in the effective number of crops. Diversifying sowing periods is expected to sustain weed management by disrupting the life cycles of weeds[19]. Each annual crop was classified as a fall, winter, spring, or summer crop based on its sowing period, while perennial crops were grouped into a fifth category (Supplementary Table 2). Based on (1), the diversity of sowing periods was calculated as a fourth crop diversity indicator, with $p_j$ the proportion of crops belonging to the sowing period $j$. Lastly, the proportion of crops preceded by cover crops was calculated.

Diversity indicators were not rarefied based on the number of fields monitored. Indeed, the number of fields monitored was chosen by the farmer to reflect crop composition (i.e., the crop species present and their relative proportion in space) with the lowest number of fields possible in order to limit monitoring time (i.e., unnecessary to monitor 20 fields for maize monocropping).

### Treatment Frequency Index (TFI)
Reliance on pesticide use was quantified through the Treatment Frequency Index (TFI). The TFI quantifies the mean number of treatments per hectare, for a given crop and year with commercial products (that possibly contains several active ingredients), weighted by the ratio of the dose used to the reference dose[54]:

$$TFI = \sum_{i=1}^{n} \frac{D_i . S_i}{Dh_i . S_t} \quad (3)$$

where $D_i$, $Dh_i$, and $S_i$ are, respectively, the applied dose, the reference dose, and the treated surface area of the field for the spraying operation $i$, and $S_t$ is the total field surface. The applied doses of commercial products were reported by farmers for each field and year. Each commercial product was classified according to the target: fungal pathogens (fungicides), insects (insecticides) or weeds (herbicides). As recommended by the French Ministry of Agriculture for TFI computation, the lowest registered dose for a given commercial product and a given crop was selected as a reference dose for TFI computation (different registered doses can be available for a given commercial product depending on the target pest and crop). All reference doses were extracted from the E-phy online database provided by the French Ministry of Agriculture[55]. Seed coating with chemical pesticides was included in the TFI computation (1.0 additional TFI point for each crop sown with coated seeds). Less hazardous and non-chemical pesticides (according to the 'biocontrol' list of the French Ministry of Agriculture[56]) were excluded from TFI computation. It should be noted that TFI quantifies pesticide reliance but does not measure the ecotoxicological impact of pesticides.

### Statistical analyses
All statistical analyses were carried out with R software version 4.3.1[57].

### Multivariate analysis: assessment of the space-for-time substitution hypothesis
A working hypothesis was that temporal crop diversity of a cropping system could be captured at a given time point (i.e., the two-to-three-year average provided by farmers upon entry in the network or the subsequent annual descriptions) through crop composition (i.e., the crop species present and their relative proportion in space). To investigate whether this hypothesis was valid, we retained all cropping systems for which crop composition was described at more than one time point ($N = 765$) and assessed the percentage of variance in crop composition explained by cropping system identifier, using permutational multivariate analysis of variance (i.e., PERMANOVA, function adonis2 available in the R package vegan)[58], a non-parametric alternative to multivariate analysis of variance (MANOVA) which does not require multivariate normality. A Bray-Curtis dissimilarity matrix was computed based on the crop composition table (3192 rows for the different combinations of cropping systems and time points and 103 columns for all the crop species present in the dataset) and used as the response for the analysis. Cropping system identifier was the only predictor considered in this analysis. Permutations ($N = 1000$) were restricted to climatic regions for significance testing.

Distance-based analysis (such as PERMANOVA) is known to confound location and dispersion effects[59] in the case of unbalanced designs[60] (such as here, considering 2 to 11 observations per cropping system identifier were initially used). To identify whether PERMANOVA results were influenced by dispersion effects, multivariate homogeneity of group dispersions (variances) was assessed. For each group (cropping system identifier), multivariate distances between each data point (combinations of cropping system identifiers and time points) and the group centroid (spatial median) were computed using the R function betadisper. The multivariate distances were then analyzed with ANOVA (R function ANOVA) to test whether within-group dispersion significantly differed between groups.

Multivariate heterogeneity of group dispersions was identified so the dataset ($N = 765$) was split into 10 balanced subdatasets (i.e., each subdataset was composed of cropping system identifiers with the same number of observations: 2, 3, ... 11) and PERMANOVAs were

carried out on each subdataset separately (with the same modeling choices as described above). This allowed us to test the robustness of the initial analysis ($N = 765$) to the lack of multivariate homogeneity of group dispersions, which was initially of importance due to the unbalanced design.

### Regression analyses: investigating relationships between climatic region, diversity indicators and pesticide use

All regression analyses were performed with generalized linear mixed effect models (GLMM) with a Tweedie distribution and a log link function, using the R package and function glmmTMB[61]. The Tweedie distribution is a flexible distribution for continuous positive data which can adequately handle zero inflation thanks to its extra index parameter $p$, which yields a compound Poisson-Gamma distribution when $1 < p < 2$[62,63]. All continuous predictors were scaled prior to analyses (i.e., centered to the mean and divided by one unit of standard deviation). All random effect structures were defined a priori based on expert knowledge and were not subject to selection.

Overall, four sets of regression models were investigated. A first set of regression models (models 1–3 in Supplementary Table 1) was fit at the cropping system level (i.e., one observation corresponds to one cropping system at one time point, $N = 3761$) and aimed to investigate the effect of climatic region on crop taxonomic and functional diversity, as well as their product (i.e., crop diversity). Only climatic region was included as a fixed effect for all three response variables. Random effects included random intercepts for cropping system identifier, period of description and interactions between climatic region and period of description, as well as between soil type and climatic region.

The three following sets of regression models were fit at the crop level (i.e., one observation corresponds to one crop in one cropping system at one time point) and focused on the 16 main crops of the dataset ($N = 14,456$). Observations for minor crops were discarded.

The second set of regression models (models 4–6 in Supplementary Table 1) aimed to investigate the influence of the 16 main crops on crop taxonomic and functional diversity, as well as their product, while controlling for climatic region effects. Hence, one model with climatic region and crop as fixed effects was fit for each of the three response variables. Random effects included random intercepts for period of description and interactions between crop and climatic region, crop and period of description, climatic region and period of description, soil type and climatic region, and between crop, climatic region, and period of description. For these analyses, cropping system identifier and its interaction with crop and description period were not included as random intercepts because of conflict between fixed and random effects.

The third set of regression models ("dilution effect models", models 7–10 in Supplementary Table 1) aimed to investigate the effect of crop on pesticide use (total, herbicide, fungicide, and insecticide) while controlling for climatic region effects. Hence, one model with climatic region and crop as fixed effects was fit for each of the four response variables. All random effects previously described were included in these analyses (i.e., the same as in the second set of models described above and those including cropping system identifier).

The last set of regression models ("regulation effect models", models 11–15 in Supplementary Table 1) aimed to investigate the effect of crop diversity indicators and cover crop frequency on pesticide use (total, herbicide, fungicide, and insecticide) and whether the effects of crop diversification were dependent on the crop considered. Crops with very low total and herbicide use ($N_{removed} = 4$, out of the 16 originally considered), fungicide use ($N_{removed} = 7$), and insecticide use ($N_{removed} = 12$) were not considered in the corresponding models. For all four response variables, a baseline model which included climatic region, crop, crop diversity indicators, and cover crop frequency as fixed effects, was compared based on Akaike Information Criteria (AIC) to four more complex models which included higher-order

interactions between crop and crop diversity indicators, while respecting marginality constraints. Only the most parsimonious of the five candidate models was selected for analyses. For herbicide use, alternative models focusing on diversity of sowing periods, rather than crop functional diversity, were subject to the same model selection procedure but the interaction between crop taxonomic diversity and diversity of sowing periods was not considered meaningful, resulting in only 4 candidate models. Random effects were identical to those included in the previous set of analyses ("dilution effect models"). A list of all models retained for analysis can be found in Supplementary Table 1.

Model assumptions were assessed with the R package DHARMa[64], which uses a simulation approach. Accounting for potential temporal autocorrelation was not achievable, as many cropping systems were described only once over time (Supplementary Fig. 13). Graphs of fitted vs. observed values (and their squared Pearson correlation coefficient) are provided in Supplementary Fig. 16 as a useful diagnostic tool for the "regulation effect models".

The significance of effects was assessed with type III Wald chi-square tests (with sum-to-zero contrasts for factors), using the function Anova from the R package car[65]. All contrasts were set up with the R package emmeans[66]. $P$-values arising from pairwise comparisons between factor levels (i.e., climatic region, crops) were adjusted using the false discovery rate method. Slopes of diversity indicators were tested against zero for the different crops using a Wald test. The effects presented are marginal in the presence of non-focal factors, meaning effects are "marginalized" (or "averaged") over the levels of the non-focal factor (i.e., climatic region in models 4–15). Wald 95% confidence intervals were computed based on the delta method and adjusted for simultaneous inference based on the Bonferroni method.

### Correlation analyses

Correlation analyses between diversity indicators, or between crop taxonomic diversity and effective number of cereal crop species (computed according to Eq. 1), are based on Pearson's product moment correlation (function cor.test in base R).

### Reporting summary

Further information on research design is available in the Nature Portfolio Reporting Summary linked to this article.

## Data availability

The datasets that support the findings of this study were deposited in the Data INRAE repository and are available at the following link: https://doi.org/10.57745/NHWIQN The E-phy database used in this study was provided by the French Ministry of Agriculture and is available online[55].

## Code availability

The R script used to analyze the data and generate the figures was deposited in the Data INRAE repository and is available at the following link: https://doi.org/10.57745/NHWIQN.

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

## Acknowledgements

We thank the farmers and farm advisors from the DEPHY network who shared and collected the data, respectively, as well as the AGROSYST team who extracted the data. The work was part of the 3rd Programme for Future Investments (France 2030). The work was operated by the SPECIFICS (ANR-20-PCPA-0008, grant recipient: S.C.) and MoBiDiv (ANR-20-PCPA-0006, grant recipient: N.M.J.) projects funded by the "Growing and Protecting crops Differently" French Priority Research Program (PPR-CPA), as part of the national investment plan operated by the French National Research Agency (ANR). We would like to thank macrovector/Freepik (oilseed rape, barley, wheat), Freepik (insect, weed), and macrovector-official/Freepik (fungal pathogen) for designing the images used in Fig. 1.

## Author contributions

M.G., G.A., N.M.-J. and S.C. designed the study; M.G., G.A. and E.C. processed the data; M.G. and G.A. analyzed the data; M.G., G.A., N.M.-J., S.C., E.C., R.N. and Y.Z. were involved in the interpretation of the results; M.G. and G.A. wrote the manuscript with inputs from N.M.-J., S.C., E.C., R.N. and Y.Z.

## Competing interests

The authors declare no competing interests.
