## [Peer Review File · Nature Communications]

Fostering temporal crop diversification to reduce pesticide useREVIEWER COMMENTS

Reviewer #1 (Remarks to the Author):

Overall comments:

This paper presents an analysis of the effect of crop system diversification on pesticide use, as measured by the TFI. The authors use data from a network of agricultural plots in France (DEPHY).

The authors present two analyses: i) the effect of the number of crops on the TFI, ii) the effect of the return delay on the TFI. The authors choose two modelling strategies: a 95% quantile modelling for the first variable, without model sensitivity analysis, and an average effect analysis for the second variable with a greater effort to better define the structure of the optimal model. Overall, the data analysis part could be refined/improved. For example, crop diversification is characterised by a simple indicator, not taking into account the different functional types of crops or cropping intensity (e.g. related to cover crops). It is possible, I think, like other large-scale studies, to use a more accurate indicator of crop diversification that takes into account these important characteristics. Second, what is the complementarity between the two ways of analysing the data (number of crops vs. return delay), is a global analysis possible? Why did you choose to model the 95% quantile in one case, and the average effect in the other? I am not convinced of the value of focusing only on the 95% quantile; why not model the others quantile too? Why did you test several model structures in one case, but not in the other? Why did you consider years as fixed effect, but do not interpret or use these parameters? Uncertainty analyses are absent, and the authors do not present confidence intervals (except in the appendices, but not interpreted), which makes a critical analysis of the results difficult. The text is sometimes confusing, especially when the authors mention about slope for a polynomial model; or for certain other circumstances. The graphs could be improved, in particular to allow readers to better understand the variability of the data: currently the density of points on the graphs is difficult to analyse (too many points, perhaps think of making boxplots, other?); too many curves appear on Fig 2 ; ...)

There are some inaccuracies or lack of clarity in the text. For example in the abstract: "For three climatic regions out of six, increasing the number of crops in the rotation from one to six slightly increases pesticide use but decreased afterward". What about the other 3 regions?; are the results significant?; what is the threshold at which the TFI starts to decrease?;..... In the results, the description of the database should be improved, you sometimes provide not optimal details and sometimes important element are missing to properly characterise the data you are working on. The detail of the results by region makes it more difficult to have a global view of the results and therefore a clear message to retain. The link between the two analyses should be better formatted/integrated. The discussion is sometimes general, and some interesting elements could be integrated into the analysis (see my comments on the model+other elements: I think you have the database to produce a more precise analysis on your own).

In conclusion, this article presents some interesting analyses, but would deserve to be improved especially on the data analysis part for the coherence of the analyses and to produce more precise results, that would allow in fine to improve the text, its precision and the key messages to be drawn from this study.

Detailed comments :

Abstract

-“We considered 1478 conventional cropping systems, which displayed a wide diversity of management strategies within and across various pedoclimatic conditions”

-> Specify the method further. Are these data from experimental network? Meta-analysis? Others?

-“or three climatic regions out of six, increasing the number of crops in the rotation from one to six slightly increases pesticide use but decreased afterward.”

- \ Decrease after what?
- \ And for the three others?

Introduction

-“Finally, temporal diversification leads to more diversified landscapes with a mosaic of crop fields that contributes to resource dilution for pests and limit their transfer from one field to another”

\ Add a reference ? see maybe Estrada, et al, 2022. PNAS
(<https://doi.org/10.1073/pnas.2203385119>) ?

- “the reduction of pesticide use allowed by temporal crop rotation diversification has never been quantified in a diversity of production situations”

-> The references cited are meta-analyses, so include a variety of production situations. Specify the argument.

-“cropping system diversification (i.e. crop rotation and management practices)”

\ The diversification of cropping systems is not limited to crop rotation.

Should you use a more complex indicator of crop diversification? E.g.

$Cropdiversity = N_{species} \times N_{group} \times N_{year}$

where $N_{species}$ is the total number of crop species, N_{group} is the total number of crop functional groups, and N_{year} is the average number of crop species per year.

Results:

- Fig1. :

\ Why focus only on the 95% quantile?

\ Better represent the diversity with boxplots? There are so many points in this figure that it becomes difficult to visualise the variability of the raw data correctly.

\ Are all crops represented on this graph? Does the model take the crops into account? If not, I think this is problematic.

-“The number of crops in the rotation ranged from 1 to 15”

\ also give the mean/median value and an indicator of dispersion?

-“the slope of the upper limit of the relationship”

\ If it is a second order polynomial regression, what parameter do you call ‘slope’.

-« For these three regions »

-> Is region the right term?

-Fig2.

-> Difficult to read. Many points/lines overlap.

-> Do you have the frequency of occurrence of the number of crops in the systems. I guess there is much less data for 10 crop rotations rather than 3 or 4.

Fig3. Here, vous focalisez sur le mean effect, plus sur le quantile 95%-> pourquoi ?

-“In most cases, increasing the return delay of crops reduced cropping system TFI (Fig.4; Supplementary Table 4).”

\ Is the decrease in the TFI significant? You do not display the CIs.

-for wheat it reduced cropping system TFI by 1.0 TFI point (-22%).

\ Please provide CIs.

-Increasing the return delay of barley from two to five

\ Is barley different from wheat -> Is it possible to group all winter cereals?

-Are the analysis on crop return a confirmation of the analyses on crop diversity? Is it possible to make only one global analysis?

Discussion

-Analyses highlighted that temporal crop diversification (high number of crops in the rotation, high return delay of each crop) could allow a substantial reduction in pesticide use at the crop and cropping system level.

\ Not agree. See Fig 1. The TFI increase until crop complexity= 5/6 crops.

-For most crops, increasing their return delay decreased pesticide use at the crop and cropping system level

\ Please precise if the decrease is significant. For all crop return delays?

-Considerable variability in pesticide use was left unexplained by temporal crop diversification

\ Did you provided a R2 or percent of variance explained to judge critically your models?

Material and methods

-“to model the upper limit ($\tau = 0.95$)” -> Why focus only on this quantile. Solutions exist to model several quantiles.

-“Indeed, preliminary analysis with generalized linear mixed-effect models highlighted a clear violation of homoscedasticity of residuals”

-> "What is the purpose of this sentence? To justify the « bell » curve? To justify taking into account repeted observations? in both cases, I am not convinced.

-> How do you deal with these problems of heteroscedasticity ? variable transformation ? explicit consideration of heteroscedasticity in the model ? other ?

- « Vertical distance » -> ???

-The upper limit hence characterizes the direct effect of the number of crops in the rotation on cropping system TFI while the vertical distance separating data points to the upper boundary represents the effect of non-considered variables (e.g., differences in crop choice for a given level of crop diversity)”

\ Not convinced. The upper limit is I think crop and context dependent.

\ What do you call direct effect?

-Convergence issues were encountered when considering the interaction between region and number of crops. Hence, separate models were fitted for each region

->What were the convergence problems? If you don't consider the regions as random, then you have one estimate per region (or two/n in the case of polynomial models), so I don't understand why there would be convergence problems.

-(thereby not allowing the comparison between slopes of different regions)

I don't understand why it is not possible to compare the parameters of the models....

-“A cropping system identifier was included as a random intercept effect”

-> Do all farmers have only one cropping system?

-> Why didn't you take into account this effect on the slope?

-“for each climatic region, cropping system TFI was modelled as a function of period of description and number of crops ”

-> What do you call period description?

->How is the effect of disease pressure taken into account? Do you take a year effect in your model?

-> Do you not take a crop effect in your model?

Supplementary Table 1

Why not put the years as random effect?
Why is there a line with "period 2014,2015,2016"?
Why display an AIC, if you are not comparing models?

Reviewer #2 (Remarks to the Author):

In this work the authors analyzed the potential reduction in pesticide use when increasing the number of crops within a rotation making use of a large database of French arable crop farms. As agronomist I consider this work very valuable and timely, since it provides reliable data to support one of the classical assumptions in crop production: i.e. less pressure of biotic agents in well-designed crop rotations.

In my opinion this work should account for productive trade-offs or externalities. In any case a TFI or pesticide use reduction is successful if the primary objective of agricultural production is not fulfilled. If not, we are at risk of recommending diversification solely to reduce pesticide use while depending on imports to fulfill the yield/caloric/protein demand. A standardized TFI accounting for this/these primary objective/s would be very interesting to account for the potential trade-offs of diversification. If there are not enough data for a "standardized" or "yield-based TFI" estimation, a paragraph should be devoted on this.

In my opinion, one of the limitations of this work is the framework on "pesticides", without focusing on control strategies to specific biotic agents: e.g. weeds, insects, fungi or bacteria, etc., or even gastropods (which can be important in some specific crops such as rapeseed). Although I understand that the objective of this work was more general, the main findings and take-home messages could be a bit simplistic. Could the authors discuss on this?

I suggest as well providing values of TFI reduction for values of return delay lower than 5 years. Apparently, the model (Figure 3) seems to indicate a strong reduction in TFI as fast as when breaking the monoculture and increasing the return delay to 3 years. The improvement from 3 to 5 years seems to be marginal in many cases.

L242-245. A small proportion of cases was monitored for more than 5 years. Could the authors explain which impact could have that limitation on the results? Were these cases equilibrated between climates studied?

L248-250. That statement makes me posing the next question: is diversification a driver of TFI reduction per se or is the consequence of introducing less productive (in economic terms) crops in the rotations? Sunflower and peas are examples of field crops that usually led to lower short-term (i.e. at the crop scale) revenues to farmers when compared to cereals.

Minor comments:

Abstract. This work is mainly focused on temporal diversification (i.e. crop rotation of sole crops) with the unique exception of "méteil" and on annual crops. Given the increasing literature on spatial diversification (e.g. intercropping) and on perennality it would be informative to mention the focus on temporal diversification of annual crops.

L9. What do authors mean by "afterward"? Could they provide a temporal framework to this statement?

L18. "allowed by intensive pesticide use". I suggest changing by "allowed by synthetic pesticide availability". Not all simple cropping systems (understood as monocultures or rotations with very low number of crops) use pesticides intensively, that is possibly more related to the level of productivity and economic margin of benefit.

L34. I suggest focusing the text on references that actually provide data on the statements. For instance, McDaniel et al. (2014) do not provide data on the diversification of soil microorganisms when diversifying crop rotations.

L132. What do authors refer to "low or medium crop diversity"? Can they provide values to this statement?

L137. Authors state that "...the proportion of barley and oilseed rape with high pesticide requirements...". Barley is a winter cereal that is very well adapted to many pedoclimatic scenarios

and related to a low usage of pesticides in different climates than the ones covered in this work (e.g. Mediterranean). I wonder if the final use of each grain is also a key element to take into account when assessing pesticide usage? For instance, barley for malt producers are more intensive in their practices than when barley is used for feed. Could the authors discuss a bit on this?

L153-154. Similar to my previous comment, management practices are not the sole cause of pesticide use. The final use of each grain produced is probably a more important driver of pesticide use, with higher intensities when grain is oriented towards human consumption (e.g. aiming at a low mycotoxin concentration) or seed multiplication compared to grain for feed or forage.

L234. Wormer. Please, check.

L235. Joly et al. (2010) citation. Please, check.

Reviewed by Daniel Plaza-Bonilla

Reviewer #3 (Remarks to the Author):

Please see attached reviewer comments for a formatted version of the following

In the present study the authors evaluate whether temporal crop diversity (i.e., crop rotation) explains variation in pesticide use over a large spatial scale using a farmer site network of over 1000 locations. They find reduced pesticide use with increased degree of crop rotation for some crops and with regional variability. The manuscript needs substantial work, but could be a valuable contribution.

I have a few major concerns, points that need clarification, and suggestions to improve the manuscript:

1. The need for this study and the research gap it fills could be better motivated. As is, its novelty is not immediately clear.
2. The methods and experimental design are frustratingly vague. Specifically, the grain and extent of temporal coverage and the variability in what constitutes a 'cropping system' are not sufficiently detailed.
3. Relatedly, the authors use of two indices to characterize crop rotation in one instance and misleading in another. Return delay is not actually measured here
4. The assumptions of statistical models used are not thoroughly tested. For example, I have concerns that temporal autocorrelation is still an issue for these data.
5. The discussion could better present caveats and offer more inference on the crop and regions specific responses observed.

The authors may be dismayed to see that my comments are numerous and critical. However, these are in line with the potential impact of the findings and the caliber of the journal. I commend the authors for leveraging an impressive dataset at a timely and important question in agro-ecological management.

I believe that this study could be of interest to the readers of Nature Communications if thoroughly improved through a major revision.

Please see my specific comments are below. Comments of most concern are in bold.

Title: Please specify that this is temporal crop diversification (or more simply crop rotation). The evidence on the effects of 'crop diversification' is already spatiotemporally scattered, you could be refreshingly clear.

ABSTRACT

L4: These effects have been quantified, so this gap statement is not true. See:

Larsen A E and Noack F 2020 Impact of local and landscape complexity on the stability of field-level pest control Nat. Sustain. 4 120–28

Nicholson, C. C., & Williams, N. M. (2021). Cropland heterogeneity drives frequency and intensity of pesticide use. Environmental Research Letters, 16(7), 074008.

L6: 'cropping system' is unclear terminology. I get into this more in my review of the Methods

L9: return delay is jargon and somewhat meaningless if the time period over which it is calculated is unknown. I get into this more in my review of the Methods

L15: 'crop rotation' is missing from keywords. This is a term that is used widely by both academics and practitioners, so it should be included.

INTRODUCTION

L18: Oversimplification is a vague term. I think I know what you mean, but I suggest using something else. Also, this sentence implies that so-called 'oversimplification' is an effect of pesticide use but couldn't increased pesticide use be an effect of 'oversimplification'. In other words, the directionality of cause and effect here is more complicated than presented. I suggest revising this opening sentence completely.

L26: Omit the parenthetical statement. It adds jargon.

L31: is a meadow really a crop? And what is a 'hardy crop'? These parenthetical examples don't clarify much. Remove or give more concrete examples

L36: please add a reference. There are many to choose from, but Root 1973 is an obvious starting point

L38: ref 15 had been published, so should not cite preprint.

L40: At this point in the manuscript you have used the following terms: cropping system diversification, crop rotation, temporal crop diversification, temporal diversification, diversified crop rotations, low return delay. Please pick a term and use it consistently

L42: Why is the generalizability of these findings questionable? I would think that evidence from several long-term studies in different geographic regions ("scattered around the globe") is generalizable. Unless the results point in different directions. But this is not stated. Moreover, how does a single region study, like the one here, address this issue of generalizability. This critique needs to be clarified or revised and made more in line with the manuscript.

L45-47: "no straightforward quantitative relationship between crop diversification and pesticide use was established" But isn't this exactly what Nicholson & Williams and Larsen & Noack found? Albeit, in the former spatial crop diversification. Please make this research gap more precise. Perhaps I don't understand the research gap you are identifying. This could be the result from the typo in L46, where there is an omitted word or phrase.

L53: 'cropping system' is unclear. What kind of spatial unit is this? please use a clearer term (e.g., farm parcel) or define.

L53: A brief description of the DEPHY data would add robustness, and help the general reader from having to sleuth for what it means in the methods

L54: This first indicator is not adequately defined. When talking about crop diversity issues of scale, so familiar to ecologists, are also paramount. To understand this first measure you need to define the grain (i.e., days, weeks, months) and extent (i.e, 1 year, 8 years, 3 decades) at which 'the number of crops present' is being tallied. After reading through the whole manuscript, I am still unsure of both the grain (I assume years) and extent (8 year periods??). I think you could talk more carefully about temporal crop diversification by making these explicit.

L55: This second indicator is fundamentally flawed, or at least what you are calculating is not accurately measuring return delay. I expand on this below.

RESULTS

L59: I understand that this index is established and used previously by EU agencies and this group. However, a brief description of what it measures would be useful. This should include stating its directionality (i.e., is a value of 15 indicative of high pesticide use). It should also include a caveat that this metric does not capture ecological/health effects bc it does not account for toxicity.

L60-66: I have a hard time following the logic of this. You identify that factors other than crop rotation could explain variance in TFI (L60-63). Then you present the fact that there is a non-linear relationship (so call 'bell-shaped relationship'), in and of itself a finding. To you, these seem to motivate the use of quantile regression. However, to my knowledge this method is typically used when data are heteroskedastic. Please motivate the use of quantile regression more clearly (i.e., if it to test the relationship between crop rotation and TFI at some limit, then state that clearly and state why this is of interest).

L64: Those unfamiliar with quantile regression will not understand what this tau value means. Suggest saying '95th quantile'. Again, justify why you are picking this limit, either briefly here or in the methods

L65: I don't follow the inference made in the parenthetical statement. Please clarify

L66-67: 'slope of the upper-limit of the relationship between the number of crops in the rotation

and cropping system TFI'. This is a very derived measure and overly technical. Please provide a plain language term or description of what this slope measure captures. This will help the general reader.

L69: based on the abstract I know that the pesticide use increased, but again you use of overly technical language obscures this. I find 'upper limit of the relationship' to be cumbersome.

L71: is a TFI value of 6 high? Again, please provide the directionality of this measure.

L72: 'Oceaning' spelling

L79: From what one has read in the manuscript so far it is uncertain how many crops can be in rotation. This stems from the lack of clarity around two important aspects of your design: 1) what a cropping system means and 2) the time period over which crop diversity is being tallied. Lack of clarity on the first item means we don't know if a cropping system can have more than one field and therefore more than one crop in a given year. Lack of clarity on the second item means that we don't know if you are comparing 'cropping systems' over the same amount of time (e.g., 8 years) or if you are comparing systems with 4, 8, 12, etc. years' worth of data.

L80: proportion in space or time? Please clarify what this is a proportion of. Moreover, please confirm that these proportions have a common denominator (e.g., 4 years of maize out of 12 total years). Otherwise, is it safe to compare these? See previous comment

L77-88: These results are overly descriptive. You make multiple statements about increase or decrease with number of crops in rotation, but do not support them with statistical tests.

Moreover, what is the point of this result? You seem to be grasping at changes in temporal evenness (i.e., shift away from monoculture) with increasing temporal richness. The relationship between richness and evenness is deeply explored in community ecology, and there are more quantitative ways to present these data. For example, diversity profiles (see refs below) or species dominance metrics. Also, your figure confuses more than it explains. This is in part because in some cases you are aggregating multiple crops into larger groups (e.g. fodder) and in other cases a crop is singular (e.g., maize, barley). Please revise this section by reporting a more quantitative analysis of the temporal evenness. If this is not the aim of this section, I still suggest revising the presentation of data, especially Figure 2

<https://besjournals.onlinelibrary.wiley.com/doi/pdf/10.1111/2041-210X.12349>

<https://ericmarcon.github.io/entropart/reference/DivProfile.html>

L92: Why only eight crops? What are they? What quantitative criteria were used for their selection?

L93: exponential? In the discussion you mention log-linear. Just be consistent. Also, you did not 'establish' this relationship. You 'found' it, or your analyses 'showed' it

L94: Was there a significant interaction between crop rotation and climatic region for other crops. Please state non-sig results too.

L99: what about the other crops? You only report 5/8. Seems like you are only cherry-picking significant findings.

L101: You include the fits of multiple regions in each panel of your figure. This leads one to believe that we are meant to compare relationships between regions, but did you test for an interaction between rotation and region on crop TFI? If so please state the outcome of this test. If not, why not?

L105: Again, these results are rendered almost useless by not understanding what 'cropping system' means.

L107: Is this actually an increase in years between crop return? Also, I thought 'return delay' was estimated as an inverse proportion, so how do you have whole integer values? This is unclear and potentially misleading

L107-118: Nice. You do a much more balanced job reporting results (both sig and non-sig) and explaining where you found interactions or not. Please do this above

DISCUSSION

L123: Drop first 3 words. Not needed

L126: or exponential? You use a log link, so just pick the correct term (I am not sure what this would be given a Tweedie distribution).

L131-134: This is a bold and important statement but requires a little more careful presentation. First, you need to be explicit that these results are based on use patterns specific to the crops in the target region. For example, in other parts of the world, wheat does not receive a lot of pesticide application. More broadly and going back to your research gap, does this study contribute to generalizability of evidence? Especially given that you see such different responses between regions. Second, this statement is only true given the crops included in this analysis. Rotation with

other crops, not considered here, may not show this type of hump shaped relationship. Third, this statement assumes that all crops in rotation are conventionally managed. These caveats (and probably others) need to be raised.

L140: where is the closing parenthesis?

L141: What is a diversification crop? And does it always mean low pesticide use? The example of legumes is misleading bc there are plenty of legume crops with high pesticide use (e.g., soy). Meslin is not widely known, so is it really a good example? Also, is meslin a 'crop mixture'?

L141: Completely unclear what you mean by regulating effect. Please clarify

L143: stable after a point, right? Also, is this saturating effect your highlighting real, or a statistical artifact based on the link function specified in your models. To make this point you may have to explicitly test non-linear functions. Also, what is meant by 'production situation'? Please consistently use clear terms.

L145-147: Yes. Sounds like what this study could have done to be more explicit about 'return delay'

L149-152: This section is unsatisfying because a major theme of the results and figures are differences between regions. Please provide more inference about the strong regional difference you report. What biogeographical or sociocultural or macro-agronomic factors underpin them?

L153: Are these only climatic regions? I thought they were also soil related?

L154-158: This all comes off as very vague and hand-waving. Use of ellipses (...) does not help your case. Moreover, the point is that these practices could vary between regions no? Make that point and support it with evidence.

L161: Careful. Use terms consistently.

L162: And what exactly is the cropping system level. If cropping systems are field than this is a patch, but if their farms with multiple parcels than this is more of a landscape phenomenon. Is this statement going to depend on scale? Suggest to revise this sentence to be more clear and draw recommendations directly from the present analysis.

L162-171: This is good. Much clearer than L139-142. However, most of these sentences could use references to support their points

L174: crop [field] vs. cropping system. Yes agree. And exactly to my point above and elsewhere.

L179: unclear what is meant by 'similar mechanisms'

L180: what is 'protein autonomy'?

L181: The word choice here is off. 'Experimenting [with]...' perhaps? Moreover, what are these types of innovative systems? Organic agriculture? Or something else?

L185-205: This section kind of rambles and seems to be a grab bag of the benefits of crop diversification (beyond crop rotation). Please revise to draw more clear conclusions from the present work and how it addresses the research gaps you identified and how this work could, despite its limitations, motivate changes in crop selection at larger spatial and temporal scales.

MATERIALS AND METHODS

L210-212: This definition does not help. In fact, it raises more questions, and potentially sources of unaccounted for variation and sampling bias. A few questions:

1. What is a plot? A whole field? Or area within a single field? Is the size of plots variable between and within cropping systems?
2. How many plots are in 'a set'? Is the number of plots considered variable between cropping systems?
3. How was it determined these plots were 'managed similarly'? And by whom? Are all the same crops grown across plots?
4. 'in the case of several', several of what?
5. How do you get the dose of application? From these descriptions?

Lack of clarity on the above leads me to believe that there are unaccounted for sampling biases in these data. The authors' indicators of temporal crop diversification are essentially diversity metrics, which are going to be influenced by these kinds of sampling effects. Is it safe to compare the number of species between 1m² and 10km²? Most ecologists would suggest one needs to account for differences in sample coverage. Yet it is unclear if the present work is comparing sampling units (i.e., cropping systems) that vary greatly in their spatial extent and number of sampling units (points 1 and 2 above, respectively). This is important because these differences would influence the number of crops present within and between years. If cropping systems are variable in their number, size, and management then this should be included in models as fixed effects.

In general, the description of these data is very vague. Please provide more explicit details about the way the data are collected, including detailed information about the spatial and temporal scales of collection. These descriptions should be supported with supplemental analysis that show the variance in cropping system in line with the questions above. Otherwise, I find it very hard to evaluate the results presented here due to the unknown influence of potential sampling bias.

L212-214: It is unclear how simply providing this information 'reduces the importance of crop x year effects'. Also, it is unclear what these effects might be and why it is important to reduce them in light of the analyses presented here.

215-216: It is still unclear what information farmers are providing, and how it is aggregated.

L220: word choice. 'resorted to...' perhaps instead 'used'

L221: Figure S1 does not clarify this point. In fact, it raises further questions. It depicts the number of crops, over what time period? What if a cropping system had more than one crop in a given year. Are these included in this summation?

L222: 'over a 3-12 year period' what is the distribution of cropping systems for each of these durations? Please provide a supplemental figure (e.g., bar plot) depicting the number of farms and the duration of their 'period of description'. Variability in this period raises concerns about your measure of 'return delay'. I raise this issue in more detail below.

L222-223: 'i.e. 3-year entry description and follow-up annual description' I do not understand this.

L224: '194 to 3894' it is unclear why there is such a range in the data available for this indicator. Please clarify

L225-226: Did you test for and detect temporal auto-correlation? Then did you check to see if the random effect actually accounted for the TAC? Please provide the outcome of these tests. I would imagine that auto-correlation would be very high for these data and you will need more complex model structures (e.g., AR1) to control for TAC.

L228-236: Throughout the text you describe pedoclimatic regions. However, based on these methods it these regional distinctions appear to only be climate based. Are soil differences considered? Please clarify

L235-236: Reference needs to be fixed. 'to appreciate' not best choice of words

L242-245: By the authors own admission (i.e., 'unfortunately') the index described is not a measure of return delay, which is clearly defined (L55) as, 'number of years separating sowing of the same crop'. It is misleading to describe this measure as return delay, it is a measure of temporal evenness, and an imperfect one at that.

Consider the two 8 year cropping sequences:

1. maize, maize, barley, rye, pea, barley, maize, maize
2. maize, barley, maize, barley, maize, barley, maize, barley

The method proposed would assign a maize 'return delay' value of 0.5 for both, when their actual return delay is quite different (4 vs. 1 year for maize, respectively).

Return delay is a clearly defined and important agriculture practice. I cannot recommend this manuscript to be published if it uses the current index but reports results for effect of 'return delay'. I suggest to either correctly measure return delay (but this seems like its limited by data availability) or to use a different term. However, even as a measure of temporal evenness I have concerns about this approach given lack of clarity on sampling coverage detailed above. This measure is going to be influenced by the number of years in consideration. This needs to be accounted for. Hill numbers could provide a useful metric. I suggest two articles for your consideration:

<https://esj-journals.onlinelibrary.wiley.com/doi/10.1111/1440-1703.12102>

<https://onlinelibrary.wiley.com/doi/10.1111/oik.07202>

L245-246: this assignment makes no sense, see example above.

L253: this definition makes no mention of the dose used. I prefer the definition used in the authors' previous work: 'the mean number of treatments per hectare with commercial products, weighted by the ratio of the dose used to the recommended dose' (Pelosi et al. 2013). Also, shouldn't this definition mention that this application is being used for a given crop, in a given year. Also, please specify that this is pesticide product, rather than pesticide active ingredient.

L256: check the indexing in this equation. I am not sure, but I think you may want a j iterator for the inner summation (e.g., D_{ij} , S_{ij}) because these are specific to the jth crop

L257: Again, how do you know the applied dose? Is it self-reported?

L258: 'St is the total plot area', What does this mean in the context of this study, see major comment above. This is not the total area or the cropping system, correct?

L259-260: what do you mean by expressed? Also, what units are the applied and reference dose

in? How did you harmonize and handle data when units differed (e.g., g/ha vs. L/ha)?

L260: Please provide caveat here that TFI does not measure the degree of environmental disturbance of pesticide use because it does not account for toxicity.

L266: is this 'list' published? Please reference or provide a list of products that were excluded.

L272: Neither of these papers use the term 'envelope curve', suggest more relevant literature. As per comment above, here is where you could motivate the use of quantile regression for at analyses at a given limit.

L276: ah so it is a practical use case.

L277-279: I have a hard time following this

L281: fit, not 'fitted'

L282: Do you adjust p values to account for multiple comparisons resulting from these separate region models? Please consider doing so

L286: And did inclusion of this term eliminate temporal autocorrelation? Please check the autocorrelation functions of you models (e.g. stats::acf(mod)). If autocorrelation exists, please account for this further.

L289: 'eight previously presented crop species' Again, it is so unclear what the temporal extent of your indices are. Here, you are stating that it's the 8 previous crops, so are you only using those cropping systems with ≥ 8 years of data reported. Please clarify.

L292: fit

L294: Why do you have a point mass at zero? Are there a lot of cropping systems that do not apply pesticides. It would useful to see summary plots of the range applied doses

L296: Wait, what is 'period of description'? A duration of time or a single year? If the former, what is the rationale for including it. If it to try an account for differences in sampling coverage, then I don't think this is doing what you want it too. Look into including duration as a fixed effect or offset term. If the latter, then simply say this and be consistent elsewhere.

FIGURES AND TABLES

Figure 1: Please make the point size scaled to the number of years under consideration. The first sentence of the figure caption is supposed to clearly say what it shows, use of technical terms (see above) makes this hard to understand. Why are there no confidence intervals? Please add CI
L434: 'and/or' which is it? As in, does a 2 mean the model had a significant linear and quadratic term, or only a significant quadratic?

Figure 2: This figure is confusing because it is unclear how many years are considered to calculate crop proportion (is it standardized?) and because of the aggregation to crop groups for some species. Please revise this figure

Figure 3: Please provide confidence intervals. A point for the discussion, but you never explain what the mechanism is for the non-linear trend. Why does it flatten? And why would slopes/intercepts differ between regions. See Discussion comments above.

L445: 'over a given period' suggest to make point size scaled to the length of this period.

L446: Nature Publishing Group has strict guidelines for what needs to be included in the captions. And your reporting summary says you have followed these. But doubles check to make sure that you are reporting the type and terms of the model used. What are the random effects?

L447: First time using the term population. What is a population here? Somewhat confusing

Figure 4: please provide CI

L461: Suggest to say 'after finding a non-significant interaction'

Figure S1: The information provided in the caption of this figure is not sufficient to understand what it is depicting.

Figure S2: Again, NPG has guidelines on what must be included in figure captions. This caption does not follow those guidelines (e.g. what do the boxes and lines depict?)

Answers to reviewers

Reviewer 1

Reviewer's comments	Authors' answers
GENERAL COMMENTS	
This paper presents an analysis of the effect of crop system diversification on pesticide use, as measured by the TFI. The authors use data from a network of agricultural plots in France (DEPHY). The authors present two analyses: i) the effect of the number of crops on the TFI, ii) the effect of the return delay on the TFI. The authors choose two modelling strategies: a 95% quantile modelling for the first variable, without model sensitivity analysis, and an average effect analysis for the second variable with a greater effort to better define the structure of the optimal model.	No comment. See further for specifics.
Overall, the data analysis part could be refined/improved. For example, crop diversification is characterised by a simple indicator, not taking into account the different functional types of crops or cropping intensity (e.g. related to cover crops). It is possible, I think, like other large-scale studies, to use a more accurate indicator of crop diversification that takes into account these important characteristics.	We agree. Crop rotation diversification is now appreciated through different indicators :  - Effective number of botanical families - Average number of crop species per botanical family - Effective number of sowing periods - Cover crop frequency
Second, what is the complementarity between the two ways of analysing the data (number of crops vs. return delay), is a global analysis possible? Why did you choose to model the 95% quantile in one case, and the average effect in the other? I am not convinced of the value of focusing only on the 95% quantile; why not model the others quantile too? Why did you test several model structures in one case, but not in the other? Why did you consider years as fixed effect, but do not interpret or use these parameters? Uncertainty analyses are absent, and the authors	Originally, we wanted to investigate whether or not more diversified cropping systems (initially appreciated through the number of crops in the rotation) limited pesticide use at the cropping system scale and whether increasing the return delay of a given crop limited pesticide use in that given crop, thus justifying the two scales of study. However, as noted by several reviewers, this notion of return delay cannot be properly investigated with this dataset. We hence now focus on crop rotation diversity, rather than crop return delay. We now focus on one core analysis at the crop scale. Indeed, the original analysis at the cropping system scale did not allow to disentangle crop dilution effects (i.e. introducing a meadow in a cropping system necessarily

do not present confidence intervals (except in the appendices, but not interpreted), which makes a critical analysis of the results difficult. The text is sometimes confusing, especially when the authors mention about slope for a polynomial model; or for certain other circumstances. The graphs could be improved, in particular to allow readers to better understand the variability of the data: currently the density of points on the graphs is difficult to analyse (too many points, perhaps think of making boxplots, other?); too many curves appear on Fig 2 ; ...)	reduces pesticide use) from regulation effects (i.e is wheat in a more diversified crop rotation less pesticide reliant ?), which is the specific interest of this manuscript. All 16 main crops (based on the average proportion across all crop rotation) are now combined in a single model and their effect accounted for. The effect of diversity indices is now tested while accounting for crop effects. This analysis now allows to answer the question : « are crops introduced in more diversified crop rotation less pesticide reliant », independently of crop type. Most reviewers did not find quantile regressions warranted, and we believe the new analysis to be more thorough, so original quantile regressions were entirely removed from the manuscript. Moreover, modelling requirements were on the limit of what is currently available for quantile regression (e.g. no crossed random effects, only one single random effect, which is why year was initially included as a fixed effect in the analysis and why there was no latitude for identifying optimal random effect structure in this analysis). Confidence intervals are now included all throughout the manuscript. Efforts were made to facilitate the reader's appreciation of the variability of the data.
There are some inaccuracies or lack of clarity in the text. For example in the abstract: "For three climatic regions out of six, increasing the number of crops in the rotation from one to six slightly increases pesticide use but decreased afterward". What about the other 3 regions?; are the results significant?; what is the threshold at which the TFI starts to decrease?;..... In the results, the description of the database should be improved, you sometimes provide not optimal details and sometimes important element are missing to properly characterise the data you are working on. The detail of the results by region makes it more difficult to have a global view of the results and therefore a clear message to retain.	This sentence is not present in the new version of the article. As a matter of fact, the choice was made to model interactions between climatic regions and diversity indicators as random, so as to focus on effects which are common across regions (in line with the next comment). Nevertheless, climatic region was considered as a fixed effect due to its low number of levels (=6), which can be considered as insufficient to compute a variance. Efforts were made to more thoroughly describe the data acquisition procedure and the corresponding database. As previously mentionned, we now focus on effects which are common across climatic regions. Interactions between diversity indicators and climatic regions are now considered as random (these effects are nevertheless shown in supplementary material, while main text essentially focuses on fixed effects). We now focus on one core analysis at the crop scale, all 16 main crops combined (see answer to previous comment). See further for specific answers concerning the discussion.

The link between the two analyses should be better formatted/integrated. The discussion is sometimes general, and some interesting elements could be integrated into the analysis (see my comments on the model+other elements: I think you have the database to produce a more precise analysis on your own).	
In conclusion, this article presents some interesting analyses, but would deserve to be improved especially on the data analysis part for the coherence of the analyses and to produce more precise results, that would allow in fine to improve the text, its precision and the key messages to be drawn from this study.	We agree and believe to have considerably improved the quality of data analysis.
ABSTRACT	
“We considered 1478 conventional cropping systems, which displayed a wide diversity of management strategies within and across various pedoclimatic conditions” -> Specify the method further. Are these data from experimental network? Meta-analysis? Others?	Efforts were made to more thoroughly describe the data acquisition procedure, the corresponding database. Data originates from the DEPHY Farmer network. This network was created to help volunteer farmers reduce pesticide use, albeit without impeding their financial profitability. Advisors, on top of accompanying farmers, monitor the evolution of each cropping system (i.e. set of plots within a farm manage with the same crop rotation) and keep track of the practices and performances.
-“or three climatic regions out of six, increasing the number of crops in the rotation from one to six slightly increases pesticide use but decreased afterward.” Decrease after what? And for the three others?	This type of sentence is not present in the manuscript anymore (for reasons detailed above).
INTRODUCTION	
“Finally, temporal diversification leads to more diversified landscapes with a mosaic of crop fields that contributes to resource dilution for pests and limit their transfer from one field to another” Add a reference ? see maybe Estrada, et al, 2022. PNAS (https://doi.org/10.1073/pnas.2203385119) ?	Thanks for the reference. It was added.

“the reduction of pesticide use allowed by temporal crop rotation diversification has never been quantified in a diversity of production situations” -> The references cited are meta-analyses, so include a variety of production situations. Specify the argument.	The meta-analyses only focused on pest control, but did not investigate how this may upscale to pesticide use in real farming contexts. This knowledge gap part of the introduction was improved.
-“cropping system diversification (i.e. crop rotation and management practices)” The diversification of cropping systems is not limited to crop rotation.	Good catch. We completely agree. We now make it clear that we focus on temporal crop diversification. Spatial crop diversification is partially appreciated through the inclusion of cereal-legume mixtures (one of the top 16 crops of the dataset), the sole type of crop mixtures present in the dataset.
Should you use a more complex indicator of crop diversification? E.g. $Cropdiversity = N_{species} \times N_{group} \times N_{year}$ ear where $N_{species}$ is the total number of crop species, N_{group} is the total number of crop functional groups, and N_{year} is the average number of crop species per year.	We agree that a more functional aspect should be investigated. Hence, crop rotation diversification is now appreciated through different indicators :  1. Effective number of botanical families 2. Average number of crop species per botanical family 3. Effective number of sowing periods 4. Cover crop frequency The first three indicators were calculated on the basis of the Hill index Overall diversity is investigated in the models through the interaction between 1. and 2. We believe this approach to be quite similar to what is proposed by this reviewer. Thank you for pointing down this line.
RESULTS	
- Fig1. : Why focus only on the 95% quantile? Better represent the diversity with boxplots? There are so many points in this figure that it becomes difficult to visualise the variability of the raw data correctly. Are all crops represented on this graph? Does the model take the crops into account? If not, I think this is problematic.	This figure is not present anymore. This analysis was replaced by a unique analysis combining all 16 main crops. Results from this analysis are now presented as a coefficient plot. In the main text, a graph with points highlights the relationship between crop rotation diversity and pesticide use for the most dominant crop (i.e. wheat) across the 6 climatic region. Others are included in supplementary material.
-“The number of crops in the rotation ranged from 1 to 15” also give the mean/median value and an indicator of dispersion?	An indicator of dispersion is now included for all the response and explanatory variables investigated.
-“the slope of the upper limit of the relationship” If it is a second order polynomial regression, what parameter do you call ‘slope’.	We agree this could have been clearer. The notion of slope is not straightforward for a quantile polynomial regression with non linear link. Nevertheless, this type of analysis was removed from the current version of the manuscript.

-« For these three regions » -> Is region the right term?	The comment doesn't appear anymore relevant in light of the new analyses and corresponding text.
-Fig2. -> Difficult to read. Many points/lines overlap. -> Do you have the frequency of occurrence of the number of crops in the systems. I guess there is much less data for 10 crop rotations rather than 3 or 4.	We agree that data density is substantial and that data visualization was not ideal. This figure was removed and is now replaced by a coefficient plot and a graph highlighting the effect of crop diversification on pesticide in wheat (the main crop). Other crops are included in supplementary material. Indeed, predictions based on fixed effects yield the same result for all crops considering interactions between crops (and climatic regions) and diversity indicators are included as random effects. Nevertheless, predictions including random effects are also presented in supplementary material. Distribution of explanatory variables is now more thoroughly described (graphically but also in the text). But yes, crop rotations including 10 crops are much less common in the dataset than crop rotations made up of 3 or 4 crops.
Fig3. Here, vous focalisez sur le mean effect, plus sur le quantile 95%-> pourquoi ?	The original justification for the quantile regression (at 95%) at the cropping system scale was that crop composition was expected to have a tremendous effect on the variability of pesticide use at a given level of crop diversification. We hence wanted to highlight a constraining effect of crop diversification on the upper boundary of the relationship (acknowledging that the variability below this upper-limit was highly driven by crop composition). This was not anymore an issue when investigating pesticide use at the crop scale, and why mean effects were investigated in this case. However, as previously mentioned, quantile regressions were replaced by more thorough analysis in the new version of the manuscript.
-“In most cases, increasing the return delay of crops reduced cropping system TFI (Fig.4; Supplementary Table 4).” Is the decrease in the TFI significant? You do not display the CIs. -for wheat it reduced cropping system TFI by 1.0 TFI point (-22%). Please provide CIs.	Confidence intervals are now systematically provided.
-Increasing the return delay of barley from two to five Is barley different from wheat -> Is it possible to group all winter cereals?	Barley and wheat were combined in the same functional group (e.g. Poaceae) when computing effective number of crop families. Winter barley and winter wheat were combined in the same sowing period (e.g. winter) when computing effective number of sowing periods. However, crops were kept separate (but in the same model) when analyzing the relationship between crop

	pesticide use and crop rotation diversification. Indeed, these seemingly similar crops do not necessarily share the exact same set of pathogens (e.g. barley yellow dwarf). Moreover, going down this line would have implied combining ryegrass, triticale, and oats (e.g. hardy crops associated to livestock) with much more intensive and pesticide dependent crops (winter wheat, winter barley...).
-Are the analysis on crop return a confirmation of the analyses on crop diversity? Is it possible to make only one global analysis?	Our original intent was to investigate the effect of crop diversification on pesticide use at the cropping system scale and to investigate whether, for a same level of crop diversification, increasing crop return delay could further reduce pesticide use in that given crop. However, crop return delay was originally computed based on crop proportion, which doesn't capture what was intended and which was negatively correlated to crop diversification. We hence now only focus on crop diversification (and not return delay) at the crop level (all crops combined).
DISCUSSION	
-Analyses highlighted that temporal crop diversification (high number of crops in the rotation, high return delay of each crop) could allow a substantial reduction in pesticide use at the crop and cropping system level. Not agree. See Fig 1. The TFI increase until crop complexity= 5/6 crops.	We agree that this previous analysis did not allow to disentangle crop identity effects (what we refer to crop dilution effects previously) from true regulation effects (i.e. a given crop is less pesticide dependent when introduced in a more diverse rotation). The tricky thing originally was that some cropping systems with low crop diversity (e.g. maize monocropping, maize – ryegrass, sunflower-wheat) were also associated with low pesticide use due to crop identity effects. The original analysis did not allow to investigate, for example, whether maize was less pesticide reliant when included in a more diversified cropping system. The analysis retained for the new version of the manuscript should overcome these problems.
-For most crops, increasing their return delay decreased pesticide use at the crop and cropping system level Please precise if the decrease is significant. For all crop return delays?	Uncertainty associated to each effect is now presented (through confidence intervals and tests of coefficients). This sentence was originally based on results from the relationship between crop return delay and pesticide use, which were removed in the current version of the manuscript.
-Considerable variability in pesticide use was left unexplained by temporal crop diversification Did you provided a R2 or percent of variance explained to judge critically your models?	This sentence was based on visual inspection of graphs. Unfortunately, calculating a global measure of model fit (such as R2) for GLMM is riddled with complications and no simple single answer can be found (different functions yield different results). Moreover, tweedie families are not supported by the commonly used packages (performance :r2(),MuMIn::r.squaredGLMM(),r2glmm package,

	piecewiseSEM package, <code>psycho::R2_nakagawa()</code>, the <code>partR2</code> package). Ben Bolker provides some simple/crude solutions which all have their own caveats, but which can be mobilized if the reviewer stands ground. Of these : Comparison of the residual variance of the full model against the residual variance of a (fixed) intercept-only null model: <code>1-var(residuals(m))/var(model.response(model.frame(m)))</code> OR the squared correlation between the response variable and the predicted values <code>cor(model.response(model.frame(m)),predict(m,type="response"))^2</code> https://bbolker.github.io/mixedmodels-misc/glmmFAQ.html
MATERIAL AND METHODS	
-“to model the upper limit ($\tau = 0.95$)” -> Why focus only on this quantile. Solutions exist to model several quantiles.	The original justification for the quantile regression (at 95%) at the cropping system scale was that crop composition was expected to have a tremendous effect on the variability of pesticide use at a given level of crop diversification. We hence wanted to highlight a constraining effect of crop diversification on the upper boundary of the relationship (acknowledging that the variability below this upper-limit was highly driven by crop composition). This modelling approach was not retained in the newly submitted version of the manuscript.
-“Indeed, preliminary analysis with generalized linear mixed-effect models highlighted a clear violation of homoscedasticity of residuals” -> "What is the purpose of this sentence? To justify the « bell » curve? To justify taking into account repeated observations? in both cases, I am not convinced. -> How do you deal with these problems of heteroscedasticity ? variable transformation ? explicit consideration of heteroscedasticity in the model ? other ?	Indeed, this was to justify the bell curve. We initially observed much more residual spread at low values of crop diversification with GLMM than at high values. Non-Independence between data points is accounted by a complex (and highly conservative) random effect structure in all models. Heteroscedasticity was not observed under the new modelling procedure (at least after accounting for 0 inflation). Variable transformation is difficultly justifiable within the GLMM framework as a link (log is our case) is already included. Dispersion models would have been the alternative in this case.
- « Vertical distance » -> ???	We agree, poor choice of words. The new modelling procedure does not call for such terms.
-The upper limit hence characterizes the direct effect of the number of crops in the rotation on cropping system TFI while the vertical distance separating data points to the upper boundary represents the effect of	We agree that this is somewhat speculative. Such explanations are not warranted by the new modelling procedure anymore.

non-considered variables (e.g., differences in crop choice for a given level of crop diversity)” Not convinced. The upper limit is I think crop and context dependent. What do you call direct effect?	
-Convergence issues were encountered when considering the interaction between region and number of crops. Hence, separate models were fitted for each region ->What were the convergence problems? If you don't consider the regions as random, then you have one estimate per region (or two/n in the case of polynomial models), so I don't understand why there would be convergence problems.	Diagnosing convergence issues is not always a simple task, especially in such complex models. This could be explained by variance inflation factors, identifiability, competition between fixed and random effects in presence of interactions... With the new modelling procedure, convergence issues were only identified in the case of the most complex models (varying slope of all three diversity indicators, and for the interaction between effective number of botanical families and average number of crops per botanical family, across combinations of crop and climatic regions, while estimating all possible correlations between slopes and intercepts).
-(thereby not allowing the comparison between slopes of different regions) I don't understand why it is not possible to compare the parameters of the models....	As you mentioned earlier, there is no easy interpretation of what the slope is
-(thereby not allowing the comparison between slopes of different regions) I don't understand why it is not possible to compare the parameters of the models....	Originally, the polynomial quantile regression model including climatic regions, number of crops , and their interaction did not converge. We therefore decided to model each climatic region separately, which did not allow to compare slopes. This is not a problem anymore with the new modelling approach.
-“A cropping system identifier was included as a random intercept effect” -> Do all farmers have only one cropping system? -> Why didn't you take into account this effect on the slope?	Yes farmers only have one cropping system included in the network. We could not allow the slope of diversity indicators to vary by cropping system ID because this slope was non identifiable in the vast majority of cases. 41 % of cropping systems were monitored only one year and crop diversity tended to be very similar from year to the other for a given cropping system (when monitored more than once over time).
-“for each climatic region, cropping system TFI was modelled as a function of period of description and number of crops “ -> What do you call period description?	Period of description is usually a given year (2011, 2012 ...). However, when farmers enter the network, they answer a questionnaire which usually covers a period of 3 years. In this case, data represents the average (proportion of crops, practices) over 3 years. This period of description, and its interaction with climatic region, and crop, is taken into account as random effects in the new version of the article.

->How is the effect of disease pressure taken into account? Do you take a year effect in your model? -> Do you not take a crop effect in your model?	Crop was not taken into account in the original polynomial quantile regression model but is now explicitly taken into account in the new version of the article.
Supplementary Table 1 Why not put the years as random effect? Why is there a line with "period 2014,2015,2016"? Why display an AIC, if you are not comparing models?	Years (or period of description, see previous comment) are now included as random effects. They were not originally included as random effects in the polynomial quantile regression model because crossed random effects were not possible (available statistical software only allowed one, which was cropping system ID). No AICs are presented anymore, as a unique, a priori defined, and conservative model is retained for each analysis.

Reviewer 2

Reviewer's comments	Authors' answers
GENERAL COMMENTS	
In this work the authors analyzed the potential reduction in pesticide use when increasing the number of crops within a rotation making use of a large database of French arable crop farms. As agronomist I consider this work very valuable and timely, since it provides reliable data to support one of the classical assumptions in crop production: i.e. less pressure of biotic agents in well-designed crop rotations.	Thanks for your interest in our work. Please note that diversity indicators were now modified to reflect different facets of crop diversity (botanical families, average number of crop per botanical family, diversity of sowing periods and cover crop frequency). New analyses were also carried out at the crop scale to account for crop identity effects.
In my opinion this work should account for productive trade-offs or externalities. In any case a TFI or pesticide use reduction is successful if the primary objective of agricultural production is not fulfilled. If not, we are at risk of recommending diversification solely to reduce pesticide use while depending on imports to fulfill the yield/caloric/protein demand. A standardized TFI accounting for this/these primary objective/s would be very interesting to account for the potential trade-offs of diversification. If there are no enough data for a "standardized" or "yield-based TFI"	Presenting and discussing the new set of analyses already requires substantial space (graphs, text), so we feared that adding the requested information would decrease the readability of the article. We nevertheless agree that this needs to be – at least – mentionned. Hence, a paragraph was devoted to the importance of potential trade-offs between crop diversification and other performance indicators, such as productivity and profitability. Studies investigating certain trade-offs on subsets of the dataset used here were mobilized, among others.

estimation, a paragraph should be devoted on this.	
In my opinion, one of the limitations of this work is the framework on “pesticides”, without focusing on control strategies to specific biotic agents: e.g. weeds, insects, fungi or bacteria, etc., or even gastropods (which can be important in some specific crops such as rapeseed). Although I understand that the objective of this work was more general, the main findings and take-home messages could be a bit simplistic. Could the authors discuss on this?	We agree. We now provide information on total pesticide use, as well as herbicides, fungicides, and insecticides.
I suggest as well providing values of TFI reduction for values of return delay lower than 5 years. Apparently, the model (Figure 3) seems to indicate a strong reduction in TFI as fast as when breaking the monoculture and increasing the return delay to 3 years. The improvement from 3 to 5 years seems to be marginal in many cases.	This is comment is now anymore relevant in light of the new analyses and corresponding text.
ABSTRACT	
Abstract. This work is mainly focused on temporal diversification (i.e. crop rotation of sole crops) with the unique exception of “méteil” and on annual crops. Given the increasing literature on spatial diversification (e.g. intercropping) and on perennality it would be informative to mention the focus on temporal diversification of annual crops.	We now refer to temporal crop diversification. However, we rather not precise « annual » as crop rotations in our dataset can include non permanent grasslands.
L9. What do authors mean by “afterward”? Could they provide a temporal framework to this statement?	This sentence was removed because irrelevant in light of the new analyses.
INTRODUCTION	
L18. “allowed by intensive pesticide use”. I suggest changing by “allowed by synthetic pesticide availability”. Not all simple cropping systems (understood as monocultures or rotations with very low number of crops) use pesticides intensively, that is possibly more related to the level	Thanks for the edit. We changed accordingly.

of productivity and economic margin of benefit.	
L34. I suggest focusing the text on references that actually provide data on the statements. For instance, McDaniel et al. (2014) do not provide data on the diversification of soil microorganisms when diversifying crop rotations.	This reference was removed.
DISCUSSION	
L132. What do authors refer to “low or medium crop diversity”? Can they provide values to this statement?	Concrete values are now provided when referring to different levels of crop diversity.
L137. Authors state that “...the proportion of barley and oilseed rape with high pesticide requirements...”. Barley is a winter cereal that is very well adapted to many pedoclimatic scenarios and related to a low usage of pesticides in different climates than the ones covered in this work (e.g. Mediterranean). I wonder if the final use of each grain is also a key element to take into account when assessing pesticide usage? For instance, barley for malt producers are more intensive in their practices than when barley is used for feed. Could the authors discuss a bit on this?	We agree. This was actually a poor choice of example in our case also. We now illustrate highly pesticide reliant crops with potatoes, and to a lesser extent, sugarbeet, and oilseed rape. We also completely agree that end use can largely determine crop pesticide use (eg wheat for biscuits, bread, feed..). The information present in our database was not always sufficient to go into such precisions but we added a section in the discussion on « other drivers of pesticide use », where this is acknowledged.
L153-154. Similar to my previous comment, management practices are not the sole cause of pesticide use. The final use of each grain produced is probably a more important driver of pesticide use, with higher intensities when grain is oriented towards human consumption (e.g. aiming at a low mycotoxin concentration) or seed multiplication compared to grain for feed or forage.	You’re right. See previous comment.
MATERIAL AND METHODS	
L234. Wormer. Please, check.	Good catch. This was a typo. We changed into « warmer ».
L235. Joly et al. (2010) citation. Please, check.	Good catch. This was corrected.
L242-245. A small proportion of cases was monitored for more than 5 years. Could the authors explain which impact could have that limitation on the results? Were these	Greater effort was made to precise how data was collected and at which scale it was analyzed.

cases equilibrated between climates studied?	Please note that temporal coverage of a cropping system cannot influence crop diversity values. Here, temporal crop diversity is assessed each year by looking at the crops present among a set of plots following the same crop rotation. The number of plots monitored a given year also aimed to represent diversity of crops present. In any case, a filter of minimum 8 plots was applied to ensure maximal representativity. A diversity of descriptive statistics on the dataset are now provided for complete transparency.
L248-250. That statement makes me posing the next question: is diversification a driver of TFI reduction per se or is the consequence of introducing less productive (in economic terms) crops in the rotations? Sunflower and peas are examples of field crops that usually led to lower short-term (i.e. at the crop scale) revenues to farmers when compared to cereals.	The new set of analyses now allow to disentangle dilution effects associated with the identity of specific crops from regulation effects (crops introduce in a more diversified crop rotation are less pesticide reliant) . Moreover, we now investigate relationships between crop diversification and crop composition to identify whether these two effects are actually combined in more diversified crop rotations.

Reviewer 3

Reviewer's comments	Authors' answers
GENERAL COMMENTS	
In the present study the authors evaluate whether temporal crop diversity (i.e., crop rotation) explains variation in pesticide use over a large spatial scale using a farmer site network of over 1000 locations. They find reduced pesticide use with increased degree of crop rotation for some crops and with regional variability. The manuscript needs substantial work, but could be a valuable contribution.	Thanks for your interest in our work. We appreciate your thorough review.
I have a few major concerns, points that need clarification, and suggestions to improve the manuscript :  1. The need for this study and the research gap it fills could be better motivated. As is, its novelty is not immediately clear. 2. The methods and experimental design are frustratingly vague. Specifically, the grain and extent of temporal coverage and the variability in what constitutes a 'cropping system' are not sufficiently detailed. 	Thank you for highlighting your main concerns. We believe that all of them were adressed.  1. The novelty of the article was better highlighted at the end of the introduction. 2. Description of data collection/handling was improved for greater comprehension. We also now define what we call crop rotation and cropping system and at what scale indicators are computed. A main figure was added (Figure 1) to try to make this completely transparent. 3. Crop rotation diversity is now characterized through other indicators, more specifically :  - Effective number of botanical families

3. Relatedly, the authors use of two indices to characterize crop rotation in one instance and misleading in another. Return delay is not actually measured here 4. The assumptions of statistical models used are not thoroughly tested. For example, I have concerns that temporal autocorrelation is still an issue for these data. 5. The discussion could better present caveats and offer more inference on the crop and regions specific responses observed.	 - Average number of crops per botanical family - The interaction of the two latter - Sowing diversity in the case of herbicides - Cover crop frequency The term « return delay » is not used anymore. 4. Accounting for temporal autocorrelation was not relevant because most cropping systems (41%) were represented by one observation in time. A diversity of descriptive statistics (including temporal coverage of the cropping systems) is now included.
The authors may be dismayed to see that my comments are numerous and critical. However, these are in line with the potential impact of the findings and the caliber of the journal. I commend the authors for leveraging an impressive dataset at a timely and important question in agro-ecological management. I believe that this study could be of interest to the readers of Nature Communications if thoroughly improved through a major revision. Please see my specific comments are below. Comments of most concern are in bold.	We understand and appreciate the thoroughness of this review.
Title: Please specify that this is temporal crop diversification (or more simply crop rotation). The evidence on the effects of ‘crop diversification’ is already spatiotemporally scattered, you could be refreshingly clear.	We agree and the title was changed accordingly.
ABSTRACT	
L4: These effects have been quantified, so this gap statement is not true. See: Larsen A E and Noack F 2020 Impact of local and landscape complexity on the stability of field-level pest control Nat. Sustain. 4 120–28 Nicholson, C. C., & Williams, N. M. (2021). Cropland heterogeneity drives frequency and intensity of pesticide use. Environmental Research Letters, 16(7), 074008.	No study investigated the effect of temporal crop diversification on total pesticide use, studies focused either on insecticide use and/or spatial crop diversification. The abstract and introduction were modified to make this point clear.
L6: ‘cropping system’ is unclear terminology. I get into this more in my review of the Methods	Cropping system is now defined as a set of plots within a farm which follow the same crop rotation.

L9: return delay is jargon and somewhat meaningless if the time period over which it is calculated is unknown. I get into this more in my review of the Methods	Please note that temporal coverage of a cropping system cannot influence crop diversity values (or return delay in previous version of the manuscript). Here, temporal crop diversity is assessed each year by looking at the crops present among a set of plots following the same crop rotation. The number of plots monitored a given year also aimed to represent diversity of crops present. In any case, a filter of minimum 8 plots was applied to ensure maximal representativity. The material and methods section was modified and a main figure (Figure 1) added to make this point more clear.
L15: 'crop rotation' is missing from keywords. This is a term that is used widely by both academics and practitioners, so it should be included.	Good catch. Crop rotation was added as a keyword.
INTRODUCTION	
L18: Oversimplification is a vague term. I think I know what you mean, but I suggest using something else. Also, this sentence implies that so-called 'oversimplification' is an effect of pesticide use but couldn't increased pesticide use be an effect of 'oversimplification'. In other words, the directionality of cause and effect here is more complicated than presented. I suggest revising this opening sentence completely.	This sentence was reworded.
L26: Omit the parenthetical statement. It adds jargon.	Done.
L31: is a meadow really a crop? And what is a 'hardy crop'? These parenthetical examples don't clarify much. Remove or give more concrete examples	The parenthetical examples were removed.
L36: please add a reference. There are many to choose from, but Root 1973 is an obvious starting point	Done. The reference provided, along with another, were added.
L38: ref 15 had been published, so should not cite preprint.	Good catch. This reference was modified accordingly.
L40: At this point in the manuscript you have used the following terms: cropping system diversification, crop rotation, temporal crop diversification, temporal diversification, diversified crop rotations, low return delay. Please pick a term and use it consistently	Good catch. This was homogenized for greater clarity.
L42: Why is the generalizability of these findings questionable? I would think that evidence from several long-term studies in different geographic regions	Evidence that diversified crop rotations can allow substantial reduction in pesticide use is currently supported by a few long-term experiments.

(“scattered around the globe”) is generalizable. Unless the results point in different directions. But this is not stated. Moreover, how does a single region study, like the one here, address this issue of generalizability. This critique needs to be clarified or revised and made more in line with the manuscript.	However, these experimental cropping systems tend to be designed to maximize this objective, often at the expense of others (e.g. financial profitability, labor requirements) and are hence not necessarily adopted by farmers. Studies have also shown that farmers, experimenters, and scientists do not have the same perception of pests, which can result in contrasted pest management practices before the same field conditions. Thus, it remains unknown whether crop diversification, as currently implemented in real farms seeking financial profitability, allows to reach substantial pesticide reduction. The text was modified accordingly. On a side note, what you refer to as a single study region encompasses a whole nation with a diversity of climates, representative of what can be found in Europe. Moreover, Lechenet (2017, PhD thesis) showed that the DEPHY network was representative of French agriculture.
L45-47: “no straightforward quantitative relationship between crop diversification and pesticide use was established” But isn’t this exactly what Nicholson & Williams and Larsen & Noack found? Albeit, in the former spatial crop diversification. Please make this research gap more precise. Perhaps I don’t understand the research gap you are identifying. This could be the result from the typo in L46, where there is an omitted word or phrase.	We now make it clear that studies focused essentially on spatial crop diversification, and insecticide use for Larsen & Noack.
L53: ‘cropping system’ is unclear. What kind of spatial unit is this? please use a clearer term (e.g., farm parcel) or define.	Sorry for this lack of preciseness. Cropping system is now defined as a set of plots within a farm which follow the same crop rotation.
L53: A brief description of the DEPHY data would add robustness, and help the general reader from having to sleuth for what it means in the methods	We agree. The whole description of the DEPHY network and dataset was completely rewritten and a figure added to facilitate comprehension.
L54: This first indicator is not adequately defined. When talking about crop diversity issues of scale, so familiar to ecologists, are also paramount. To understand this first measure you need to define the grain (i.e., days, weeks, months) and extent (i.e, 1 year, 8 years, 3 decades) at which ‘the number of crops present’ is being tallied. After reading through the whole manuscript, I am still unsure of both the grain (I assume years) and extent (8 year periods??). I think you could talk more carefully about	Sorry for this. Again, this probably arised from lack of clarity on the definition of a cropping system and on assessment of diversity indicators. We believe to have considerably improved this. Please note that temporal coverage of a cropping system cannot influence crop diversity values. Here, temporal crop diversity is assessed each year by looking at the crops present among a set of plots following the same crop rotation. The number of plots monitored a given year also aimed to represent diversity of crops present. In any case, a filter of minimum 8 plots was applied

temporal crop diversification by making these explicit.	to ensure maximal representativity. This information was added.
L55: This second indicator is fundamentally flawed, or at least what you are calculating is not accurately measuring return delay. I expand on this below.	Agreed. The new version of the manuscript makes no mention of « return delay », except as a perspective.
RESULTS	
L59: I understand that this index is established and used previously by EU agencies and this group. However, a brief description of what it measures would be useful. This should include stating its directionality (i.e., is a value of 15 indicative of high pesticide use). It should also include a caveat that this metric does not capture ecological/health effects bc it does not account for toxicity.	The distribution of TFI is now shown and analysis highlight the differences among crops. We added a text to precise that TFI was an indicator of pesticide reliance, not pesticide toxicity.
L60-66: I have a hard time following the logic of this. You identify that factors other than crop rotation could explain variance in TFI (L60-63). Then you present the fact that there is a non-linear relationship (so call 'bell-shaped relationship'), in and of itself a finding. To you, these seem to motivate the use of quantile regression. However, to my knowledge this method is typically used when data are heteroskedastic. Please motivate the use of quantile regression more clearly (i.e., if it to test the relationship between crop rotation and TFI at some limit, then state that clearly and state why this is of interest).	This analysis was completely dropped as not judged satisfactory by all reviewers. The choice for this original analysis was motivated by the fact that cropping system TFI was highly influenced by crop composition. Analyses are now performed at the scale of each crop, to disentangle crop identity (dilution) and diversity (regulation) effects.
L64: Those unfamiliar with quantile regression will not understand what this tau value means. Suggest saying '95th quantile'. Again, justify why you are picking this limit, either briefly here or in the methods	Quantile regression analyses were dropped from the new version of the manuscript. More straightforward analyses are proposed.
L65: I don't follow the inference made in the parenthetical statement. Please clarify	Quantile regression analyses were dropped from the new version of the manuscript so this comment was not considered as relevant anymore.
L66-67: 'slope of the upper-limit of the relationship between the number of crops in the rotation and cropping system TFI'. This is a very derived measure and overly technical. Please provide a plain language term or description of what this slope measure captures. This will help the general reader.	Quantile regression analyses were dropped from the new version of the manuscript so this comment was not considered as relevant anymore.

L69: based on the abstract I know that the pesticide use increased, but again you use of overly technical language obscures this. I find 'upper limit of the relationship' to be cumbersome.	Quantile regression analyses were dropped from the new version of the manuscript so this comment was not considered as relevant anymore.
L71: is a TFI value of 6 high? Again, please provide the directionality of this measure.	The distribution of TFI is now shown and analysis highlight the differences among crops.
L72: 'Oceaning' spelling	The whole result section was rewritten and this comment did not appear relevant anymore.
L79: From what one has read in the manuscript so far it is uncertain how many crops can be in rotation. This stems from the lack of clarity around two important aspects of you design: 1) what a cropping system means and 2) the time period over which crop diversity is being tallied. Lack of clarity on the first item means we don't know if a cropping system can have more than one field and therefore more than one crop in a given year. Lack of clarity on the second item means that we don't know if you are comparing 'cropping systems' over the same amount of time (e.g., 8 years) or if you are comparing systems with 4, 8, 12, etc. years' worth of data.	It appears clear from your comments that this was poorly described. Sorry for that. Greater effort was made to make all this clear. The description of the dataset was completely rewritten and a main figure (n°1) was added for greater clarity.
L80: proportion in space or time? Please clarify what this is a proportion of. Moreover, please confirm that these proportions have a common denominator (e.g., 4 years of maize out of 12 total years). Otherwise, is it safe to compare these? See previous comment	The whole result section was rewritten and this comment did not appear relevant anymore. The corresponding result figure was removed. Assesment of diversity indicators was further precised (in particular that number of plots monitored aimed to capture crop diversity and that crop rotation to be defined over at least 8 plots, as a safe guard).
L77-88: These results are overly descriptive. You make multiple statements about increase or decrease with number of crops in rotation, but do not support them with statistical tests. Moreover, what is the point of this result? You seem to be grasping at changes in temporal evenness (i.e., shift away from monoculture) with increasing temporal richness. The relationship between richness and evenness is deeply explored in community ecology, and there are more quantitative ways to present these data. For example, diversity profiles (see refs below) or	All results are now supported with statistical tests. The whole results section was rewritten in light of new analyses. Proportion of the 16 main crops (non aggregated in groups) is now presented in Table 2 and in supplementary Figure 13 New regression analyses investigate the relationship between diversity indicators and the presence of specific crop species.

species dominance metrics. Also, your figure confuses more than it explains. This is in part because in some cases you are aggregating multiple crops into larger groups (e.g. fodder) and in other cases a crop is singular (e.g., maize, barley). Please revise this section by reporting a more quantitative analysis of the temporal evenness. If this is not the aim of this section, I still suggest revising the presentation of data, especially Figure 2 https://besjournals.onlinelibrary.wiley.com/doi/pdf/10.1111/2041-210X.12349 https://ericmarcon.github.io/entropart/reference/DivProfile.html	
L92: Why only eight crops? What are they? What quantitative criteria were used for their selection?	These 8 crops included dominant crops, high pesticide reliant crops, and one crop farmers tend to use to diversify their cropping system (pea). We now focus on the top 16 major crops of the dataset, which all represent at least 1% of total crop proportion (see Table 2).
L93: exponential? In the discussion you mention log-linear. Just be consistent. Also, you did not 'establish' this relationship. You 'found' it, or your analyses 'showed' it	The whole results section was rewritten and this comment did not appear relevant anymore.
L94: Was there a significant interaction between crop rotation and climatic region for other crops. Please state non-sig results too.	The new analyses do not focus on interactions between crops and climatic regions (these are considered as random). We aim to answer the question « are crops introduced in more diversified crop rotations less pesticide reliant, irrespective of crops present in the crop rotation » ?
L99: what about the other crops? You only report 5/8. Seems like you are only cherry-picking significant findings.	See previous comment. New analyses do not focus on crop specific responses and include all 16 main crops. Crop specific responses are nevertheless acknowledged through random effects and discussed when necessary.
L101: You include the fits of multiple regions in each panel of your figure. This leads one to believe that we are meant to compare relationships between regions, but did you test for an interaction between rotation and region on crop TFI? If so please state the outcome of this test. If not, why not?	Including interactions in the new set of analyses would have generated complex models with rank deficiency (6 climatic regions and 16 crops, not necessarily present in all regions), which were not of specific interest condising the central question of the article (see previous comment).
L105: Again, these results are rendered almost useless by not understanding what 'cropping system' means.	Truly sorry for this. The description of the dataset was completely rewritten and a main figure (n°1) was added for greater clarity.
L107: Is this actually an increase in years between crop return? Also, I thought 'return delay' was estimated as an	The notion of return delay was completely dropped, the new version of the manuscript focuses on diversity of the crop rotation.

inverse proportion, so how do you have whole integer values? This is unclear and potentially misleading	
L107-118: Nice. You do a much more balanced job reporting results (both sig and non-sig) and explaining where you found interactions or not. Please do this above	Thank you. Unfortunately, this part was completely rewritten in order to comply with the new analyses. In this new version of the manuscript we tried to report both sig and non-sig results
DISCUSSION	
L123: Drop first 3 words. Not needed	The discussion was nearly completely rewritten. These three words were not included in the new version.
L126: or exponential? You use a log link, so just pick the correct term (I am not sure what this would be given a Tweedie distribution).	Linear effects on log scale correspond to exponential effects on the response scale. Anyways, we did not see the use for such wording in the new discussion.
L131-134: This is a bold and important statement but requires a little more careful presentation. First, you need to be explicit that these results are based on use patterns specific to the crops in the target region. For example, in other parts of the world, wheat does not receive a lot of pesticide application. More broadly and going back to your research gap, does this study contribute to generalizability of evidence? Especially given that you see such different responses between regions. Second, this statement is only true given the crops included in this analysis. Rotation with other crops, not considered here, may not show this type of hump shaped relationship. Third, this statement assumes that all crops in rotation are conventionally managed. These caveats (and probably others) need to be raised.	Differences in pesticide reliance between all 16 main crops are now clearly illustrated and supported with statistical information. We focused on pesticide use in all 16 main crops but please note that diversity indicators accounted for all crops present in the rotations (103 crop species in total). The significance of crop diversity effects is now evaluated across all climatic regions (the interaction is considered as random) to test the generalizability of the observed relationships. Analyses focused on conventional cropping systems (organic systems were filtered out) but not all crops were necessarily sprayed with pesticides, thus justifying the need for zero-inflated models in certain cases.
L140: where is the closing parenthesis?	The discussion was nearly completely rewritten.
L141: What is a diversification crop? And does it always mean low pesticide use? The example of legumes is misleading bc there are plenty of legume crops with high pesticide use (e.g., soy). Meslin is not widely known, so is it really a good example? Also, is meslin a 'crop mixture'?	We now refer to these crops as crops associated to diversified crop rotations. New analyses showed that these crops (alfalfa, ryegrass, spring barley) were usually of lower pesticide reliance. Meslin is now referred to cereal-legume mixtures.
L141: Completely unclear what you mean by regulating effect. Please clarify	This was to contrast with dilution effects (the fact that introducing a low pesticide reliant crop (e.g. ryegrass) will necessarily reduce pesticide use at the crop rotation level).

	We speak of regulation effect when a given crop is less pesticide reliant when introduced in a more diversified rotation. We tried to be clearer about this.
L143: stable after a point, right? Also, is this saturating effect your highlighting real, or a statistical artifact based on the link function specified in your models. To make this point you may have to explicitly test non-linear functions. Also, what is meant by 'production situation'? Please consistently use clear terms.	We meant « consistent ». This was changed accordingly. The link function is justified by the fact that the response variable is continuous and hence, to keep predictions within realistic boundaries. The concavity of the response isn't anymore visible. We changed production situation by « climatic region » to remain consistent.
L145-147: Yes. Sounds like what this study could have done to be more explicit about 'return delay'	This was presented as future avenues for agroecological research because unfortunately, not allowed by the current dataset.
L153: Are these only climatic regions? I thought they were also soil related?	This sentence was completely reworded. Climatic regions refer only to climate but other random effects account for the interaction between climatic region and soil type.
L154-158: This all comes off as very vague and hand-waving. Use of ellipses (...) does not help your case. Moreover, the point is that these practices could vary between regions no? Make that point and support it with evidence.	This section was completely rewritten to place emphasis on the fact that crop diversification could be seen as a means to diversify pest management practices.
L161: Careful. Use terms consistently.	We're sorry but we do not see what this comment refers to.
L162: And what exactly is the cropping system level. If cropping systems are field than this is a patch, but if their farms with multiple parcels than this is more of a landscape phenomenon. Is this statement going to depend on scale? Suggest to revise this sentence to be more clear and draw recommendations directly from the present analysis.	Cropping system is now defined as a set of plots within a farm which follow the same crop rotation.
L162-171: This is good. Much clearer than L139-142. However, most of these sentences could use references to support their points	Thank you. References were added.
L174: crop [field] vs. cropping system. Yes agree. And exactly to my point above and elsewhere.	We meant distinguishing tactical (at the scale of one crop a given year) and strategic pest management (at the scale of the crop rotation). We tried to be more cautious and replaced « cropping system » by « crop rotation » under numerous circumstances.
L179: unclear what is meant by 'similar mechanisms'	This sentence was completely rewritten.
L180: what is 'protein autonomy'?	This concerns mainly livestock farmers whom need balanced feed rations for their animals (e.g. grass or silage often need to be balanced by high protein feed). We precised that this concerned mainly livestock farmers.

L181: The word choice here is off. 'Experimenting [with]...' perhaps? Moreover, what are these types of innovative systems? Organic agriculture? Or something else?	This was better precised.
L185-205: This section kind of rambles and seems to be a grab bag of the benefits of crop diversification (beyond crop rotation). Please revise to draw more clear conclusions from the present work and how it addresses the research gaps you identified and how this work could, despite its limitations, motivate changes in crop selection at larger spatial and temporal scales.	We reduced this section accordingly and rewrote the paragraph focusing more on our own results.
MATERIAL AND METHODS	
L210-212: This definition does not help. In fact, it raises more questions, and potentially sources of unaccounted for variation and sampling bias. A few questions:  1. What is a plot? A whole field? Or area within a single field? Is the size of plots variable between and within cropping systems? 2. How many plots are in 'a set'? Is the number of plots considered variable between cropping systems? 3. How was it determined these plots were 'managed similarly'? And by whom? Are all the same crops grown across plots? 4. 'in the case of several', several of what? 5. How do you get the dose of application? From these descriptions? Lack of clarity on the above leads me to believe that there are unaccounted for sampling biases in these data. The authors' indicators of temporal crop diversification are essentially diversity metrics, which are going to be influenced by these kinds of sampling effects. Is it safe to compare the number of species between 1m² and 10km²? Most ecologists would suggest one needs to account for differences in sample coverage. Yet it is unclear if the present work is comparing sampling units (i.e., cropping systems) that vary greatly in their spatial extent and	To adress these comments, we :  - Rewrote the entire part of M&M concerning the description of the DEPHY network and assessment of diversity indicators - Added a flowchart figure of data collection and handling - Added a diversity of descriptive statistics of the dataset Nevertheless, here are the specific answers to the comments raised :  1. Yes a plot is a whole field. The size of plots can be variable across farms, and to much lesser extent, within. However, this has not been identified as a main barrier to greater crop diversity on a farm because fields can easily be divided. 2. The number of plots monitored is variable across cropping systems but aims to represent the diversity of crops present. We nevertheless added a safe guard of a minimum of 8 plots monitored to be included in the analysis. 3. Farmers and DEPHY network advisors decide which fields belong to which cropping systems. In practice, usually one cropping system is present on a farm. However, in certain cases (e.g. absence of irrigation on part of the farm), multiple cropping systems can coexist on the farm (e.g. sunflower-wheat on non-irrigated fields and maize monocropping on irrigated fields). Plots belonging to the same cropping system do not necessarily have the crop a given year, they simply follow the same crop rotation under the same constraints (e.g. absence/presence of irrigation). Hence, we assumed that crop diversity described for a given cropping system a given year was representative of the crop rotation diversification (space for time substitution)

number of sampling units (points 1 and 2 above, respectively). This is important because these differences would influence the number of crops present within and between years. If cropping systems are variable in their number, size, and management then this should be included in models as fixed effects. In general, the description of these data is very vague. Please provide more explicit details about the way the data are collected, including detailed information about the spatial and temporal scales of collection. These descriptions should be supported with supplemental analysis that show the variance in cropping system in line with the questions above. Otherwise, I find it very hard to evaluate the results presented here due to the unknown influence of potential sampling bias.	 4. In the case of several cropping systems, see example in 3. 5. French farmers have the obligation to keep a register of all their pesticide applications at the field level. This was shared by farmers.
L212-214: Its is unclear how simply providing this information ‘reduces the importance of crop x year effects’. Also, its unclear what these effects might be and why its important to reduce them in light of the analyses presented here.	This 3 year average simply aimed to be a representative entry-level network reference for each farmer.
215-216: It is still unclear what information farmers are providing, and how it is aggregated.	This should be more clearly explained now. For each plot and each year, the farmers provided information on the crop and pesticide applications (date, dose, commercial product, surface applied...). For analysis, all plots with the same crop within a given crop rotation were averaged and we hence obtain one observation per crop and year. We hope Figure 1 greatly helps understand the procedure.
L220: word choice. ‘resorted to...’ perhaps instead ‘used’	This sentence was changed when rewriting this section.
L221: Figure S1 does not clarify this point. In fact, it raises further questions. It depicts the number of crops, over what time period? What if a cropping system had more than one crop in a given year. Are these included in this summation?	This figure was not considered for the new version of this manuscript. Again, time coverage is not an issue as crop temporal diversity is assessed each year, through the diversity of crops present across a set of plots which follow the same rotation.
L222: ‘over a 3-12 year period’ what is the distribution of cropping systems for each of these durations? Please provide a supplemental figure (e.g., bar plot) depicting the number of farms and the duration of their ‘period of description’. Variability in this period raises concerns about you measure of ‘return delay’. I raise this issue in more detail below.	Descriptive statistics, including temporal coverage per cropping system, are now provided. In 41% of the cases, one cropping system = one crop rotation (because described simply once in time).

L222-223: 'i.e. 3-year entry description and follow-up annual description' I do not understand this.	This is line with the comment made L212-214. When entering the network, farmers are asked for the average proportion of each crop and pesticide applications per crop over the last three years to provide an entry-level reference. If farmers decide to stay in the network, descriptions are then made on an annual basis.
L224: '194 to 3894' it is unclear why there is such a range in the data available for this indicator. Please clarify	One point corresponds to a given crop in a given rotation. Hence, geographically localized crops (e.g potatoes) or minor crops are much less frequent than crops grown nearly everywhere (e.g wheat).
L225-226: Did you test for and detect temporal auto-correlation? Then did you check to see if the random effect actually accounted for the TAC? Please provide the outcome of these tests. I would imagine that auto-correlation would be very high for these data and you will need more complex model structures (e.g., AR1) to control for TAC.	For most cropping systems, only one time point (i.e. rotation) was available. Hence, accounting for potential temporal autocorrelation within the subset of systems monitored in time was not feasible. See Supplementary Figures for descriptive statistics.
L228-236: Throughout the text you describe pedoclimatic regions. However, based on these methods it these regional distinctions appear to only be climate based. Are soil differences considered? Please clarify	Good catch, we changed this into climatic region. However, please note that we included an interaction between climatic region and soil type as a random effect in the analyses.
L235-236: Reference needs to be fixed. 'to appreciate' not best choice of words	Thanks. This was modified.
L242-245: By the authors own admission (i.e., 'unfortunately') the index described is not a measure of return delay, which is clearly defined (L55) as, 'number of years separating sowing of the same crop'. It is misleading to describe this measure as return delay, it is a measure of temporal evenness, and an imperfect one at that. Consider the two 8 year cropping sequences: 1. maize, maize, barley, rye, pea, barley, maize, maize 2. maize, barley, maize, barley, maize, barley, maize, barley The method proposed would assign a maize 'return delay' value of 0.5 for both, when their actual return delay is quite different (4 vs. 1 year for maize, respectively). Return delay is a clearly defined and important agriculture practice. I cannot recommend this manuscript to be published if it uses the current index but reports results for effect of 'return delay'.	You are right, the term "return delay" does not fit the previously calculated variable. Based on the comments of the various reviewers, we have decided to characterize crop rotation diversification in a more detailed and exhaustive way, using 4 indicators:  - The effective number of botanical families in the crop rotation - The average number of crops per botanical family - The effective number of sowing periods - Cover crop frequency

I suggest to either correctly measure return delay (but this seems like its limited by data availability) or to use a different term. However, even as a measure of temporal evenness I have concerns about this approach given lack of clarity on sampling coverage detailed above. This measure is going to be influenced by the number of years in consideration. This needs to be accounted for. Hill numbers could provide a useful metric. I suggest two articles for you consideration: https://esj-journals.onlinelibrary.wiley.com/doi/10.1111/1440-1703.12102 https://onlinelibrary.wiley.com/doi/10.1111/oik.07202	
L245-246: this assignment makes no sense, see example above.	Due to the changes in the manuscript, this comment does not appear relevant anymore.
L253: this definition makes no mention of the dose used. I prefer the definition used in the authors' previous work: 'the mean number of treatments per hectare with commercial products, weighted by the ratio of the dose used to the recommended dose' (Pelosi et al. 2013). Also, shouldn't this definition mention that this application is being used for a given crop, in a given year. Also, please specify that this is pesticide product, rather than pesticide active ingredient.	We have modified the definition as suggested and specified that this index is calculated per hectare for a given crop and year. We specified that a commercial product could contain several active ingredients.
L256: check the indexing in this equation. I am not sure, but I think you may want a j iterator for the inner summation (e.g., D_{ij}, S_{ij}) because these are specific to the jth crop	Good catch. However, only TFI calculated at the crop level has been retained in this revised version of the manuscript. The equation has been modified accordingly and indexing with j is no longer relevant
L257: Again, how do you know the applied dose? Is it self-reported?	The applied doses of the commercial products were reported by the farmers for each plot (with a given crop) and each year. It was specified in the text.
L258: 'St is the total plot area', What does this mean in the context of this study, see major comment above. This is not the total area or the cropping system, correct?	We hope that the more detailed description of the dataset, along with the flowchart in Figure 1, will provide a better understanding of what the total plot area (St) is.
L259-260: what do you mean by expressed? Also, what units are the applied and reference dose in? How did you harmonize and handle data when units differed (e.g., g/ha vs. L/ha)?	This term was removed from the text. The applied dose and the reference dose are systematically given in the same unit. The unit is specific to each commercial product

L260: Please provide caveat here that TFI does not measure the degree of environmental disturbance of pesticide use because it does not account for toxicity.	We added the following sentence in the text: “It should be noted that TFI quantifies pesticide reliance but does not measure the ecotoxicological impact of pesticides.”
L266: is this ‘list’ published? Please reference or provide a list of products that were excluded.	The complete list of products classified as biocontrol is available via the following link : MASA. Note de service DGAL/SDSPV/2022-949 du 22/12/2022 : Liste des produits phytopharmaceutiques de biocontrôle, au titre des articles L.253-5 et L.253-7 du code rural et de la pêche maritime https://agriculture.gouv.fr/quels-sont-les-produits-de-biocontrôle (2023). The reference was added to the Reference list
L272: Neither of these papers use the term ‘envelope curve’, suggest more relevant literature. As per comment above, here is where you could motivate the use of quantile regression for at analyses at a given limit.	Due to the changes in the manuscript, this comment does not appear relevant anymore.
L276: ah so it is a practical use case.	Due to the changes in the manuscript, this comment does not appear relevant anymore.
L277-279: I have a hard time following this	Due to the changes in the manuscript, this comment does not appear relevant anymore.
L281: fit, not ‘fitted’	Are you sure? Both seem correct. We changed nevertheless.
L282: Do you adjust p values to account for multiple comparisons resulting from these separate region models? Please consider doing so	P-values from multiple comparisons were adjusted using the false discovery rate adjustment.
L286: And did inclusion of this term eliminate temporal autocorrelation? Please check the autocorrelation functions of you models (e.g. stats::acf(mod)). If autocorrelation exists, please account for this further.	For most cropping systems, only one time point (i.e. rotation) was available. Hence, accounting for potential temporal autocorrelation within the subset of systems monitored in time was not feasible. See Supplementary Figures for descriptive statistics.
L289: ‘eight previously presented crop species’ Again, it is so unclear what the temporal extent of your indices are. Here, you are stating that it’s the 8 previous crops, so are you only using those cropping systems with >=8 years of data reported. Please clarify.	The eight previously cited crops, which are the dominant crops of the dataset (16 are now included in the current manuscript). The new material and methods section should limit this confusion.
L292: fit	Are you sure? Both seem correct. We changed nevertheless.
L294: Why do you have a point mass at zero? Are there a lot of cropping systems that do not apply pesticides. It would useful to see summary plots of the range applied doses	Such plots are now provided and in most cases, zero-inflated models are supported.

L296: Wait, what is ‘period of description’? A duration of time or a single year? If the former, what is the rationale for including it. If it to try an account for differences in sampling coverage, then I don’t think this is doing what you want it too. Look into including duration as a fixed effect or offset term. If the latter, then simply say this and be consistent elsewhere.	Period of description can be two things :  - The 3-year average provided by farmers when entering the network - An annual description (when farmers remain in the network) Note that temporal crop diversity is assessed through the proportion of crops present a given year across a set of plots following the same rotation.
FIGURES AND TABLES	
Figure 1: Please make the point size scaled to the number of years under consideration. The first sentence of the figure caption is supposed to clearly say what it shows, use of technical terms (see above) makes this hard to understand. Why are there no confidence intervals? Please add CI	This figure was not included in the new version of the manuscript. CI are now included everywhere.
L434: ‘and/or’ which is it? As in, does a 2 mean the model had a significant linear and quadratic term, or only a significant quadratic?	Due to the changes in the manuscript, this comment does not appear relevant anymore.
Figure 2: This figure is confusing because it is unclear how many years are considered to calculate crop proportion (is it standardized?) and because of the aggregation to crop groups for some species. Please revise this figure	This should now be clearer with the new M&M and Figure 1.
Figure 3: Please provide confidence intervals. A point for the discussion, but you never explain what the mechanism is for the non-linear trend. Why does it flatten? And why would slopes/intercepts differ between regions. See Discussion comments above.	CI are now included everywhere. Effects display much less concavity now so this comment does appear relevant anymore.
L445: ‘over a given period’ suggest to make point size scaled to the length of this period.	See previous comments. Temporal coverage does not affect diversity indicators because they are assessed through the proportion of crops present a given year across a set of plots following the same rotation.
L446: Nature Publishing Group has strict guidelines for what needs to be included in the captions. And your reporting summary says you have followed these. But double check to make sure that you are reporting the type and terms of the model used. What are the random effects?	We read the guidelines and looked at other Nature articles with similar figures. We referred to Supplementary Tables in Figure captions for further information on models.
L447: First time using the term population. What is a population here? Somewhat confusing	This is pure mixed modelling terminology. Population level refers to an average subject (predictions unconditioned on

	random effects) where individual level predictions taking into account random effects. This was specified.
Figure 4: please provide CI	CI are now provided.
L461: Suggest to say 'after finding a non-significant interaction'	Due to the changes in the manuscript, this comment does not appear relevant anymore.
Figure S1: The information provided in the caption of this figure is not sufficient to understand what it is depicting.	This figure was removed but we tried to account for all these comments in the new figures.
Figure S2: Again, NPG has guidelines on what must be included in figure captions. This caption does not follow those guidelines (e.g. what do the boxes and lines depict?)	We precised that the figures were boxplots depicting interquartile ranges.

REVIEWER COMMENTS

Reviewer #1 (Remarks to the Author):

General comments:

The authors have done a great deal of work to take into account the many comments made by the various reviewers. The new version of the article is now quite different from the previous one. It concentrates on the analysis of one variable and the article presented is therefore more coherent and the analysis more thorough. The results are potentially interesting and new. I have reviewed the entire article again.

My main comments concern the form of the document:

- Be careful to present your methods and analyses correctly before analyzing the results. The fact that the MM is at the end of the document doesn't help. I sometimes found it hard to understand what you were talking about or the specificity of the analyses (some of your results are quite technical). If this article is aimed at a broad audience, there is room for improvement, I think.
- Perhaps selecting/prioritizing the results? For me, Figure 4-5-6 X is the article's added value. The presentation of the basic data (or basic analysis) is essential (as mentioned in the previous review), but some could perhaps be put in an appendix, to provide more detail on the core of the article?).
- The grammar and structure of some English sentences need to be reviewed.

Regarding the substance of the article:

- Be clearer about the results of excess zeros, marginal effects, and average effects. This is in line with my previous comment, but I think that the results on excess zeros and marginal effects will appear obscure to many readers if they are not better explained.
- Why didn't you analyze the effect of diversification on excess zeros?
- Present the overall results as well (not just the conditional effects and the results linked to excess zeros).
- How well do your models perform? Do they explain a large part of the variance in TFI? Unless I'm mistaken, I haven't spotted a fine enough analysis of this question.

Detailed comments :

Title :

Crop rotation diversification or 'only' crop diversification/ or crop rotation? crop rotation diversification seems strange to me ?

Abstract :

3761 crop rotation-> It's not easy to understand the differences in the text (globally) between 1334 (cropping system), 3761 (crop rotation) and 14556 (crop). Do you need to present these three levels of information? Which one do your analyses focus on?

L8: 'increasing the average number of crops per botanical family'. On my first reading (without reading the rest of the text), this sentence didn't seem easy to understand.

What exactly does 'effective number' mean? Does it refer to a particular concept? Why not just mention 'number of'? OK, I understood after reading the material and method. But as it's at the end of the document, I'd like to make it clearer/more comprehensible from here on?

L11. positive effect'. Positive in what sense? An increase? or positive for the environment (a decrease).

Keywords. Crop rotation is missing, isn't it?

Introduction

L18. Reference 1. Reference in French in a journal with no impact factor (?) dating from 1991. Is this subject so little covered?

Sometimes there are spaces before the reference numbers, sometimes not. Should this be standardized?

L34. Combine or combined?

L41. Do these references show that there are few studies on the subject, or are they just a few studies that have looked at this subject?

L45. Replace and are hence by and hence are?

L46. "Scientist" appears twice. The text must have been re-edited a bit quickly for resubmission.

L47. Before ? Strange translation from French..

L55. Investigate or investigated?

L57. Focal or main ? you sometimes change throughout the text. Homogenize.

L58. Plots-> so they are not fields?

L58. Does this definition of cropping system correspond to that of Sébillotte et al?

See my previous comment: it might be wise to simplify the presentation of the data. Do you really use this 'cropping system' level? Or at least start presenting the data that will be at the heart of your analyses?

L61-62. I don't understand this sentence. Why put 'a given year'? I must have missed something.

L64. Same question as above. Does "effective" have a particular meaning?

L65. At this stage we don't know how the cover crop frequency is calculated. Should it be mentioned?

L66. Is the notion of TFI common, shared enough by the scientific community (and more)? Not sure. This could make it harder for a wider community to understand the paper. Explain and define what this indicator is and how it is calculated?

Results :

The presentation of the data is very important and provides a good understanding of the core of the analyses. My question is: for this type of journal with a large audience, should this data be presented first in the results section, or should the authors directly address the strong results of their article (and put this information as a material supplement)? I would have chosen the second option.

L83. Is $r=0.60$ moderate?

L88. Does it also refer to correlations for cover crops?

L91. Title not easy to understand on first reading.

L92 "regression analysis" Which one?

L97. What does crop identity mean?

L97-99. Strange way of formulating this information?

L104. You put statistical details here, $t=97$, $df= \dots$, but not the other times, why?

L109. Strange wording?

L111. In my opinion, this part is the heart of the article and the results are interesting (but need to be reworked, see my comments).

L113 'Type II analysis of deviance'. Why use a type II here? Don't you take interactions into account?

L116 . Complicated to understand without having had any details of the statistical analyses you have carried out. The vast majority of readers won't understand what you mean. You should at least say that you have a distribution with a large number of 0s and values >0 , and that you are analysing these two components independently?

L116-120 So you are analysing the effect of each crop on the TFI. That's not the most interesting part, because your aim is to study the effect of diversifying crop rotations. The question that would interest me would be to see whether the effect of diversification is stable or, on the contrary, sensitive depending on the crop. The way in which you present this fragmented information does

not make it possible to answer/analyse the answer to your main question.

L117. Strange to find grassland here. Could you specify that these are temporary grasslands, included in rotations to avoid confusion?

L129. Your analysis between the two 'components' of the distribution (i.e. the zeros and the rest of the distribution) could be confusing.

Have you performed a complete analysis of the factors associated to the excess zeros, then the factors affecting the level of TFI? Why do the diversification indicators effects on the excess of zero have not been analyzed (this is a strong hypothesis?) ? Why do you present only the conditional mean, and not the average effect? Unless, I am mistaken, your model could be used to estimate the mean TFI for a particular level of diversification (i.e. considering the probability of being 0 , and the probability >0). Would be particularly useful, if there is an effect of the diversification on the excess of zeros.

I think there is some work to be done on presenting the method better and analysing the results (the most interesting ones) in a better structured way.

L130. Why an increase of 1 SD. It seems to me that it should be by one unit of the variable. There's something I haven't understood or that's badly explained here. It would be easier to interpret if you spoke in units that could be interpreted: for example, adding a species to a rotation results in XXXX. OK, your variables have been calculated using Hill's formula. Is it still possible to present the results more explicitly?

I think it will also be useful to present the performance of your model. A plot showing the predicted distributions of IFT versus the observed could help to judge the quality of the model, and identify whether there are risks of under/overestimation for certain values.

How did you select the best structure for our model? On the basis of AICs?

Discussion :

L168 "Negative effect of crop diversification"-> a negative effect means a reduction in IfT, right? Try formulating it differently, to avoid possible confusion. Would you rather talk about a decrease or increase in the use of pesticides? To be used throughout the text

L168 ' for all of the 16 main crops". Have you run a model for each crop to test this? Or tested the absence of crop*diversification indicator interactions?

L176: "across all crops and climatic region" same remark.

L293 'maximal'?

MM

L323. "Only non-organic" -> Conventional? Or are there other types of systems, a wide diversity included in the database?

L356. 'Significant' ?

L346. Why not include weather variables directly in the model? That would certainly be more accurate?

L367. Of the city' ? do you mean commune/ municipality?

L371 'of all main crops'? why 'all'?

L1391. This sentence has no verb?

L395. Why use this indicator rather than the number of crops?

L441. The structure of random models is very complex. Have you checked that all the terms are relevant?

Figures

L636. "therefore, crop rotation refers to the set of crops present within a given cropping system at a given time point?" at a given time point?

Figure2: the x bar is written in lower case in the legend and in upper case in the figure. Why are the axes of graphs C and D so wide when the data are not so spread out?
Put a definition of TFI here for readers?

Figure 3. subplot A. 'effective number of botanical families'. Add number of crops per botanical....'. Same remark for the x bar.

Figure 4. I can only see 15 crops here, whereas you mention 16 in the text... have you forgotten or have the contrasts been badly defined?

You haven't modelled the effect of diversification indicators on excesses of 0? why?

Grassland-> Put temporary grassland to avoid possible confusion?

Present the combined results rather than the conditional/zero inflated decomposition?

L668. Delta method? What is it? Have you presented it?

Improve the title ('estimate plot' -> ?)

Figure 5

You mentioned once in the text that the maximum number of crops per botanical gfamily is 4, but here on the graph the data go up to 8...

Title: in winter wheat ? (match the title of Figure 6)

Why only present the results for wheat?

if I'm not mistaken, the models presented in table S1 show some interaction (at least for the intercept) between the diversification indicator and the crops. It might therefore be interesting to present the results for all the crops? (same question for Fig 6)

is it normal that the relationship is linear with that you have used a log link function and that your data is presented back-transformed?

Reviewer #2 (Remarks to the Author):

I have carefully read the revised version of the manuscript which has been significantly improved by the authors. My additional comments are few and mainly focus on the agronomy behind the results obtained.

My first comment focuses on the title, which written as an affirmation. It is true that the title is supported by the data. However, does the small level of pesticide use reduction (3.0%) when diversifying one crop species per botanical family warrant it?

I understand that farms participated in the Dephy network on a volunteer basis, possibly the most innovative and/or environmentally-concerned ones. Could have this biased the results obtained? In relation to this, important production areas such the "Landes" (important maize production area if I am not wrong) seem to be almost uncovered by the network. Which could be the consequences on the data and the results obtained?

L8-11. Please, report the magnitudes of the decrease or increase. Readers without an agronomic background can be misled by these statements.

L15. Lacks a comma.

L172-173. Could the authors provide further information about what is considered by "simple rotation" and by "diversification" in the cited work? That will make the statement much more straightforward to the readers.

L181. In current times the term "industrial cropping systems" is often seen as pejorative by mass media and consumers. I would advocate changing the term by "... in cropping systems including industrial crops". In my opinion this last is more informative and less prone to misunderstandings.

L195-197. It is worth mentioning that introduction of forages entails a different market, machinery and/or livestock integration.

L198-202. Could this unexpected result be -partially- explained by the rather short duration of

data acquisition by the Dephy framework. For weeds someone would expect crop rotation effects on the seedbank in the long-term. That is, after a number of complete rotation cycles.

L216-218. I do not totally agree with this statement. The link between botanical families and machinery is not that straightforward. In my opinion, machinery is mainly determined by the use of the crop and/or its architecture. See, the different types of seeders (or planters) and harvesters (or mowers) used, for instance, in ryegrass, winter cereals for grain and maize cultivation.

L283-287. I am not totally convinced about the role played on weed control by different row distances: the species that we control in the spring or beginning of the summer in row crops by hoeing are different from the ones that are controlled in winter crops. In my opinion the main cultural control here is related to the different tillage periods (e.g. introducing a summer crop delays tillage and allows to mechanically destroy a winter weed before seed production).

L300. Typo

Supplementary Table 1. A typo (spring oat).

Supplementary Figure 7. A typo (potato).

Reviewer #3 (Remarks to the Author):

Please see attached

Reviewer #3 (Attachment):

Reviewer 3 here. Great job in the revision. It is a different manuscript, but a better one. I still think work needs to be done. As before, comments of most concern are in bold

Introduction

L18: Sentence unclear. Not sure what is meant by keystone. Maybe just say something like, “synth pesticide use characterizes simplified cropping systems”. Also simplified crop rotations is somewhat oxymoronic, no? I assume systems where crops are rotated are not simple.

L24: “Crop diversification”. Define what this is and note here that it can be both spatial and temporal, like “Crop diversification – increasing the heterogeneity of crop cultivars, species or families in space and time – is a fundamental...” Or something like this

L26: no need for ‘diversified’, its implied

L26-37: Excellent background information, well done

L40-41: your pivot statement (“...but only a few studies investigated the impact of *complex landscapes on pesticide use*”) is true, but I think it could be more specific. Two reasons. First, the “many” spatial crop diversification studies you identify are in fact studying varying degrees of “complex landscapes”. Second, I think you should be clear and stick to the specific focus of this article: temporal crop diversification (TCD)

L49: I like your use of “commercial” before, thus you could say the more succinct “... as currently implemented in commercial production systems”

L49: “allows to reach” change to “can achieve”

L52-54: Great points. I quite like this dilution vs regulation framing. Note, L55 could say “needs investigation”

L56: Is the aim really to quant how use “could be reduced” (this sounds more like a modeling objective). I would state this rather as “whether X reduces Y”. Please revise this objective statement to be more clear

L63: Again, be consistent with terms: Crop rotation diversity vs temporal crop diversity. Is there a difference? If so, you have not made it clear. If there isn’t a difference, then pick one term

L59-68: This is useful methods information and needed given the format of Nat. Comm. However, I miss a statement of your main research questions or hypotheses. For example, do you have expectations about the effect of TCD on dilution vs regulation? I am not asking you to post hoc create hypotheses, but rather state either questions or substantiated expectations.

L59-62: these study design details are still unclear (esp. L:61-62), but I am going to wait to read the methods before commenting.

Results

L72: Without reading the methods TFI is a little abstract. Can you provide units? Or at least a brief definition of what an average of 3.7 indicates?

L98: What is meant by “Their product”. More importantly, do you mean the actual values of these two variables multiplied together (which is what the Nilsson method does) or their interaction (Which is what it looks like in Table S1). Make sure you are describing (and doing) what you mean to. The figure caption text (and table S1) makes me think you looked at their interaction, but this is different than looking at their product as Nilsson and colleagues do. Please clarify

L99-103: I assume that these association with specific crops are for when these crops are included in the rotation, but this is not clear. Please state this

L116: I am incredibly confused by Figure 4, its in text reporting, and its description. Namely, you have contrasting results (!) between your conditional model and zero-inflated model (at least assuming that the coefficients are for the same response), but these are not presented or discussed. Also, what is presented in text does not match the figure, e.g., “Probability of excess zeros [where is this in the figure]... was significantly greater for grasslands (0.81..) [I do not see grassland coefficient values == 0.81] Moreover, what is meant exactly by conditional model? Please clarify this figure and present these results more clearly. Then, please discuss why you have such different responses and what this means for your inferences of diversification effects. Overall, I find the inclusion of the zero-inflated model confusing and its contrasting results worrying to the main conclusions drawn by the authors

L120: Unclear what “when treated with an average TFI...” means. Please clarify

L122: Similar to my last review, please be careful about how you describe your pesticide use response variable. Here you say “total pesticide use”, typically this means you have information about how much was applied over a specific area. But really what you are measuring is a frequency, no? Please be specific with you terms

L129-139: These results all talk about pesticide use in the 16 crops, however Figs 5 & 6 show effect for only wheat and winter wheat. Is this an error? If not, the the results text whould specify this crop and substantiate the focus on this crop

L145: Correct supplemental figure for this point? Please correct and make sure to review all figure references in the text

L148: Change “Analysis” to Analyses

L150-160: Again here you say “use”, Please specify ‘use frequency’. I understand its easy to be colloquial, but it might mislead those readers who don’t take the dive into you methods if it isn’t specified somewhere in the results what you mean by “use”

L162: Wow. Rather large reduction in insecticide use. Important finding

Discussion

L172-174: Yes! Good point and well said

L179-180: “...tended to be greater when local pest pressure for a given crop was higher”, but you didn’t measure pest pressure, correct? So how can you claim this? Please clarify

L189-190: I like your use of an example, but I don’t think its saying what you mean. If [pesticide reliance of cereals] < [pesticide reliance of sunflower], then adding sunflower would not dilute, but in

fact the opposite concentrate use (as in your example in L193-194). Unless I am mistaken or there is a typo somewhere, I don't think this example is doing what you want. Please adjust your example

L200: "appreciated" perhaps change to 'approximated'

L205-206: Again, perhaps I should wait for the methods, but in this new manuscript you are using "average number of crops species per botanical family" as your measure of crop species diversity right? I think this makes sense, but I still wish you just had the simple measure of crop species richness (as this is commonly used, as you say). Moreover, I wish that somewhere (probably last paragraph of the Intro) you explain in brief what each of these different diversity metrics captures. E.g., effective number of botanical families approx. *functional diversity*; average number of crops species per botanical family approx. *taxonomic diversity* etc.

L200-201: You could (should) cite recent work by Riccardo Bommarco here (and elsewhere):

Smith, M. E., Vico, G., Costa, A., Bowles, T., Gaudin, A. C., Hallin, S., ... & Bommarco, R. (2023). Increasing crop rotational diversity can enhance cereal yields. *Communications Earth & Environment*, 4(1), 89.

Schaak, H., Bommarco, R., Hansson, H., Kuns, B., & Nilsson, P. (2023). Long-term trends in functional crop diversity across Swedish farms. *Agriculture, Ecosystems & Environment*, 343, 108269.

And this you cite, but also relevant here:

Nilsson, P., Bommarco, R., Hansson, H., Kuns, B., & Schaak, H. (2022). Farm performance and input self-sufficiency increases with functional crop diversity on Swedish farms. *Ecological Economics*, 198, 107465.

L208-211: So it's a strong botanical family identity effect

L215: Perhaps misleading bc elsewhere you provide an example of oilseed and mustard (2 crops) having similar pests, but here you say that Brassicaceae has only one species. Also, is this true? There are no cabbages, kales, or tubers (e.g. Turnips are listed in Supp. Table 2) grown within the Delphy network of field?

L257-267: Well thought through justifications. Nice!

L296: reasonable caveats. Nice

L310: Didn't Nilsson focus on functional groups (which, admittedly are strongly related to botanical families). It raises the questions why you don't use the functional group diversity in the current work, similar to these other authors. It would adhere with previous work, and be a few less words to repeat if you define what you mean by functional diversity up front

L315-316: Although I love the phrase "bouquet of services", you do not really look at ecosystem services here, and thus it seems a strange final remark for this paper (please use bouquet of services elsewhere though, its lovely)

Methods

L331: "proportion of each crop remained stable" This seems so critical for you space-for-time assumption. Can you confirm this in any way or show us that this is true?

L334-335: I am still unclear about these data to some degree (Fig 1 is a vast improvement). What is meant by “cropping systems were described identically but on an annual basis”. Who described? What is being described? Is “described” the right word?

L339: I think the inclusion of safeguard is sensible. However, a lingering concern of mine with these data are the number of plots within a cropping system, as this will affect your diversity metrics. A safe approach would be to only sample from cropping systems with the same number of plots (or area), otherwise I still worry that there are sampling biases present from not accounting for sample coverage.

I think the issues raised in my previous comment go unaddressed:

“The authors’ indicators of temporal crop diversification are essentially diversity metrics, which are going to be influenced by these kinds of sampling effects. Is it safe to compare the number of species between 1m² and 10km²? Most ecologists would suggest one needs to account for differences in sample coverage. Yet it is unclear if the present work is comparing sampling units (i.e., cropping systems) that vary greatly in their spatial extent and number of sampling units (points 1 and 2 above, respectively). This is important because these differences would influence the number of crops present within and between years. If cropping systems are variable in their number, size, and management then this should be included in models as fixed effects.”

L339-342: This assumption (“working hypothesis”) is critical, and potentially erroneous and therefore disqualifying of your approach. You need to show that this is true, otherwise you are not truly measuring temporal crop diversity and rather just spatial crop diversity in different places/years. Just because in one year a cropping system has five different crops in 8 plots doesn’t mean it won’t be all winter wheat the next year, right? One way to approach this would be to subset your data and use those cropping systems that have more than one year of data and confirm that the identities of crops, *and their relative proportions*, are constant between years. Then, this space for time assumption needs to be caveated in the discussion.

Also, I would not say “over a given period” and rather “in a given period”. Finally, I do not like referring to these as “crop rotations” (L342), because it is not strictly a rotation in time. Please do not use this term and stick to cropping systems.

L371: average proportion within cropping systems?

L378: I think it is misleading to call these rotations (bc of your space for time assumption). I would call them cropping systems

L401-402: Based on your space for time substitution how do you know this? I would assume to know what was “preceded” would require truly temporal datasets

L426-: For these “sets” of models it would be helpful to refer to them by number(s) in Supp. Table 1

L443: not sure what this means or what to make of it. Please clarify or remove

Figures and tables

Figure 1: This is such an improvement! Brav! And it helps clarify some of my lingering concerns/questions about cropping systems and “crop rotation” (but see comments above). I think you could add more labels to the figure (you’re missing “B”) and use these labels to greater effect in

the text where you describe your responses and predictors. Overall, a great study design figure for a very complicated dataset.

Figure 2: I do not know the 'rule-of-thumb' method (sounds vague), is there a reference? In C and D why does the y-axis extend so high, it is obscuring (squishing) the distribution

Figure 3: This figure and Figure 2 could be moved to the supplement in my opinion. They are pretty descriptive and simply show the values range and tendencies of responses and predictors. In my mind, Nat. Comm. Papers do not usually have so many graphical items

Figure 4: see major comment above about how to interpret the two panels (conditional vs. 0-inflated) of this figure. I think given the strong contrasting results presented it warrants brief explanation in the the figure caption (and of course elsewhere)

Figure 5: please see above comment about the focus on winter wheat for these figures

Figure 6: ditto.

Supplemental figures and tables

Note both tables are listed as "Table 1"

Supp Table 2: In plant family column, why is "Divers" listed for taxa that have a single species, e.g., Basil. Also would "poaceae/fabaceae" be considered a unique botanical family? Correcting this should also change your diversity values, apologies but good to be accurate.

Reviewer #1

General comments:

The authors have done a great deal of work to take into account the many comments made by the various reviewers. The new version of the article is now quite different from the previous one. It concentrates on the analysis of one variable and the article presented is therefore more coherent and the analysis more thorough. The results are potentially interesting and new. I have reviewed the entire article again.

Thanks again for this thorough review and the time spent doing it. We highly appreciate it and are thankful for the quality of the comments. We provide a point-by-point answer below.

General comments	
My main comments concern the form of the document: Be careful to present your methods and analyses correctly before analyzing the results. The fact that the MM is at the end of the document doesn't help. I sometimes found it hard to understand what you were talking about or the specificity of the analyses (some of your results are quite technical). If this article is aimed at a broad audience, there is room for improvement, I think	Thanks for pointing this out. We agree that efforts can be made to increase readability for a wide audience. We believe to have addressed this point by:  - Rewriting the last paragraph of the introduction in order to provide elements on data handling, computation of indicators ... - Providing key elements on data analysis in the results section
Perhaps selecting/prioritizing the results? For me, Figure 4-5-6 X is the article's added value. The presentation of the basic data (or basic analysis) is essential (as mentioned in the previous review), but some could perhaps be put in an appendix, to provide more detail on the core of the article?).	You are right, better to focus on the core analyses and results. The presentation of the basic data and corresponding analysis were moved to supplementary material.
- The grammar and structure of some English sentences need to be reviewed.	The entire article was reviewed by a native English speaker. Some subtle changes were made to improve clarity or increase readability (i.e. some sentences were simplified).
Regarding the substance of the article: -Be clearer about the results of excess zeros, marginal effects, and average effects. This is in line with my previous comment, but I think that the results on excess zeros and marginal effects will appear obscure to many readers if they are not better explained	After discussion with a statistical consultant (Alain Zuur), we realized that zero-inflated tweedie was an overkill, as the tweedie family is already a good candidate for zero inflated continuous and positive data thanks to its extra dispersion parameter (as in the negative binomial for count data). We provided references to back up this point and re-ran all the models as such. Statistical procedures and corresponding results should now be more easily understood.
-Why didn't you analyze the effect of diversification on excess zeros?	Please see previous comment. Zeroes and non-zeroes are now not distinguished anymore as zero inflated tweedie was an initial overkill.
Present the overall results as well (not just the conditional effects and the results linked to excess zeros).	This is now the case as zero-inflation was not initially warranted with a tweedie distribution.
-How well do your models perform? Do they explain a large part of the variance in TFI? Unless	Calculating a global measure of model fit is riddled with complications in the case of GLMMs and no simple single number can be found. No recipe will have all of the properties of R^2 in the simple linear model

I'm mistaken, I haven't spotted a fine enough analysis of this question.	case. Moreover, the tweedie distribution is not handled by the different available functions in R (piecewiseSEM, MuMin...). As a sign of good faith, we included plots showing the relationships between predicted and observed values, and provided their squared correlations.
Title: Crop rotation diversification or 'only' crop diversification/ or crop rotation? crop rotation diversification seems strange to me ?	Due to a comment from another reviewer, which found the title too strong in light of the results, we changed it to "Fostering temporal crop diversification to reduce pesticide use". We believe that this title reflects more our current discussion, i.e. that crop diversification itself can contribute to pesticide reduction, but probably to a greater extent when farmers foster the opportunities associated to it (e.g. diversification of farming practices).
Abstract	
3761 crop rotation-> It's not easy to understand the differences in the text (globally) between 1334 (cropping system), 3761 (crop rotation) and 14556 (crop). Do you need to present these three levels of information? Which one do your analyses focus on?	Our core analysis on the effect of crop diversification on crop pesticide use focuses on the crop level. We now place emphasis on this level in the abstract and end of the introduction. Cropping system refers to a set of plots managed according to the same crop rotation, constraints, and decision rules. Instead of referring to crop rotation, we now specify cropping system a given year.
L8: 'increasing the average number of crops per botanical family'. On my first reading (without reading the rest of the text), this sentence didn't seem easy to understand.	The name of the indicators was simplified to crop taxonomic and functional diversity to increase readability. Upon introduction, we clearly specify how these were computed.
What exactly does 'effective number' mean? Does it refer to a particular concept? Why not just mention 'number of'? OK, I understood after reading the material and method. But as it's at the end of the document, I'd like to make it clearer/more comprehensible from here on?	We addressed this comment simultaneously with the previous one at the end of the introduction.
L11. positive effect'. Positive in what sense? An increase? or positive for the environment (a decrease).	"Positive effect on herbicide use" was replaced by "an increase in herbicide use". We avoided the use of positive/negative effect throughout the text.
Keywords. Crop rotation is missing, isn't it?	Crop rotation was added as a keyword. Other keywords were changed alongside.
Introduction	
L18. Reference 1. Reference in French in a journal with no impact factor (?) dating from 1991. Is this subject so little covered?	We found a more international reference which backs up the same idea. The reference was replaced by "Storkey, J., Bruce, T. J. A., McMillan, V. E. & Neve, P. in Agroecosystem Diversity; Reconciling Contemporary Agriculture and Environmental Quality (eds G Lemaire, P C F Carvalho, S Kronberg, & S Recous) Ch. 12, 199–209 (Academic Press, 2019)" We added a new and recent reference that backs up the fact the crop diversity has decreased over time (as warranted by Reviewer 3).
Sometimes there are spaces before the reference numbers, sometimes not. Should this be standardized?	Good catch. All the space before the reference numbers were removed.

L34. Combine or combined?	Good catch. "Combine" was replaced by "combined"
L41. Do these references show that there are few studies on the subject, or are they just a few studies that have looked at this subject?	We apologize for not understanding the nuance. Nevertheless, we rewrote this "knowledge gap" section to highlight the novelty.
L45. Replace and are hence by and hence are?	Good catch. We changed as suggested.
L46. "Scientist" appears twice. The text must have been re-edited a bit quickly for resubmission.	Good catch. Scientists was replaced by farmers, as was initially intended.
L47. Before ? Strange translation from French.	"before" can be positional or refer to time. However, we do agree that this use may be too formal for a wide audience so we replaced it by "when confronted to".
L55. Investigate or investigated?	Good catch again. "Investigate" was replaced by "investigated"
L57. Focal or main ? you sometimes change throughout the text. Homogenize.	Thanks for catching this. "Focal" was replaced by "main" throughout the text
L58. Plots-> so they are not fields?	"Plots" was replaced by "fields" throughout the text, as they are used in the exact same sense.
L58. Does this definition of cropping system correspond to that of Sébillotte et al?	Not precisely. The definition of Sébillotte states that farming practices are rigorously identical between plots of a given cropping system. According to Sébillotte, a cropping system is defined as a sequence of crops and set of farming practices associated to each one. Hence, if we take the definition literally: maize-plot 1-year 2020 would not belong to the same cropping system as maize-plot2-2021 if they did not receive the same herbicide. Here, the variability of farming practices between plots is appreciated by the farmer. Here cropping system is rather defined in terms of decision rules (I do this if I see this) and constraints (access to irrigation or not, till vs. no-till...). We hence added this information.
See my previous comment: it might be wise to simplify the presentation of the data. Do you really use this 'cropping system' level? Or at least start presenting the data that will be at the heart of your analyses?	The cropping system level is primarily used to explore relationships between crop diversity and the presence of certain crops. We agree that special emphasis should be placed on the "crop level", the one corresponding to the core analysis. We tried to highlight this level, rather than others, at the end of the introduction.
L61-62. I don't understand this sentence. Why put 'a given year'? I must have missed something.	This refers to space-for-time substitution. Temporal crop diversity can be appreciated each year based on the diversity of crops present across a set of plots managed according to the same rotation/constraints/decision rules. We tried to be as clear as possible at the end of the introduction. Figure 1 should also help understanding.
L64. Same question as above. Does "effective" have a particular meaning?	Yes, this refers to Hill's numbers. 4 effective crop species means 4 species representing each 25% of cropping system area. We tried to be clearer about this at the end of the introduction and specified that these indicators were computed using Hill's numbers.
L65. At this stage we don't know how the cover crop frequency is calculated. Should it be mentioned?	We now mention that this is the proportion of fields preceded by a cover crop a given year.
L66. Is the notion of TFI common, shared enough by the scientific community (and more)? Not sure. This could make it harder for a wider community to understand the paper. Explain and define what this indicator is and how it is calculated?	Pesticide reliance is commonly described by two indicators: the quantity of active ingredients applied per hectare or the number of applications at the full recommended dose. We believe the second to be more meaningful as for ex., one active ingredient at 25g/ha can have the same effect as another at 1.5kg/ha.

	In particular, TFI is used by the European Commission to measure pesticide reliance. https://www.oecd.org/greengrowth/sustainable-agriculture/1916629.pdf This indicator was used in numerous publications from our research group but also in other European studies, for ex. : https://www.sciencedirect.com/science/article/pii/S0261219419300262 https://www.sciencedirect.com/science/article/pii/S2352550922000616 https://www.sciencedirect.com/science/article/pii/S0167880904001495 https://www.sciencedirect.com/science/article/pii/S0048969723008537 And is also becoming more and more frequent internationally, for ex.: https://onlinelibrary.wiley.com/doi/full/10.1002/ps.5249 https://www.mdpi.com/1660-4601/15/2/204 We specified how this indicator was computed at the end of the introduction.
Results	
The presentation of the data is very important and provides a good understanding of the core of the analyses. My question is: for this type of journal with a large audience, should this data be presented first in the results section, or should the authors directly address the strong results of their article (and put this information as a material supplement)? I would have chosen the second option.	We agree. Description of the data was moved to supplementary figures to highlight core findings.
L83. Is $r=0.60$ moderate?	Always tricky as this appreciation may be highly depend on the field of study. We turned the sentence around to not specify low/moderate/high.
L88. Does it also refer to correlations for cover crops?	Initially it did not but cover crop frequency is now referred to as a temporal crop diversity indicator, so we included it in the correlation plot (and the text describing it).
L91. Title not easy to understand on first reading.	The section title has been changed as follows to make it easier to understand: "Relationships between climatic region, crops, and crop diversity indicators"
L92 "regression analysis" Which one?	We agree this is vague. We know precise elements on what statistical analyses were carried out specifically before reporting corresponding results.
L97. What does crop identity mean?	Stating "identity" probably raises more confusion than anything else. We now simply specify "crop" and turned the sentences around to be clear, , e.g. "to investigate the effect of crop on ...";" highlighted a significant of crop on..." and so on.
L97-99. Strange way of formulating this information?	The section was entirely written. Please also see previous comment.

L104. You put statistical details here, $t=97$, $df= \dots$, but not the other times, why?	We initially did this because all other correlations (those between diversity indicators) and their associated statistical details were included as a correlation plot in supplementary material. This specific information is now added to this supplementary material to be homogeneous.
L109. Strange wording?	The sentence was modified for greater clarity.
L111. In my opinion, this part is the heart of the article and the results are interesting (but need to be reworked, see my comments).	We agree. The main originality of the article is on the regulation effect. We moved the description of variables to supplementary material in order to not dilute core findings among descriptive elements. We nevertheless start off with the “dilution effect” as this section determines which crops are worth focusing on thereafter (we show that certain crops are rarely treated with pesticides so there is no specific interest in investigating the effect of temporal crop diversity on pesticide use in these crops).
L113 'Type II analysis of deviance'. Why use a type II here? Don't you take interactions into account?	This is a source of controversy and arouses strong emotions among some statisticians as type III sum of squares violate the principle of marginality. All analyses of deviance are now carried out with type III sum of squares and sum-to-zero contrasts.
L116 . Complicated to understand without having had any details of the statistical analyses you have carried out. The vast majority of readers won't understand what you mean. You should at least say that you have a distribution with a large number of 0s and values >0, and that you are analysing these two components independently?	Please see previous comment on zero-inflated tweedie being an overkill. Statistical elements were introduced in the presentation of the results in order for the reader to have an idea on the underlying analyses.
L116-120 So you are analysing the effect of each crop on the TFI. That's not the most interesting part, because your aim is to study the effect of diversifying crop rotations. The question that would interest me would be to see whether the effect of diversification is stable or, on the contrary, sensitive depending on the crop. The way in which you present this fragmented information does not make it possible to answer/analyse the answer to your main question.	Interactions between crop and diversity indicators were introduced in all models, except for cover crop frequency because we did not have any specific hypothesis concerning this matter. Furthermore, the models already contain numerous parameters.
L117. Strange to find grassland here. Could you specify that these are temporary grasslands, included in rotations to avoid confusion?	We agree this is a good thing to precise. Grassland was replaced by Temporary grassland throughout the text and in the Figures
L129. Your analysis between the two 'components' of the distribution (i.e. the zeros and the rest of the distribution) could be confusing. Have you performed a complete analysis of the factors associated to the excess zeros, then the factors affecting the level of TFI? Why do the	Please see previous comment on zero-inflated tweedie being an overkill. Zeroes and non-zeroes are not distinguished anymore.

diversification indicators effects on the excess of zero have not been analyzed (this is a strong hypothesis?) ? Why do you present only the conditional mean, and not the average effect? Unless, I am mistaken, your model could be used to estimate the mean TFI for a particular level of diversification (i.e. considering the probability of being 0 , and the probability >0). Would be particularly useful, if there is an effect of the diversification on the excess of zeros.	
I think there is some work to be done on presenting the method better and analysing the results (the most interesting ones) in a better structured way.	Elements concerning methods were introduced when presenting the results. Some results were moved to supplementary material to focus on the core findings in the text.
L130. Why an increase of 1 SD. It seems to me that it should be by one unit of the variable. There's something I haven't understood or that's badly explained here. It would be easier to interpret if you spoke in units that could be interpreted: for example, adding a species to a rotation results in XXXX. OK, your variables have been calculated using Hill's formula. Is it still possible to present the results more explicitly?	This is debatable. The advantage of presenting 1 SD is that the effect of different diversity indicators become comparable. Moreover, 1 unit of the variable depends on the scale on which it is expressed (e.g. 1 gram vs 1 kilogram). However, such comparisons are not made in the current version of the manuscript. Hence, we decided to report effect sizes when functional or taxonomic crop diversity increased from 1 (the minimum) to 4 (which corresponds roughly to the minimum + 4 SD for both indicators, which would cover most of the population in case of a gaussian distribution). For cover crops, we report effects when frequency increase from 0 to 1.
I think it will also be useful to present the performance of your model. A plot showing the predicted distributions of IFT versus the observed could help to judge the quality of the model, and identify whether there are risks of under/overestimation for certain values.	Completely agree and thanks for this alternative to the highly debated R^2 for GLMMs.
How did you select the best structure for our model? On the basis of AICs?	Yes comparisons were made based on AIC (this is now specified in M&M) and the selected model always presented a delta AIC>2 when compared with the second best performing model.
Discussion	
L168 "Negative effect of crop diversification"-> a negative effect means a reduction in IFT, right? Try formulating it differently, to avoid possible confusion. Would you rather talk about a decrease or increase in the use of pesticides? To be used throughout the text	Thanks for pointing this out. We now express all the results in terms of "reduced" / "increased" and completely avoid "negative/positive" effect.
L168 ' for all of the 16 main crops". Have you run a model for each crop to test this? Or tested the absence of crop*diversification indicator interactions?	We initially wanted to avoid this level of complexity but interactions are now introduced in all models (except with cover crop frequency, which is justified in a previous comment).
L176: "across all crops and climatic region" same remark.	Agreed. This was not supported by the data/analyses and hence removed.
L293 'maximal'?	The sentence was modified to avoid this word.

M&M	
L323. "Only non-organic" -> Conventional? Or are there other types of systems, a wide diversity included in the database?	We avoided this debatable classification and specified "only cropping systems which resorted to synthetic pesticides...". This better captures the diversity of cropping system types (conventional/business as usual, conservation agriculture, integrated weed management ...).
L356. 'Significant' ?	"Significant" was replaced by "high"
L346. Why not include weather variables directly in the model? That would certainly be more accurate?	The idea here is to control for certain large climatic effects, not to capture the maximum variability in pesticide use that can be explained by weather variables. The advantage of this approach is its ability to synthesize a large number of climatic variables (climate types were based on 14 climatic variables, which would drastically increase model complexity and potential collinearity). Furthermore, for these climatic variables to be meaningful, an interaction with crop would be warranted (summer maximum temperature is expected to have an effect on maize growth/pests but not winter wheat), which would again explode the number of parameters; this is currently captured in our models by random interactions between crop and climatic region or between crop, climatic region, and year.
L367. Of the city' ? do you mean commune/ municipality?	"City" was replaced by "municipality"
L371 'of all main crops'? why 'all'?	Thanks, indeed "all" was not required and hence removed from the sentence.
L1391. This sentence has no verb?	Good catch, better with a verb. The sentence was reworded correspondingly.
L395. Why use this indicator rather than the number of crops?	Unlike species richness, this indicator (crop diversity) takes into account the proportion of crops. The two indicators were however strongly correlated ($r^2=0.84$) and this is explicitly specified. Moreover, this was the indicator used in version 1 and we were criticized for its simplicity (lack of functional aspects).
L441. The structure of random models is very complex. Have you checked that all the terms are relevant?	Yes, all random effects were properly estimated (none were on the boundary). Code and script are available for anyone who wants to check it. Moreover, random effects can be specified based on the design/a priori knowledge, rather than parsimony. Including random effects estimated at 0 has no effect on model parameters. Nevertheless, thank you this comment as it is uncommon to propose to simplify models.
Figures	
L636. "therefore, crop rotation refers to the set of crops present within a given cropping system at a given time point'?" at a given time point? Figure2: the x bar is written in lower case in the legend and in upper case in the figure. Why are the axes of graphs C and D so wide when the data are not so spread out? Put a definition of TFI here for readers?	We know avoid the term "crop rotation" when referring to a given cropping system a given year. "at a given time point" reflects our space for time substitution approach: for a given cropping system (for example monitored from 2012 to 2015) we can assess crop diversity at multiple time points (e.g. 2012, 2013, 2014, 2015). We tried to be clear about this at the end of the introduction and hope that Figure 1 backs up this point. Letter cases were standardized.

	Graphs C and D appear so wide because of few high values (potatoes in case of graph C, and spring pea/oilseed rape in case of graph D), not easily visible. These graphs are now in supplementary material. A definition of TFI was introduced.
Figure 3. subplot A. 'effective number of botanical families'. Add number of crops per botanical....'. Same remark for the x bar.	Good catch, very thorough, thank you. This comment was taken into account.
Figure 4. I can only see 15 crops here, whereas you mention 16 in the text... have you forgotten or have the constrasts been badly defined? You haven't modelled the effect of diversification indicators on excesses of 0? why? Grassland-> Put temporary grassland to avoid possible confusion? Present the combined results rather than the conditional/zero inflated decomposition? L668. Delta method? What is it? Have you presented it? Improve the title ('estimate plot' -> ?)	Winter wheat was the intercept so the intercept + 15 = 16. Zero-inflated tweedie was initially an overkill so some of these comments are not relevant anymore. We now use “temporary grasslands” everywhere. Thank you for this proposition. Note that this graph is not present in the current version of the manuscript anymore. The Wald/delta method is a common way to compute confidence intervals. We introduced a reference.
Figure 5 You mentioned once in the text that the maximum number of crops per botanical gfamily is 4, but here on the graph the data go up to 8... Title: in winter wheat ? (match the title of Figure 6) Why only present the results for wheat? if I'm not mistaken, the models presented in table S1 show some interaction (at least for the intercept) between the diversification indicator and the crops. It might therefore be interesting to present the results for all the crops? (same question for Fig 6)	Yes, good catch. This discrepancy arises from the fact that 4 is the maximum in the sense that it represents mean + 2SD whereas 6 (and not 8) is an extreme value that was observed (very few data points between 4 and 6). Note that this graph is not present in the current version of the manuscript anymore. For precision, we initially showed only winter wheat (the most common crop) because interactions between crop and diversity indicators were not initially considered. Hence, the slope of diversity indicator was identical for all crops. Indeed, the slight deviations you observed were due to the random intercepts between crop and climatic region.
is it normal that the relationship is linear with that you have used a log link function and that your data is presented back-transformed?	The choice of a log link is based on a priori knowledge. For positive continuous data, it appears as a reasonable choice. The trends appeared as linear due to the weak slope effect over the x values chosen. Data is indeed back-transformed for presentation.

Reviewer #2

Thanks for your positive comments. We welcomed your comments concerning the agronomy behind the manuscript.

General comments	
I have carefully read the revised version of the manuscript which has been significantly improved by the authors. My additional comments are few and mainly focus on the agronomy behind the results obtained	Thanks a lot for your feedback. We believe you could appreciate the first and new paragraph of the discussion section, which clearly focuses on agronomy and cropping system structure.
My first comment focuses on the title, which written as an affirmation. It is true that the title is supported by the data. However, does the small level of pesticide use reduction (3.0%) when diversifying one crop species per botanical family warrant it?	The title was not originally meant to be a strong affirmation, rather a passive form of the question “can temporal crop diversification reduce pesticide use?” (Which is not allowed by Nat Com). It was meant to be understood as a general theme. However, we understand that other readers might apprehend it differently (as you did), and that a discrepancy might appear between the title and the relatively weak “regulation effects” on pesticide use reported in text, at least for certain crops. Please note that we now introduce an interaction between crop and diversity indicators, so the effects vary from “no effect” to “moderate” effects depending on the crop. Therefore, we changed the title to “fostering crop diversification to reduce pesticide use”. We believe that this title reflects more our discussion, i.e. that crop diversification itself can contribute to pesticide reduction, but probably to a greater extent when farmers foster the opportunities associated to it (e.g. diversification of farming practices).
I understand that farms participated in the Dephy network on a volunteer basis, possibly the most innovative and/or environmentally-concerned ones. Could have this biased the results obtained? In relation to this, important production areas such the “Landes” (important maize production area if I am not wrong) seem to be almost uncovered by the network. Which could be the consequences on the data and the results obtained?	Previous work has showed that the DEPHY network was representative of French crops and farming practices (thesis of Martin Lechenet, 2017). Nevertheless, the aim of this work is to highlight the potential effects of temporal crop diversification on pesticide use. To do this, a gradient of temporal crop diversity is warranted. Focusing solely on mainstream cropping systems would not have allowed to explore this gradient. We discuss the discrepancy between the relatively weak effects reported and the stronger ones reported in the literature. The weaker effects identified here are possibly due to the fact that within the network, crop diversification is not only implemented to reduce pesticide use, whereas crop diversification is generally implemented in long term studies with the specific aim of reducing pesticide use. Concerning the “Landes” department, you are right, it is the most intensive maize monocropping region in France and poorly represented in the network (not necessarily the farmers most motivated by pesticide reduction). Some cropping systems in the south of the “Landes” department are nevertheless captured (alongside similar maize monocropping systems in the Alsace region). If anything, the

Editorial Note: Screenshot below from Weisberger D, Nichols V, Liebman M (2019) Does diversifying crop rotations suppress weeds? A meta-analysis. PLOS ONE 14(7): e0219847. <https://doi.org/10.1371/journal.pone.0219847>

	poor representation of this region probably dampens the effect of temporal crop diversification on pesticide use in maize (because in these regions maize is monocropped and pesticide use the highest for maize).
L8-11. Please, report the magnitudes of the decrease or increase. Readers without an agronomic background can be misled by these statements.	Good point. Effect sizes were added in the abstract.
L15. Lacks a comma.	Good catch. Key words were modified and the comma added.
Discussion	
L172-173. Could the authors provide further information about what is considered by “simple rotation” and by “diversification” in the cited work? That will make the statement much more straightforward to the readers.	Good idea. This information was added in the text. For your interest, here is an extraction from the original paper : Data extraction and processing Data was extracted from text, figures (GetData graph digitizer, http://getdata-graph-digitizer.com/), and datasets acquired through personal communication when necessary. All data manipulation, analysis and graphics were completed with the R (version 3.5.1) [21] packages readxl [22] and tidyverse [23]. A complete description of the extraction process and resulting information is provided (S2 Text). Weed response variables were biomass (g m⁻²) or density (plants m⁻²). The resulting raw dataset (n = 891) is available through Iowa State University's DataShare [24]. Comparisons of simple versus diverse rotations were determined on a per-paper basis using the number of species present in the rotation as a guide. Within studies, this resulted in the following rotations identified as simple: (a) monocrop (1 species, n = 265), (b) monocrop with alternating years of fallow (1 species, n = 19), and (c) a two-year rotation (2 species grown over two years, n = 63). Continuously growing one crop or two crops in succession (a and c) are salient examples of simple rotations within contemporary agricultural production. Intermittent fallow systems (b) are a form of a simple rotation that is commonplace in arid regions worldwide [5, 25]. If more than one diverse rotation was present in the study, the simple rotation was compared to each diverse rotation separately. Our classification scheme resulted in a total of 64 comparisons for weed biomass and 247 for weed density. For each comparison, we used the extracted information to create nine factors that could potentially influence the effect of crop diversification on weed dynamics. These included seven categorical variables: (a) latitude class (temperate vs. sub-tropical and tropical), (b) tillage regime (tilled vs. zero-tillage), (c) use of herbicides (yes vs. no), (d) use of fallowing in the simple rotation (yes vs. no), (e) use of a perennial in the diverse rotation (yes vs. no), (f) diversifying from a monoculture (yes vs. no), and (g) the unit of weed measurement (single species vs. sum of multiple species). We also assessed the influence of two continuous variables: the ratio of the number of species in the diverse rotation to the simple rotation and the difference in the coefficient-of-variation of months between planting operations in the diverse versus the simple rotation [26].
L181. In current times the term “industrial cropping systems” is often seen as pejorative by mass media and consumers. I would advocate changing the term by “... in cropping systems including industrial crops”. In my opinion this last is more informative and less prone to misunderstandings.	Completely agree. We hadn't thought about this aspect. Nevertheless, this entire section was rewritten and the current version does not contain such wording. We checked the rest of the document to see if this wording was used elsewhere but it was not the case.
L195-197. It is worth mentioning that introduction of forages entails a different market, machinery and/or livestock integration.	Completely agree. We cannot leave a wide audience believing that a grain farmer can introduce all crops possible without changing anything else. We added this information.
L198-202. Could this unexpected result be –partially– explained by the rather short duration of data acquisition by the Dephy framework. For weeds someone would expect crop rotation effects on the seedbank in the long-term. That is, after a number of complete rotation cycles.	Although slight changes can be made over the years, the cropping systems monitored remain relatively stable in time (see new section on stability of crop composition in time). These cropping systems are not initiated when the farmers enter the network. They are implemented routinely. We now discuss why this may be (e.g. because sowing periods can be relatively constrained in certain geographical areas, because certain crops with similar sowing periods can have key traits related to weed management – such as triticale/ryegrass within the winter grasses...).

L216-218. I do not totally agree with this statement. The link between botanical families and machinery is not that straightforward. In my opinion, machinery is mainly determined by the use of the crop and/or its architecture. See, the different types of seeders (or planters) and harvesters (or mowers) used, for instance, in ryegrass, winter cereals for grain and maize cultivation.	Agreed, maize and winter wheat entail nothing similar. However, in light of the new results, this section did not appear warranted anymore and was completely rewritten.
L283-287. I am not totally convinced about the role played on weed control by different row distances: the species that we control in the spring or beginning of the summer in row crops by hoeing are different from the ones that are controlled in winter crops. In my opinion the main cultural control here is related to the different tillage periods (e.g. introducing a summer crop delays tillage and allows to mechanically destroy a winter weed before seed production).	Yes. We rewrote this section to be clear.
L300. Typo	Good catch. The sentence was corrected “observe->observed”, “design”-> “specifically implemented”
Supplementary M&M	
Supplementary Table 1. A typo (spring oat).	Good catch. This was corrected.
Supplementary Figure 7. A typo (potato).	Good catch. However, this figure does not exist anymore.

Reviewer 3

here. Great job in the revision. It is a different manuscript, but a better one. I still think work needs to be done. As before, comments of most concern are in bold

Thanks a lot for your thorough review and the time spent on the article. We believe that these comments improved the manuscript once again, and are grateful for that.

Introduction	
L18: Sentence unclear. Not sure what is meant by keystone. Maybe just say something like, “synth pesticide use characterizes simplified cropping systems”. Also simplified crop rotations is somewhat oxymoronic, no? I assume systems where crops are rotated are not simple.	This sentence was rewritten and we avoided “simple crop rotations”. Nevertheless, we do not believe this term to be oxymoronic. This is used in the literature, for ex. Weisberger, 2019 : Does diversifying crop rotations suppress weeds? A meta-analysis : Data extraction and processing Data was extracted from text, figures (GetData graph digitizer, http://getdata-graph-digitizer.com/), and datasets acquired through personal communication when necessary. All data manipulation, analysis and graphics were completed with the R (version 3.5.1) [21] packages readxl [22] and tidyverse [23]. A complete description of the extraction process and resulting information is provided (S2 Text). Weed response variables were biomass (g m⁻²) or density (plants m⁻²). The resulting raw dataset (n = 891) is available through Iowa State University’s DataShare [24]. Comparisons of simple versus diverse rotations were determined on a per-paper basis using the number of species present in the rotation as a guide. Within studies, this resulted in the following rotations identified as simple: (a) monocrop (1 species, n = 265), (b) monocrop with alternating years of fallow (1 species, n = 19), and (c) a two-year rotation (2 species grown over two years, n = 63). Continuously growing one crop or two crops in succession (a and c) are salient examples of simple rotations within contemporary agricultural production. Intermittent fallow systems (b) are a form of a simple rotation that is commonplace in arid regions worldwide [5, 25]. If more than one diverse rotation was present in the study, the simple rotation was compared to each diverse rotation separately. Our classification scheme resulted in a total of 64 comparisons for weed biomass and 247 for weed density. For each comparison, we used the extracted information to create nine factors that could potentially influence the effect of crop diversification on weed dynamics. These included seven categorical variables: (a) latitude class (temperate vs. sub-tropical and tropical), (b) tillage regime (tilled vs. zero-tillage), (c) use of herbicides (yes vs. no), (d) use of fallowing in the simple rotation (yes vs. no), (e) use of a perennial in the diverse rotation (yes vs. no), (f) diversifying from a monoculture (yes vs. no), and (g) the unit of weed measurement (single species vs. sum of multiple species). We also assessed the influence of two continuous variables: the ratio of the number of species in the diverse rotation to the simple rotation and the difference in the coefficient-of-variation of months between planting operations in the diverse versus the simple rotation [26]. For example, common crop rotations in France include winter wheat/sunflower, maize/ryegrass, maize/winter wheat, which are all considered as “simple”.
L24: “Crop diversification”. Define what this is and note here that it can be both spatial and temporal, like “Crop diversification – increasing the heterogeneity of crop cultivars, species or families in space and time – is a fundamental...” Or something like this	Good idea. This was introduced.
L26: no need for ‘diversified’, its implied	Agreed. This was removed.
L26-37: Excellent background information, well done	Thanks!
L40-41: your pivot statement (“...but only a few studies	The knowledge gap section was rewritten to account for these comments.

investigated the impact of complex landscapes on pesticide use) is true, but I think it could be more specific. Two reasons. First, the “many” spatial crop diversification studies you identify are in fact studying varying degrees of “complex landscapes”. Second, I think you should be clear and stick to the specific focus of this article: temporal crop diversification (TCD)	
L49: I like your use of “commercial” before, thus you could say the more succinct “... as currently implemented in commercial production systems”	Thanks, this was taken into account.
L49: “allows to reach” change to “can achieve”	This was taken into account. Thanks for the proposition.
L52-54: Great points. I quite like this dilution vs regulation framing. Note, L55 could say “needs investigation”	We tried to be clear about what we mean when we introduce these two different effects.
L56: Is the aim really to quant how use “could be reduced” (this sounds more like a modeling objective). I would state this rather as “whether X reduces Y”. Please revise this objective statement to be more clear	This comment was taken into account.
L63: Again, be consistent with terms: Crop rotation diversity vs temporal crop diversity. Is there a difference? If so, you have not made it clear. If there isn’t a difference, then pick one term	We now use temporal crop diversification everywhere throughout the text.
L59-68: This is useful methods information and needed given the format of Nat. Comm. However, I miss a statement of your main research questions or hypotheses. For example, do you have expectations about	We now state that we expect temporal crop diversification to reduce pesticide use through both dilution and regulation effects. This whole paragraph was actually rewritten to be more thorough on the methods.

the effect of TCD on dilution vs regulation? I am not asking you to post hoc create hypotheses, but rather state either questions or substantiated expectations	
L59-62: these study design details are still unclear (esp. L:61-62), but I am going to wait to read the methods before commenting.	This was rephrased to be as clear as possible.
Results	
L72: Without reading the methods TFI is a little abstract. Can you provide units? Or at least a brief definition of what an average of 3.7 indicates?	We understand. We now state at the end of the introduction: “Pesticide use (total and per pesticide type) was assessed at the crop level by the number of applications at the full recommended dose (i.e. treatment frequency index)”
L98: What is meant by “Their product”. More importantly, do you mean the actual values of these two variables multiplied together (which is what the Nilsson method does) or their interaction (Which is what it looks like in Table S1). Make sure you are describing (and doing) what you mean to. The figure caption text (and table S1) makes me think you looked at their interaction, but this is different than looking at their product as Nilsson and colleagues do. Please clarify	In Nilsson et al., overall diversity is computed as $H' = H_{\text{bota_family}} + H_{\text{intra_family}}$ Expressed in terms of Hill’s numbers ($q=1$), this yields : $\exp(H') = \exp(H_{\text{bota_family}} + H_{\text{intra_family}})$ And this can expand to : $\exp(H') = \exp(H_{\text{bota_family}}) \times \exp(H_{\text{intra_family}})$ with $\exp(H_{\text{bota_family}}) \times \exp(H_{\text{intra_family}})$ being the effect modelled by our interaction. Nevertheless, we specify “interaction” as this is done within the modelling procedure rather than a priori through a product.
L99-103: I assume that these association with specific crops are for when these crops are included in the rotation, but this is not clear. Please state this	The three sentences (or four considering the abstract) formulated this way were rewritten.
L116: I am incredibly confused by Figure 4, its in text reporting, and its description. Namely, you have contrasting results (!) between your conditional model and zero-inflated model (at least assuming that	This figure is not present in the article due to the fact that zero-inflated tweedie was considered an overkill. However, please note that the previous results were not contradictory, a negative effect on non-zeroes (conditional model) implies a reduction in pesticide use where a positive effect on zeroes (zero-inflated model) implies an increase in the probability of pesticides not being applied. We see on the following graph that these go hand-in-hand for most crops (i.e. when

the coefficients are for the same response), but these are not presented or discussed. Also, what is presented in text does not match the figure, e.g., “Probability of excess zeros [where is this in the figure]... was significantly greater for grasslands (0.81..) [I do not see grassland coefficient values == 0.81] Moreover, what is meant exactly by conditional model? Please clarify this figure and present these results more clearly. Then, please discuss why you have such different responses and what this means for your inferences of diversification effects. Overall, I find the inclusion of the zero-inflated model confusing and its contrasting results worrying to the main conclusions drawn by the authors

a given crop reduces pesticide use in the conditional model compared to another crop, it also generally increases the probability of excess zeroes in the zero-inflation model):

The discrepancy between the coefficients originally reported in the text and the ones in the figure originates from the scale (ln() for the conditional model and logit() for the zero-inflation model in the dot and whiskers plot, whereas probability scale in text).

L120: Unclear what “when treated with an average TFI...” means. Please clarify

This comment does not appear relevant anymore as it referred to the conditional model of a zero-inflated tweedie, which was considered an overkill.

L122: Similar to my last review, please be careful about how you describe your pesticide use response variable. Here you say “total pesticide use”, typically this means you have information about how much was applied over a specific area. But really what you are measuring is a frequency, no? Please be specific with you terms

We have all information to compute total pesticide use (quantity applied with respect to the reference dose and the surface of the plot on which it was applied). This can be seen in the formula reported in M&M:

Treatment Frequency Index (TFI)

Reliance on pesticide use was quantified through the Treatment Frequency Index (TFI). The TFI quantifies the mean number of treatments per hectare, for a given crop and year with commercial products (that possibly contains several active ingredients), weighted by the ratio of the dose used to the reference dose⁴⁶:

$$TFI = \sum_{i=1}^n \frac{D_i \cdot S_i}{D_{h_i} \cdot S_c} \quad | \quad (3)$$

where D_i , D_{h_i} , and S_i are, respectively, the applied dose, the reference dose, and the treated surface area of the **plotfield** for the spraying operation i ; and S_c is the total **plotfield** surface. The applied doses of commercial products were reported by farmers for each **plotfield** and each year. Each commercial

Hence, the indicator (although named treatment frequency index) does not reflect a simple frequency.

Let’s imagine an application of glyphosate at 1.5kg/ha on the whole field (and a reference dose at 3kg/ha), this yields a TFI of 0.5. If the same application was made on half the field, then this yield a TFI of 0.25. If this was simple frequency, then all these applications would yield the same value.

	Moreover, frequency would not capture what we aim to capture. For example, 2 applications at 0.75kg/ha would yield a greater value than 1 application at 1.5kg/ha, while TFI yields the same value. We precise at the end of the introduction that pesticide use was assessed by the number of applications at the full recommended dose.
L129-139: These results all talk about pesticide use in the 16 crops, however Figs 5 & 6 show effect for only wheat and winter wheat. Is this an error? If not, the the results text whould specify this crop and substantiate the focus on this crop	This graph is not present in the new version of the manuscript. For precision, we initially showed only winter wheat (the most common crop) because interactions between crop and diversity indicators were not initially considered. Hence, the slope of diversity indicator was identical for all crops. Now interactions are included and focusing on a given crop is not warranted anymore.
L145: Correct supplemental figure for this point? Please correct and make sure to review all figure references in the text	Yes, but possibly not the best one to highlight these effects. In any case, this figure is not present anymore. We checked that the correspondence between all figure references in the text and figure elements were correct.
L148: Change “Analysis” to Analyses	Good catch. We checked that plural form was used correctly and vice versa throughout the text.
L150-160: Again here you say “use”, Please specify ‘use frequency’. I understand its easy to be colloquial, but it might mislead those readers who don’t take the dive into you methods if it isn’t specified somewhere in the results what you mean by “use”	Please see previous comment (4 comments up). We kept “use” as “frequency” would be truly incorrect. We now precise at the end of the introduction that pesticide use was quantified through the number of applications at the full recommended dose (treatment frequency index).
L162: Wow. Rather large reduction in insecticide use. Important finding	We believe the new results to be even more impactful as they focus on crops where insecticides are commonly used (oilseed rape, spring pea) and the effects are even larger.
Discussion	
L172-174: Yes! Good point and well said	Thank you. We also believe this to be an important point. Furthermore, we now precise what is implied by “Weisberger et al. (2019)” when they state “simple rotations”.
L179-180: “...tended to be greater when local pest pressure for a given crop was higher”, but you didn’t measure pest pressure, correct? So how can you claim this? Please clarify	We agree. This was simply based on a visual observation of random effects combined with expert knowledge on pest pressure in different crops in different climatic regions. However, we agree that these elements are not sufficient to state this so we removed it.
L189-190: I like your use of an example, but I don’t think its saying what you mean. If [pesticide reliance of cereals]	We believe the sentence to be correct. We specify the opposite: pesticide reliance of cereals > than that of soybean or sunflower, so including sunflower or soybean in a rotation essentially composed of cereals, does indeed dilute pesticide use. Here is the corresponding text:

< [pesticide reliance of sunflower], then adding sunflower would not dilute, but in fact the opposite concentrate use (as in your example in L193-194). Unless I am mistaken or there is a typo somewhere, I don't think this example is doing what you want. Please adjust your example	Crops associated to livestock (feed crops such as grasslands, ryegrass, alfalfa, and cereal-legume mixtures) showed the lowest level of pesticide reliance, followed by sunflower and soybean, whereas dominant (winter cereals, oilseed rape) and industrial crops (potatoes, sugar beet) showed an intermediate to high level of pesticide reliance, respectively.
L200: "appreciated" perhaps change to 'approximated'	This part of the manuscript was completely rewritten but we avoided the use of "appreciated" altogether (we rather use "assessed").
L205-206: Again, perhaps I should wait for the methods, but in this new manuscript you are using "average number of crops species per botanical family" as your measure of crop species diversity right? I think this makes sense, but I still wish you just had the simple measure of crop species richness (as this is commonly used, as you say). Moreover, I wish that somewhere (probably last paragraph of the Intro) you explain in brief what each of these different diversity metrics captures. E.g., effective number of botanical families approx. functional diversity; average number of crops species per botanical family approx. taxonomic diversity etc.	We tried to find a happy middle with this comment and the previous feedbacks made by other reviewers. We initially used species richness in the first version of the article but this was criticized due to its simplicity and lack of functionality. Hence, changing back to species richness would imply overlooking these comments. Nevertheless, we now provide all correlations between crop functional diversity, crop taxonomic diversity, crop diversity, diversity of sowing periods, cover crop frequency AND crop species richness and show that crop species richness is highly correlated to functional crop diversity (r=0.67) and crop diversity (r=0.89). Diversity indicators are now presented at the end of the introduction (and their name simplified to taxonomic and functional crop diversity as you propose – Thanks for this idea).
L200-201: You could (should) cite recent work by Riccardo Bommarco here (and elsewhere): Smith, M. E., Vico, G., Costa, A., Bowles, T., Gaudin, A. C., Hallin, S., ... & Bommarco, R. (2023). Increasing crop rotational diversity can enhance cereal yields. Communications Earth & Environment, 4(1), 89. Schaak, H., Bommarco, R., Hansson, H., Kuns, B., &	These citations are now included (Smith when discussing trade-offs with other performance indicators and Schaak in the introduction to back up the fact that crop diversity has decreased over time). The bit of text where you propose to cite Nilsson again has been greatly modified and the citation would now appear out of context.

Nilsson, P. (2023). Long-term trends in functional crop diversity across Swedish farms. Agriculture, Ecosystems & Environment, 343,108269. And this you cite, but also relevant here: Nilsson, P., Bommarco, R., Hansson, H., Kuns, B., & Schaak, H. (2022). Farm performance and input self-sufficiency increases with functional crop diversity on Swedish farms. Ecological Economics, 198, 107465.	
L208-211: So it's a strong botanical family identity effect	Yes taxonomic crop diversity is largely correlated to the diversity of cereal crops. We now dedicate a large paragraph in the discussion that explains why temporal crop functional diversity or taxonomic diversity have effects on different crops. We relate this to the structure of cropping systems in which these crops are included.
L215: Perhaps misleading bc elsewhere you provide an example of oilseed and mustard (2 crops) having similar pests, but here you say that Brassicaceae has only one species. Also, is this true? There are no cabbages, kales, or tubers (e.g. Turnips are listed in Supp. Table 2) grown within the Delphy network of field?	We understand that this sentence can be misleading because in one case we refer to all crops of the dataset and in the other we simply focus on the top 16. However, this bit of text was completely rewritten and this sentence does not appear anymore.
L257-267: Well thought through justifications. Nice!	I hope you appreciate the new ones just as much! This was completely rewritten in light of the new results.
L296: reasonable caveats. Nice	Thanks! This was maintained.
L310: Didn't Nilsson focus on functional groups (which, admittedly are strongly related to botanical families). It raises the questions why you don't use the functional group diversity in the current work, similar to these other authors. It would adhere with previous work, and be a few less words to repeat if you define what you mean by functional diversity up front	We had an in-depth look at Nilsson's classification of functional groups. First, as you highlight, these are highly based on botanical families, so highly similar to ours. Moreover, we capture another element through another complementary indicator (diversity of sowing periods). Second, their classification also focuses on end-use, which has no ecological meaning in our case (certain herbaceous grasses end up in the case category as trees). From our standpoint, crops belonging to the same botanical family tend to be more similar than crops belonging to the same end-use category. It appears logical that functional groups can be defined differently depending on the aim of the study. Nevertheless, our classification is nearly identical to that of Nilsson's, especially for the 16 main crops considered in detail (which represent 96.5% of total crop proportion). We now introduce diversity indicators at the end of the introduction.

L315-316: Although I love the phrase “bouquet of services”, you do not really look at ecosystem services here, and thus it seems a strange final remark for this paper (please use bouquet of services elsewhere though, its lovely)	True. We replaced by “cropping system multi-performance”.
L331: “proportion of each crop remained stable” This seems so critical for you space-for-time assumption. Can you confirm this in any way or show us that this is true?	We agree that this needs to be tested. We now use PERMANOVA on the subset of cropping systems monitored in time to quantify to what extent crop composition was stable over time (and hence why temporal crop diversity can be assessed a given year by the diversity of crops present across the set of fields belonging to a given cropping system).
M&M	
L334-335: I am still unclear about these data to seem degree (Fig 1 is a vast improvement). What is meant by “cropping systems were described identically but on an annual basis”. Who described? What is being described? Is “described” the right word?	Upon entry in the network, farmers fill out a questionnaire spanning the last three years. We specify that, in the following years, farmers describe on an annual basis the crops grown on the different fields belonging to the same cropping system, alongside the farming practices associated to each.
L339: I think the inclusion of safeguard is sensible. However, a lingering concern of mine with these data are the number of plots within a cropping system, as this will affect your diversity metrics. A safe approach would be to only sample from cropping systems with the same number of plots (or area), otherwise I still worry that there are sampling biases present from not accounting for sample coverage. I think the issues raised in my previous comment go unaddressed: “The authors’ indicators of temporal crop diversification are essentially diversity metrics, which are going to be influenced by these kinds of sampling effects. Is it safe to compare the number of species	The number of fields monitored for a given cropping system is variable but not random. When a farmer wishes to enter the network, the network monitor asks the farmer to pick a set of fields which are representative of their cropping system. Therefore, diversified cropping systems require more fields to be monitored than maize monocropping. Public money would be partially wasted if network monitors chose the same number of fields for all cropping systems. For this to be especially clear, we added “Diversity indicators were not rarefied based on the number of fields monitored because the number of fields monitored was chosen by the farmer to reflect cropping system composition with the lowest number of fields possible in order to limit monitoring time (i.e. unnecessary to monitor 20 fields for maize monocropping).

between 1m² and 10km²? Most ecologists would suggest one needs to account for differences in sample coverage. Yet it is unclear if the present work is comparing sampling units (i.e., cropping systems) that vary greatly in their spatial extent and number of sampling units (points 1 and 2 above, respectively). This is important because these differences would influence the number of crops present within and between years. If cropping systems are variable in their number, size, and management then this should be included in models as fixed effects.”	
L339-342: This assumption (“working hypothesis”) is critical, and potentially erroneous and therefore disqualifying of your approach. You need to show that this is true, otherwise you are not truly measuring temporal crop diversity and rather just spatial crop diversity in different places/years. Just because in one year a cropping system has five different crops in 8 plots doesn’t mean it won’t be all winter wheat the next year, right? One way to approach this would be to subset your data and use those cropping systems that have more than one year of data and confirm that the identities of crops, and their relative proportions, are constant between years. Then, this space for time assumption needs to be caveated in the discussion. Also, I would not say “over a given period” and rather “in a given period”. Finally, I do	We have not come across the case where one farmer growing three crops would grow a different one each year over their whole farm. This could be possible if the plots monitored were randomly chosen on a given farm but this is not the case (please see previous comment: the plots monitored aim to reflect cropping system composition, which ones and how many are required is decided by the farmer). Nevertheless, we agree that this is a strong working hypothesis and that it would be a pity not to test it, especially given the fact that part of the data allows it (i.e. cropping systems monitored for more than one year). Hence, we now introduce a new PERMANOVA analysis which tests the stability of crop composition for a given cropping system over time. Results show high support for our working hypothesis: cropping system identifier explains 83% of the variance in crop composition over time. We acknowledge that true temporal datasets are required in the discussion: “Furthermore, the effect of temporal diversification was here assessed through time for space substitution (with caution that all fields followed the same crop rotation within a given cropping system). Future studies focusing on crop order and crop return delay (i.e. temporal datasets) could shed light on specific crop sequences (i.e. preceding crop – crop, rotation) and their characteristics that enhance pest, weed and disease control, and lower pesticide use.” Both “over a given period” and “in a given period” are grammatically correct and based on the following link, we believe “over a given period” to be more accurate as it distributes the event across time: https://ell.stackexchange.com/questions/206311/over-or-in-the-period-periods-given#:~:text=Both%20can%20be%20correct%2C%20but,to%20happen%20all%20at%20once. We now avoid the term “crop rotation” to refer to the set of crops present (at a given point over the set of fields monitored), as we agree that this is misleading.

not like referring to these as “crop rotations” (L342), because it is not strictly a rotation in time. Please do not use this term and stick to cropping systems.	
L371: average proportion within cropping systems?	It was changed accordingly
L378: I think it is misleading to call these rotations (bc of your space for time assumption). I would call them cropping systems	Crop rotation was replaced by cropping system throughout the text
L401-402: Based on your space for time substitution how do you know this? I would assume to know what was “preceded” would require truly temporal datasets	For a given field a given year, the database specifies all farming practices from harvest of the previous crop to harvest of the current crop. Therefore, this includes sowing of a cover crop. Cases where the cover crop was sown in the previous crop are also flagged in the database. Please note that farming practices are specified by the farmer (and hence encompass time), not visually by the network monitor at a given time point.
L426-: For these “sets” of models it would be helpful to refer to them by number(s) in Supp. Table 1	Very good idea. We now number the models (Supplementary Table 1) and refer to these numbers when relevant.
L443: not sure what this means or what to make of it. Please clarify or remove	This was removed.
Figure and tables	
Figure 1: This is such an improvement! Brav! An it helps clarify some of my lingering concerns/questions about cropping systems and “crop rotation” (but see comments above). I think you could add more labels to the figure (you’re missing “B”) and use these labels to greater effect in the text where you describe your responses and predictors. Overall, a great study design figure for a very complicated dataset.	We also believe this clarifies network description and dataset handling. Thank you for pointing us down this line.
Figure2: I do not know the ‘rule-of-thumb’ method (sounds vague), is there a reference? In C and D why does the y-axis extend so high, it is obscuring (squishing) the distribution	This is Silverman’s rule-of-thumb, a method commonly used for kernel bandwidth estimation. From Wiki:

A rule-of-thumb bandwidth estimator [edit]

If Gaussian basis functions are used to approximate univariate data, and the underlying density being estimated is Gaussian, the optimal choice for h (that is, the bandwidth that minimises the mean integrated squared error) is:^[23]

$$h = \left(\frac{4\hat{\sigma}^5}{3n} \right)^{\frac{1}{5}} \approx 1.06 \hat{\sigma} n^{-1/5},$$

An h value is considered more robust when it improves the fit for long-tailed and skewed distributions or for bimodal mixture distributions. This is often done empirically by replacing the standard deviation $\hat{\sigma}$ by the parameter A below:

$$A = \min \left(\hat{\sigma}, \frac{IQR}{1.34} \right) \text{ where IQR is the interquartile range.}$$

Another modification that will improve the model is to reduce the factor from 1.06 to 0.9. Then the final formula would be:

$$h = 0.9 \min \left(\hat{\sigma}, \frac{IQR}{1.34} \right) n^{-\frac{1}{5}}$$

where n is the sample size.

This approximation is termed the *normal distribution approximation*, *Gaussian approximation*, or *Silverman's rule of thumb*.^[23] While this rule of thumb is easy to compute, it should be used with caution as it can yield widely inaccurate estimates when the density is not close to being normal. For example, when estimating the bimodal Gaussian mixture model

$$\frac{1}{2\sqrt{2\pi}} e^{-\frac{1}{2}(\pi-10)^2} + \frac{1}{2\sqrt{2\pi}} e^{-\frac{1}{2}(\pi+10)^2}$$

from a sample of 200 points, the figure on the right shows the true density and two kernel density estimates — one using the rule-of-thumb bandwidth, and the other using a solve-the-equation bandwidth.^[17] The estimate based on the rule-of-thumb bandwidth is significantly oversmoothed.

We added “Silverman’s” and the corresponding reference.

In figure C and D, the y-axis cannot be reduced because of a spike close to zero (which appears as a thin line):

Please note that this figure is now in Supplementary Material.

Completely agree. We moved these two figures to Supplementary Material.

Figure 3: This figure and Figure 2 could be moved to the supplement in my opinion. They are pretty descriptive and simply show the values range and tendencies of responses and predictors. In my mind, Nat. Comm. Papers do not usually have so many graphical items

Figure 4: see major comment above about how to interpret the two panels (conditional vs. 0-inflated) of this figure. I think given the strong contrasting results presented it warrants brief explanation in the figure caption (and of course elsewhere)

Please see previous comments. There was no clearly contrasting results between the conditional and zero-inflation model initially (a positive effect on zero probability reduces pesticide use and so does a negative effect in the conditional model). Moreover, this figure is not present in the manuscript anymore as zero-inflated tweedie was an overkill: the tweedie family is already a good candidate for zero-inflated continuous and positive data.

Figure 5: please see above comment about the focus on winter wheat for these figures

This was due to the fact that interactions between diversity indicators and crops were not initially considered. Slopes for diversity indicators were hence the same for all crops (and we did not want to repeat 16 graphs with the same slope). Nevertheless, this is not the case anymore and no graphs simply focus on one crop.

Figure 6: ditto.	Duly noted. Please see previous comment.
Supplementary M&M	
Note both tables are listed as "Table 1"	Good catch. This was corrected.
Supp Table 2: In plant family column, why is "Divers" listed for taxa that have a single species, e.g., Basil. Also would "poaceae/fabaceae" be considered a unique botanical family? Correcting this should also change your diversity values, apologies but good to be accurate.	Good catch. This was modified accordingly. Intercrops were essentially represented by cereal-legume mixtures in our datasets. Due to synergistic effects between these two components (e.g. complementarity), we could not (or did not wish to) consider these mixtures as a simple sum of their components. Moreover, we did not have information concerning the final proportion of cereals and legumes in the mixture, making it difficult to assign a proportion to these two categories. Finally, please note that cereal-legume mixtures (the most common type of annual mixture in the dataset) only represent 2.5% of total crop proportion.

REVIEWERS' COMMENTS

Reviewer #1 (Remarks to the Author):

Thank you to the authors for their transparency efforts in providing code for the analysis and databases!

Once again, the authors have put in a lot of work into revising the manuscript based on the comments from various reviewers. As a result, both the text and the analyses have evolved significantly since the last version. The text is now simpler, clearer, and more convincing. The results are well-described, and the statistical analyses behind them appear convincing to me. I no longer have any obstacles to the publication of this paper. This series of exchanges with the authors has been interesting and enriching.

I have carefully read the responses point by point, and the authors' responses seem relevant and well-argued.

However, I do have several comments and questions that I would like to list:

1. Reference 1 is about Sweden. Should you find a more global reference? Is the idea behind your sentence to make a general observation?
2. Line 83. This is the first time the term "through crop composition" appears. What exactly does it represent? A number of species? Diversity?
3. You used PERMANOVA because it does not assume homoscedasticity and is more suitable for non-normally distributed data? Is this the case for your data?
4. Line 94. A double space?
5. "Crop functional diversity was higher in the Deteriorated Oceanic (2.19)". Is 2.19 the mean, I assume? Or the median? Should confidence intervals be included?
6. When you say, "To investigate the effect of crops on crop taxonomic and functional diversity, and their interaction (i.e. crop diversity)," please be careful. It makes me think directly of an interaction that you introduce into the model (i.e., as an explanatory variable). However, if I'm not mistaken, you are referring to the 'global' effect of crop diversity (i.e., as a response variable).
7. The crop taxonomic and other indicators are studied at the system level. Please clarify.
8. In line 91, why did you perform these models:
"(To investigate the effect of climatic region on crop taxonomic and functional diversity, and their interaction (i.e. crop diversity), one generalized linear mixed effect model (GLMM) with climatic region as the only fixed effect was fit for each of the three response variables," when you also conducted these models:
"To investigate the effect of crops on crop taxonomic and functional diversity, and their interaction (i.e. crop diversity), one GLMM with climatic region and crop as fixed effects was fit for each of the three response variables."
If I understand correctly, the climatic variable is present in both models. Could you analyze the effect of climatic variables in the model described in line 106? There is likely no independence of data, i.e., crops are not evenly distributed. (This might be the reason you decided to split it into two steps, but be cautious about the analysis/interpretation if there are confounding effects). Does analyzing the overall (i.e., without interaction) climatic effects in the models in line 106 yield the same results? I just checked the R script, and if I'm not mistaken, the climate effect is not significant (climate Chisq 6.1739 5 P-value 0.2897) for the model `hill_BF_rot ~ climate + crop_id + [...]`. I also noticed in the code that models 1, 2, and 3 are based on the ROTATION database, while models 4, 5, and 6 use the CROP dataset. Should this be clarified more clearly in the text?

9. Figure 2. The density maxima of the violin plots do not always correspond to the mean estimate (red points). I think this might be because you are using marginal means (with the emmeans package to calculate the red point), which consider the effects. And you display the 'raw' values for the violin plot. Is that right?

10. When you perform post-hoc tests, you adjust confidence intervals using the Bonferroni method (even if it's not specified in your code, the cld function automatically readjusts confidence intervals of emmeans results with this method), and then you control the 1-alpha error with FDR, is that correct? Is this a common method? "How do these two types of corrections interact?"

11. There's an extra word in the sentence?: "For herbicide use, an alternative model focusing on diversity of sowing periods was investigated (model 13, Supplementary Table 1) but was not as supported by the data (AIC=31253) as the one focusing on crop functional diversity (AIC=31211)."

12. There is a missing space in line 431. Also, there is a spacing issue in line 87.

13. In Table 3 of the appendix, what do the infinity symbols for degrees of freedom mean? Is the model rank-deficient?

14. "Results highlighted that temporal crop diversification could reduce total pesticide use in all dominant field crops in which pesticides are commonly applied (7/12), except for winter straw cereals (5/12)." The fractions 7/12 and 5/12 are not very clear.

15. "Weisberger et al 17 197. found that diversifying simple rotations (i.e. monocultures or two-year rotations) reduced weed density by 49%, although no significant effect was found on weed biomass." Please specify that this is a meta-analysis, and provide the number of data analyzed. Is this meta-analysis global (i.e., not focused on a single region)?

16. In the script:

- You haven't displayed the code snippet for conducting the PERMANOVA?
- Line 621. Error: MOD_IPT_TOT instead of MOD_TFI_TOT.
- Was the selection of random effects for the models done beforehand (not displayed here)?
- You forgot the line library(geodata).?"

Reviewer #3 (Remarks to the Author):

Reviewer 3 here.

Another impressive revision. I think the manuscript is greatly improved. Most of my previous concerns have been clarified or appropriately caveated. There are a few minor language usage issues (e.g., L125 "stacked", L226 "allowed"), so one more careful read may help. Otherwise, I believe this manuscript can advance in the editorial process

Reviewer #1

General comments:

Thank you to the authors for their transparency efforts in providing code for the analysis and databases!

Once again, the authors have put in a lot of work into revising the manuscript based on the comments from various reviewers. As a result, both the text and the analyses have evolved significantly since the last version. The text is now simpler, clearer, and more convincing. The results are well-described, and the statistical analyses behind them appear convincing to me. I no longer have any obstacles to the publication of this paper. This series of exchanges with the authors has been interesting and enriching.

I have carefully read the responses point by point, and the authors' responses seem relevant and well-argued.

Thanks again for this thorough review and the time spent doing it. We are very grateful and recognized your work in the acknowledgement section. We provide a point-by-point answer for the remaining comments.

Comments	Response
1. Reference 1 is about Sweden. Should you find a more global reference? Is the idea behind your sentence to make a general observation?	Two additional references were added to support this statement :  Aguilar, J. et al. Crop species diversity changes in the United States: 1978–2012. PloS one 10, e0136580, doi:https://doi.org/10.1371/journal.pone.0136580 (2015). Bradshaw, B., Dolan, H. & Smit, B. Farm-level adaptation to climatic variability and change: crop diversification in the Canadian prairies. Clim. Change 67, 119-141, doi:https://doi.org/10.1007/s10584-004-0710-z (2004).
2. Line 83. This is the first time the term "through crop composition" appears. What exactly does it represent? A number of species? Diversity?	We agree that the reader would probably welcome a precision. Hence, we added "the crop species present and their relative proportion in space". We added "in space" to make sure the reader understands that this relates to space for time substitution.
3. You used PERMANOVA because it does not assume homoscedasticity and is more suitable for non-normally distributed data? Is this the case for your data?	PERMANOVA is a non-parametric alternative to MANOVA. It is appropriate for sets of variables that do not meet the assumptions of MANOVA, namely multivariate normality. We checked multivariate normality of the crop composition table with the R function mvn() (by default Henze-Zirkler test of multivariate normality) and H0 was rejected (no support for multivariate normality). This was highly expected considering the high proportion of zeroes (minor crops) in the dataset, so this statistical step was not included in the manuscript (but was included in the R code). <pre>> mvn(ROTATION2[,c(10:64,66:70,73:100,103,110:111)]) # collinear crops (thyme, matricaria) and crops absent were removed \$multivariateNormality Test HZ p value MVN 1 Henze-Zirkler 38.75324 0 NO</pre> However, PERMANOVA can be sensitive to homoscedasticity in the case of unbalanced designs. Therefore, we checked homoscedasticity using

	the betadisper() function, a multivariate analogue of Levene's test for homogeneity of variances. Once again, heteroscedasticity was expected considering the fact that the vast majority of cropping systems maintain the same crop composition over time, while others slightly vary. Heteroscedasticity can also be driven by the different number of observations per cropping system identifier. Finally, with roughly 750 cropping systems monitored over time, variability was bound to differ among certain pairs of cropping systems. Hence, we decided to be transparent and report the lack of homoscedasticity. Furthermore, we divided the dataset into 10 balanced subdatasets (i.e. same number of observations per cropping system identifier; 2, 3,..., 11) and carried out the PERMANOVA analyses on each of these balanced subdatasets to investigate the robustness of results to lack of homoscedasticity, initially warranted due to the lack of balanced design. Another possibility would have been to present the groups on a multivariate ordination (eg PCoA) but the number of groups is simply too important; the resulting ordination is overcrowded (it is included in the R script), cropping systems would have to be split into 10 bins and projected separately, resulting in a high number of graphs. Statistical elements concerning multivariate homogeneity of variance were included in Methods and the corresponding permanova section of the Results.
4. Line 94. A double space?	It appeared so because the text was justified but isn't the case.
5. "Crop functional diversity was higher in the Deteriorated Oceanic (2.19)". Is 2.19 the mean, I assume? Or the median? Should confidence intervals be included?	Thanks for this precision. We now specify that these values correspond to the mean. Confidence intervals were however not included because – if we were to add them here, we would have to add them all throughout the text (drivers of crop diversity section) and this greatly reduces readability of the text. All these values are shown in violin plots (Supplementary Material) in which least square means and associated confidence intervals are included.
6. When you say, "To investigate the effect of crops on crop taxonomic and functional diversity, and their interaction (i.e. crop diversity)," please be careful. It makes me think directly of an interaction that you introduce into the model (i.e., as an explanatory variable). However, if I'm not mistaken, you are referring to the 'global' effect of crop diversity (i.e., as a response variable).	You are right. Thank you for catching this. As an explanatory variable, species diversity is modelled by the interaction between taxonomic and functional diversity but here species diversity is analyzed as a response variable, which is simply defined by the product between taxonomic and functional diversity. We hence used (all throughout the text) the term "interaction" when referring to species diversity as an explanatory variable and "product" when referring to a response variable.
7. The crop taxonomic and other indicators are studied at the system level. Please clarify.	We honestly do not see what we can add more without repeating ourselves or reducing the readability of the manuscript. We specify the scale at which diversity indicators were computed at the end of the introduction. We specify scale of analysis (one observation=...) in the material and methods section, in the figure captions, and in the Supplementary table 1 with all models.

8. In line 91, why did you perform these models: "(To investigate the effect of climatic region on crop taxonomic and functional diversity, and their interaction (i.e. crop diversity), one generalized linear mixed effect model (GLMM) with climatic region as the only fixed effect was fit for each of the three response variables," when you also conducted these models: "To investigate the effect of crops on crop taxonomic and functional diversity, and their interaction (i.e. crop diversity), one GLMM with climatic region and crop as fixed effects was fit for each of the three response variables." If I understand correctly, the climatic variable is present in both models. Could you analyze the effect of climatic variables in the model described in line 106? There is likely no independence of data, i.e., crops are not evenly distributed. (This might be the reason you decided to split it into two steps, but be cautious about the analysis/interpretation if there are confounding effects). Does analyzing the overall (i.e., without interaction) climatic effects in the models in line 106 yield the same results? I just checked the R script, and if I'm not mistaken, the climate effect is not significant (climate Chisq 6.1739 5 P-value 0.2897) for the model <code>hill_BF_rot ~ climate + crop_id + [...]</code>. I also noticed in the code that models 1, 2, and 3 are based on the ROTATION database, while models 4, 5, and 6 use the CROP dataset. Should this be clarified more clearly in the text?	Indeed, climatic region is present in all models, as shown in Supplementary Table 1. However, depending on the models, climatic regions cannot be interpreted identically. First of all, climatic region can only have an influence on crop diversity by modulating crop composition (i.e. allowing the introduction or not of specific crops). Climatic region -> crops -> crop diversity Models 1 to 3 aim to answer: Is crop diversity influenced by climatic region (irrespective of which crop species are included or not in each climatic region) ? Here, the effect of climatic region is confounded with the effect of crop. Models 4 to 6 aim to answer: Is crop diversity driven by any of the 16 main crops of the dataset? Here climatic region is merely included to control for the fact that certain regions might promote a subset of minor crops (not included in the 16 considered). Therefore, it appears more sensical to test for differences in crop diversity between climatic regions using models 1 to 3. Concerning scale of analysis, please refer to the previous comment. We honestly tried adding precisions in the Results section but all proposals seemed to reduce readability.
9. Figure 2. The density maxima of the violin plots do not always correspond to the mean estimate (red points). I think this might be because you are using marginal means (with the emmeans package to calculate the red point), which consider the effects. And you display the 'raw' values for the violin plot. Is that right?	Yes, you are entirely right. Violon plots are created based on the observed data, whereas mean estimates (red points) represent adjusted means. We believe that the two are complementary. Violin plots are a mere projection of the data that was used in the models whereas red points correspond to model estimates, which account for the fact that not all points are independent and the influence of climatic region. We added a sentence in the caption to make this point clear.
10. When you perform post-hoc tests, you adjust confidence intervals using the Bonferroni method (even if it's not specified in your code, the <code>cld</code> function automatically readjusts confidence intervals of emmeans results with this method), and then you control the 1-alpha error with FDR, is that correct? Is this a common method? "How do these two types of corrections interact?"	We understand how this may be confusing. This arises from the fact that confidence intervals are displayed (as these are understood by a wide audience) but that pairwise contrasts were carried out to highlight differences between groups. The Bonferroni correction applies to confidence intervals, whereas the FDR correction applies to pvalues in multiple testing. There is no interaction between the two corrections because they are not applied to the same statistic.

11. There's an extra word in the sentence?: "For herbicide use, an alternative model focusing on diversity of sowing periods was investigated (model 13, Supplementary Table 1) but was not as supported by the data (AIC=31253) as the one focusing on crop functional diversity (AIC=31211)."	We do not believe that any word is extra. However, we rephrased the sentence for greater clarity as we believe that – if pointed out – something was not clear enough. The sentence was changed to: "...but was not as parsimonious as the one focusing on crop functional diversity..."
12. There is a missing space in line 431. Also, there is a spacing issue in line 87.	L87 : this is due to text justification L431 : a space was added before the beginning of the next sentence
13. In Table 3 of the appendix, what do the infinity symbols for degrees of freedom mean? Is the model rank-deficient?	No this does not relate to rank deficiency. Using infinite degrees of freedom essentially means that the distribution of the test statistic is approximated using a normal distribution rather than a t-distribution. This approximation can be useful in situations where calculating exact degrees of freedom is challenging (as GLMMS), and it simplifies the distributional assumptions. Quoting Ben Bolker "Finite-size corrections are rarely considered for GLMs or GLMMs, and for GLMMs in particular there is little theoretical work that would even specify how to compute them. That's why emmeans report df as Inf. df in emmeans output represents the "denominator degrees of freedom" (i.e. the nu2 value you would use if testing against an F distribution $F_{\{nu1, nu2\}}$), which is something like (number of observations - number of parameters estimated) for simple (non-mixed) models like a linear regression or simple ANOVA, but which is considerably harder to define for multilevel models (i.e. linear mixed models). For generalized linear (and linear mixed) models, it gets even worse." It is specified in the caption that "Inf stands for infinite degrees of freedom and refers to the fact that asymptotic results were used."
14. "Results highlighted that temporal crop diversification could reduce total pesticide use in all dominant field crops in which pesticides are commonly applied (7/12), except for winter straw cereals (5/12)." The fractions 7/12 and 5/12 are not very clear.	Thanks for pointing this out. We changed into (7 out of 12 crops) and (5 out of 12).
15. "Weisberger et al 17 197. found that diversifying simple rotations (i.e. monocultures or two-year rotations) reduced weed density by 49%, although no significant effect was found on weed biomass." Please specify that this is a meta-analysis, and provide the number of data analyzed. Is this meta-analysis global (i.e., not focused on a single region)?	This is a global meta-analysis based on 298-paired observations from 54 studies across six continents. We added this information.
16. In the script: - You haven't displayed the code snippet for conducting the PERMANOVA? - Line 621. Error: MOD_IPT_TOT instead of MOD_TFI_TOT. - Was the selection of random effects for the	Sorry for this oversight. The code is now included in the R script, along with multivariate normality and homoscedasticity. L621: Yes! Good catch. We appreciate it.

models done beforehand (not displayed here)?
- You forgot the line library(geodata).?"

Random effects were not selected. They were defined a priori based on expert knowledge. We included this information in Methods.

The map was initially created with the sf and raster packages. Yet, the getData function that we used to download the geographic data will be remove in the future version of raster. We hence change the code using the rnaturalearth package. The line library(rnaturalearth) was added.

Reviewer 3

Another impressive revision. I think the manuscript is greatly improved. Most of my previous concerns have been clarified or appropriately caveated. There are a few minor language usage issues (e.g., L125 "stacked", L226 "allowed"), so one more careful read may help. Otherwise, I believe this manuscript can advance in the editorial process

A sincere thanks for your thorough review and all your proposals, which we believe significantly improved the quality of manuscript. We recognize the time spent and are grateful for that. We acknowledge your contribution in the acknowledgement section. We changed the two words you pointed out (allowed appeared four times and was changed every time, stacked was changed into combined). Cheers.